# Hierarchically Encapsulated Representation for Protocol Design in Self-Driving Labs

**Yu-Zhe Shi**[1][*], **Mingchen Liu**[2][*], **Fanxu Meng**[1], **Qiao Xu**[1], **Zhangqian Bi**[2], **Kun He**[2], **Lecheng Ruan**[1][✉], **Qining Wang**[1][✉]

[1] Department of Advanced Manufacturing and Robotics, Peking University
[2] School of Computer Science and Technology, Huazhong University of Science and Technology
[*]Equal contribution   ✉ ruanlecheng@ucla.edu, qiningwang@pku.edu.cn

## Abstract

Self-driving laboratories have begun to replace human experimenters in performing single experimental skills or predetermined experimental protocols. However, as the pace of idea iteration in scientific research has been intensified by Artificial Intelligence, the demand for rapid design of new protocols for new discoveries become evident. Efforts to automate protocol design have been initiated, but the capabilities of knowledge-based machine designers, such as Large Language Models, have not been fully elicited, probably for the absence of a systematic representation of experimental knowledge, as opposed to isolated, flatten pieces of information. To tackle this issue, we propose a multi-faceted, multi-scale representation, where instance actions, generalized operations, and product flow models are hierarchically encapsulated using Domain-Specific Languages. We further develop a data-driven algorithm based on non-parametric modeling that autonomously customizes these representations for specific domains. The proposed representation is equipped with various machine designers to manage protocol design tasks, including planning, modification, and adjustment. The results demonstrate that the proposed method could effectively complement Large Language Models in the protocol design process, serving as an auxiliary module in the realm of machine-assisted scientific exploration.

## 1 Introduction

The rapid advancement of Artificial Intelligence (AI) models for the assistance of scientific discovery (Wang et al., 2023b) has precipitated an increased demand for rapid iteration of ideas, from the generation to the verification of hypotheses. Although AI models have expedited the process of hypothesis generation, the validation phase still requires intensive empirical experimentation from human. The concept of self-driving laboratory has been introduced to substantially accelerate the validation process, in organic chemical synthesis (Mehr et al., 2020; Burger et al., 2020), cell biology for medical research (Kanda et al., 2022), and novel material discovery (Szymanski et al., 2023). With the expertise and effort of experimental scientists and automation engineers, mobile robots and Internet of Things (IoT) pipelines are configured to perform a sequence of actions in accordance with a detailed description of the specific experimental procedure, referred to as the *protocol*.

While existing protocols suffice for some experimental tasks, discovery processes often demand a higher degree of specificity, including: (i) confirmation of unverified experimental objectives to seek specific findings; (ii) testing parallel hypotheses or solutions; and (iii) replication of established experiments within the constraints of available laboratory resources. These necessitate the *design* of new protocols, going beyond the reuse of existing ones available in the protocol databases. Particularly, this includes the *planning* of novel protocols, and the *modification* and *adjustment* of current protocols as appropriate, respectively. Unfortunately, self-driving laboratories currently only execute isolated and duplicated experimental skills (Bédard et al., 2018; Steiner et al., 2019), or pre-specified protocols with sequential actions (Rohrbach et al., 2022; Manzano et al., 2022). Any innovation in protocols imposes intensive manual design burden (McNutt, 2014; Baker, 2016), potentially becoming a bottleneck in accelerating scientific discovery. Consequently, there is a quest for the automatic design of protocols tailored to specific goals for self-driving laboratories.

Designing new protocols is a non-trivial task even for human scientists. Novice scientists tend to adhere strictly to established protocols and may be at a loss when faced with the need for variations,

from minor adjustments like different available devices to more significant shifts in the overall experimental goal. In contrast, veteran scientists typically have the capability to create or modify protocols as needed, from variations in available resources (*"what I have"*) to desired outcomes (*"what I want"*), even in situations where a similar protocol was not encountered before.

The distinction arises because veteran scientists possess a *systematic* understanding of every ingredient and procedure, contextualizing them globally within the domain of experiment. They know *"what kind of ingredient is used for what purposes"* and *"what kind of operation is used under what conditions'*, while novice scientists mechanically memorize the sequential execution orders and corresponding parameters in a local context. This systematic understanding, or *conceptual knowledge* (Ryle & Tanney, 1949), includes the background knowledge of ingredients and atomic operations, as well as the relationships between them. Experienced experimental scientists develop such conceptual knowledge as a *representation* for protocol design (McCarthy, 1959), which serves as the vehicle for reasoning processes. Reasoning over conceptual knowledge leverages the rich context of generalized, abstracted concepts of ingredients and operations rather than specified, instantiated ones, which spans a semantic space where originally isolated dots are connected with each other, thereby enhancing the simplicity and flexibility of protocol design (Boden, 1980; Newell, 1982). In summary, veteran scientists' capability to design new protocols stems from an appropriate representation of background knowledge that supports reasoning processes (see Fig. 1A).

To implement automatic protocol design on machines, a reasonable choice may be leveraging a Large Language Model (LLM). Trained on extensive corpora, including scientific documents, LLMs possess the potential to facilitate protocol design with the corresponding background knowledge (AI4Science & Quantum, 2023). Recently, researchers have made beneficial attempts to design new protocols using LLMs based on descriptions of new experimental goals (Boiko et al., 2023; M. Bran et al., 2024). Regrettably, benchmarking results indicate that the expected capability of LLMs in protocol design is not fully elicited (O'Donoghue et al., 2023). One significant limitation is that LLMs excel at generating new protocols similar to existing ones, *i.e.*, protocols with similar sequential execution orders, but fail to generate those with distinct dependency distributions. This limitation hampers LLMs in scenarios where experimental goals change in high intensity. Another limitation is that the generated protocols sometimes lose critical configuration details for operation execution, necessitating manual correction. These empirical evidences suggest that LLMs exhibit limitations akin to those of novice human experts, implying that LLMs may necessitate a more suitable representation of background knowledge to fully unleash their potential in protocol design.

Protocol design is a multi-faceted, multi-scale effort requiring the integration of information from different perspectives, from low-level to high-level. This information includes detailed configurations of each atomic operation, temporal relationships between atomic operations, the scope of application for atomic operations with the same reference name, and the reactive relationships between reagents and operations. While LLMs undoubtedly capture such knowledge from their training corpora, the pieces of knowledge remain isolated, unorganized, and not articulated. These flatten background knowledge, rather than conceptual knowledge, hinders LLMs from *flying over* a global view of the novel objectives and *diving into* the details of operations. Therefore, we propose developing a **multi-faceted and multi-scale representation** for protocol design that provides the designer, such as LLMs, with a vehicle to reason over conceptual knowledge of ingredients and procedures.

We draw inspiration from both cognitive science literature on rationality (Monsell, 2003; Griffiths, 2020), which suggests that we cannot consider information from different views and scales in a single thread (Shi et al., 2023a). We also learn from computer science literature on hierarchical abstraction (Liskov, 1987), which indicates that higher-level abstraction semantics possess more powerful expressivity compared to their lower-level counterparts (Abelson & Sussman, 1996; Hopcroft et al., 1996). Combining these insights, we suggest that our desired representation should encapsulate information of different granularities in corresponding hierarchies of abstraction, gaining global design insights with higher-level semantics while completing execution configurations with lower-level semantics. Specifically, we investigate three levels of encapsulation (see Fig. 3B). Starting from the set of original protocols, namely the basic level, we have (i) **protocol element instantialization**, which decomposes full protocols into instance operations with attributes, within the local context of the specific protocol, resulting a structural representation of the elementary information; (ii) **function abstraction**, which offers an operation-centric view that generalizes the precondition, postcondition, and execution configurations of each operation in the global context of the experiment domain, resulting a sequential representation of the operations; (iii) **model abstraction**, which offers an reagent and intermediate product centric view that unifies the status transitions in the global context of the experiment domain, resulting a continuous representation of the experimental environ-

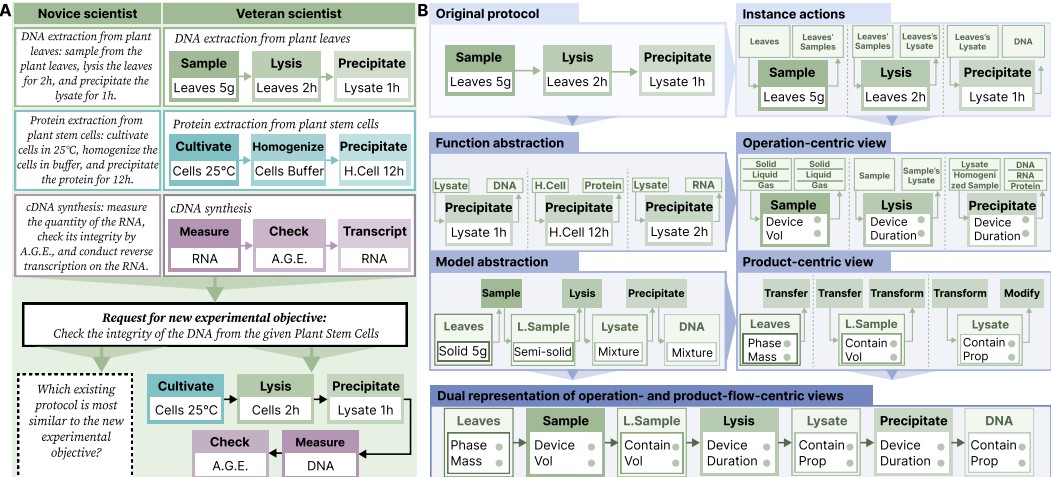

Figure 1: **The representations for protocol design. (A)** The example of protocol design by novice and veteran experimental scientists. **(B)** The hierarchies of our proposed representation, from original full protocol representation, to dual representation of operation- and product-flow-centric views.

ment. This hierarchical structure provides the designer with a representation to consider all possible associations among operations, among products, and between operations and products, with a high degree of freedom, by disentangling originally intertwined information. We implement the representation using Domain-Specific Languages (DSLs). The hierarchical syntax of DSLs maintains both the abstract semantics at the high-level and the precise information at the low-level. Furthermore, the *compositionality* of DSL syntax facilitates the flexible protocol designs, addressing the "flying over global views" requirement; while DSL program verification over the generated protocols upholds their *soundness* and *completeness*; addressing the "diving into details" requirements.

However, the proposed representation does not come without drawbacks—it can be highly dependent on domain-specific knowledge (Mernik et al., 2005; Fowler, 2010). The distributions of reagents, operations, and execution dependencies vary significantly across different domains in experimental sciences, such as *Genetics*, *Medical*, *Bioengineering*, and *Ecology*. Manually crafting DSLs specialized for these domains requires deep integration between domain experts and programming language experts, which is labour-intensive, case-by-case, and costly (Shi et al., 2024a;b). This obstacle hinders the application of our representation to a broader set of domains (Shi et al., 2024d). To make the representation specification more affordable, we develop an algorithm that conducts multi-hierarchy encapsulation automatically driven by the domain-specific corpus of existing protocols. Ultimately, we may be able to take a critical step toward closing the loop of *autonomous* scientific discovery by establishing these two building blocks: (i) the automatic generation of **representation** for protocol design; and (ii) the automatic **designer** working on the representation.

Our contributions in this work are three-fold: (i) we identify the problem of representation for protocol design and develop a hierarchically encapsulated representation for protocol design (Sec. 2); (ii) we propose a data-driven algorithm that automatically generates the representation for protocol design specialized for the domain of application (Sec. 3); and (iii) we demonstrate the utility of the resulting representation by conducting protocol planning, modification and adjustment tasks using a variety of machine designers across different domains (Sec. 4). This further indicates that our proposed automatic representation generation approach possesses the potential to function as an auxiliary module for LLMs, enhancing their capability on protocol design.

## 2 REPRESENTATION FOR PROTOCOL DESIGN

In this section, we describe our representation for protocol design (see Fig. 1B). We first formulate the basic protocol design problem in Sec. 2.1. Afterwards, starting from the original full protocol, we introduce the three hierarchies of representations: (i) structural representation, *i.e.*, instance actions with attributes (Sec. 2.2); (ii) sequential operation-centric representation, *i.e.*, function abstraction (Sec. 2.3); and (iii) continuous product-flow-centric representation, *i.e.*, model abstraction (Sec. 2.4). Furthermore, we describe how the dual representation of operation-centric and product-flow-centric views reciprocatively facilitates the verification of the designed protocols in Sec. 2.5.

## 2.1 THE PROTOCOL DESIGN PROBLEM

Protocol design problem PD $= (\Phi \mid \omega^*, \mathcal{P}, \Omega)$ is generating a desired protocol $\Phi$ given the **new coming** experimental objective $\rho$, domain of experiment $\mathcal{P}$, and available reagents $\Omega$. A protocol $\Phi = \langle \varphi_1, \varphi_2, \dots \rangle$ is a sequence of experimental steps $\varphi_t$. An experimental objective $\omega^*$ is the expected final product of the experiment. Experimental objectives can range from preparing a desired product, to testing the significance of a specific hypothesis and detecting a predicted behavior, with the latter two potentially followed by additional standalone steps for property test, observation, and interpretation. We denote domains of experiment as $\mathcal{P}$, which influences the distributions of protocols by means of the distributions of operations, reagents, and execution orders, *etc*. The set of available reagents $\Omega$ includes originally accessible reagents and excludes those requiring production.

## 2.2 INSTANCE ACTIONS WITH ATTRIBUTES

Protocols are originally represented in Natural Language (NL), which is the representation suitable for humans' comprehension, but not for machines (Bartley et al., 2023). Without a syntax decomposing a NL-based protocol into information elements precisely, machines are likely to capture only overall, coarse-grained information of protocols and may only retrieve within existing protocols for the one that is most similar to the new experimental objective. Consequently, according to the standards and conventions of experimental sciences (Baker, 2021), the prerequisite of representation for a machine protocol designer should be a structural representation which decompose NL-based protocols into instance actions with attributes $\{\varphi_t \mid (\varphi_t^{\text{prec}}, \varphi_t^{\text{post}}, \varphi_t^{\text{exec}})\}$. The instance actions are decomposed by execution order and their attributes are the exact context for their execution, namely the precondition $\varphi_t^{\text{prec}}$, *i.e.*, the availability of resources required for this action, postcondition $\varphi_t^{\text{post}}$, *i.e.*, resulting product of the operation, and execution configurations $\varphi_t^{\text{exec}}$. Execution configurations includes the configuration parameters and their corresponding values, *e.g.*, the device for conducting the operation and required experimental conditions such as duration, acidity, and lightening. An instance action can be reusable in another protocol once the execution context is matched.

With such reusability, we are on the first time to have *building blocks* for constructing a new protocol rather than retrieving existing ones. These building blocks capture fine-grained execution configuration parameters through maintaining the nested data structures of key-value pairs. This structural representation serves as a syntactic constraint on the preciseness of designed protocols. Practical attempts have been made echoing this idea (O'Donoghue et al., 2023; Leonov et al., 2024).

## 2.3 OPERATION-CENTRIC VIEW WITH FUNCTION ABSTRACTION

The reusability of instance actions with attributes is highly limited, as their semantics are highly specified in the low-level. The total amount of the instance actions can be extremely high, *i.e.*, about 150K per domain, thus the probability of the exact matching between execution contexts can be extremely low. Consider the three different instance actions with attributes *"Homogenization of mouse liver tissue using a bead mill"*, *"Homogenization of bacterial cell suspension using an ultrasonic homogenizer"*, and *"Homogenization of bacterial air samples using a nebulizer"*. Although they come with totally different preconditions, postconditions, and execution configurations, particularly the required device varying according to the phase of the experimental subject, they share the semantic identifier *"Homogenization"* for reference. Sharing semantic identifier indicates that these instance actions share the same *purpose* on the semantics level. In experimental sciences, *"Homogenization"* always refers to the breakdown of a sample into a uniform mixture. Whether it's tissue, cell suspension, or gas doesn't change the purpose of the operation. This is critical for protocol design, since it essentially requires satisfying the ultimate goal through a series of subgoals. Therefore, the desired representation should generalize the semantics of operations to any possible contexts in the corresponding domain of experiment, rather than only specific contexts.

We implement such generalization by encapsulating varied instances of preconditions, postconditions, and execution configurations into an *interface* for the operation. Namely, we refer to an operation with semantic identifier $\varphi$ through an interface $\phi$ to a set of execution contexts, in the form of $\langle \varphi \mapsto \phi \mapsto \{(\varphi^{\text{prec}}, \varphi^{\text{post}}, \varphi^{\text{exec}})\} \rangle$. The operation $\varphi$ can be grounded to a corresponding instance action in any matched execution contexts, echoing *modular design* (Hirtz et al., 2002). The reusability of encapsulated operations comes with greater significance than that of instance actions, as there are only about 1K operations per domain in total, which is only $1/150$ of that of instance actions. As flexible building blocks, operations can be easily fitted into any breakpoints with suitable preconditions and postconditions in the constructing experiment sequence. This sequential representation

of the operations serves as a semantic constraint on the compact permissible set of primitives for protocol design (Shi et al., 2023b), maintaining both degree of freedom and correctness.

## 2.4 PRODUCT-FLOW-CENTRIC VIEW WITH MODEL ABSTRACTION

Sequence of operations make up of protocols. However, operations are the methods to realize rather than the objectives to achieve. For experimental objectives of testing, preparing, or detecting (Schwab & Held, 2020), the common focus is always the specific status of final product, not the operations. Starting from initial reagents, the status of product flow is manipulated step-by-step by the operations, till the final product. Unfortunately, the information of product status transition is *latent* in protocols and is *twisted with* descriptions of experimental steps. For the operation-centric view, the transitions of product flow statuses remains a *black box environment*. For example, the operation description *"Centrifuge the tubes at 15,000 x g for 20 minutes"* does not directly reveal the transition from product in mixture status to products in distinct phases. The lack of coherent tracking of the product flow is problematic of protocol design, as the product flow holds *spatial-temporal invariance*, just the same as the general physical environment. Status transitions of the product flow are primarily caused by the effects of operations, thereby it serves as the *invariant* in executing the protocol from the perspective of programming. Therefore, the desired representation should also serve as the *model* interacting with the sequence of operations.

To disentangle product status from their latent representation in the operation-centric view, we propose an explicit product flow centric view that tracks the status of the product flows with detail, such as component, volume, container, and other physical and chemical properties of the product, and also the predecessor operation that yields the product and the successor operation that takes the product as input. Each product flow unit, *i.e.*, one individual component in the product flow between two adjacent steps, is an instance with attributes $\{\omega_t \mid (\omega_t^{\text{pred}}, \omega_t^{\text{succ}}, \omega_t^{\text{prop}})\}$. Analogous to the generalization of operations' semantics, product flow units share commonalities between components with the same semantic identifier for reference—they may share a specific range of predecessor operations $\omega^{\text{pred}}$ and successor operations $\omega^{\text{succ}}$, and a selected set of key properties to consider $\omega^{\text{prop}}$. For example, the *"supernatant"* is usually generated by a *"centrifugation"* operation, passing into *"filtration"* or *"spectrophotometric analysis"*, and focusing on the properties *acidity* and *viscosity* rather than other possible properties. Thus, we encapsulate the information of contexts and properties into the semantics of product flow units, in the form of $\langle \omega \mapsto (\omega^{\text{pred}}, \omega^{\text{succ}}, \omega^{\text{prop}}) \rangle$. As solid pipelines bridging the building blocks, product flow units can verify the coherency of the entire designed protocol. This continuous representation of the environments serves as a program verifier, checking the prerequisite and simulating the effect of each operation, alleviating unpredictable behaviors among the interaction between operations and product flows.

## 2.5 RECIPROCATIVE VERIFICATION OVER THE DUAL REPRESENTATION

The dual representation of operation-centric and product-flow-centric views intrinsically equips with a verification mechanism through a reciprocative process akin to two interacting threads. The first thread focuses on verifying the operation flow, taking as input an operation $\varphi_t$ along with its precondition $\varphi_t^{\text{prec}}$ and postcondition $\varphi_t^{\text{post}}$. The second thread handles the verification of the product flow, taking as input a product $\omega_t$ along with its predecessor operation $\omega_t^{\text{pred}}$ and successor operation $\omega_t^{\text{succ}}$.

Specifically, for the operation verification (corresponding to OFVERIFICATION in Alg. 1), we ensure that each operation can be correctly executed given its input reagents and that it yields the expected output products. This involves checking that the preconditions are satisfied by the available products from

---

**Algorithm 1** Reciprocative Verification

**procedure** OFVERIFICATION($M, \varphi$)
   ▷ *Check that the pre/ post conditions are met*
   CHECKOPCONDITIONS($\varphi, \varphi^{\text{prec}}, \varphi^{\text{post}}$)
   **if** $\varphi^{\text{prec}} \subseteq M(\Omega)$ **then**
      $M(\Omega) \leftarrow (M(\Omega) \setminus \varphi^{\text{prec}}) \cup \varphi^{\text{post}}$
      ▷ *Proceed to verify each output product*
      **for** product $\omega$ in $\varphi^{\text{post}}$ **do**
         PFVERIFICATION($M, \omega$)
**procedure** PFVERIFICATION($M, \omega$)
   ▷ *Check necessary properties of the product*
   CHECKPROPERTIES($\omega, \omega^{\text{prop}}, \varphi'^{\text{req}}$)
   ▷ *if required by subsequent operations.*
   **if** $\exists \varphi' \ s.t. \ \omega \in \varphi'^{\text{prec}}$ **then**
      ▷ *Verify operations using the product*
      OFVERIFICATION($M, \varphi'$)

---

preceding operations and that the postconditions are well-defined for subsequent use. Concurrently, the product flow verification (corresponding to PFVERIFICATION in Alg. 1) involves tracking each

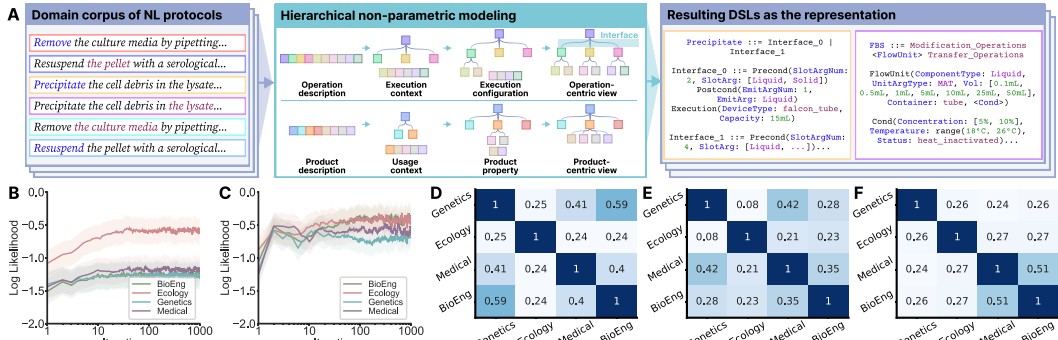

Figure 2: **Diagram of automatic representation generation. (A)** Illustration of the workflow. **(B)** Convergence curve of automatic function abstraction. **(C)** Convergence curve of automatic model abstraction. **(D-F)** Confusion matrices on operation distribution (D), product distribution (E), and device distribution (F), between DSLs across domains. Correlation scores are low except the ones along the diagonals, indicating the significant inter-domain distinctions between the resulting DSLs.

unit of product flow through the protocol. We verify that the product is generated by the specified operation and that it possesses the necessary properties $\omega_t^{\text{prop}}$ for consumption by the next operation.

The interaction between these two threads forms a feedback loop where the verification of operations and products mutually inform and constrain each other. This reciprocative method allows us to iteratively refine the protocol, ensuring that each step is both operationally feasible and chemically coherent. LLMs are employed to implement the functions CHECKOPCONDITIONS and CHECK-PROPERTIES, extracting and verifying operation conditions and product properties from natural language protocol descriptions through instruction-following in-context learning (Wei et al., 2021; Brown et al., 2020). For the prompts employed, readers are referred to Appx. D.6.

# 3 AUTOMATIC REPRESENTATION GENERATION

In this section, we describe the proposed data-driven algorithm to automatically generate the hierarchically encapsulated representation for protocol design (see Fig. 2A). We first define the problem of generating the desired representation by means of DSL design (Sec. 3.1). We then introduce methods for generating operation-centric (Sec. 3.2) and product-flow-centric (Sec. 3.3) DSL views.

## 3.1 THE REPRESENTATION GENERATION PROBLEM

We denote the problem of generating the representation for protocol design within a given domain as $\text{RG} = (\{\langle\varphi\rangle, \langle\omega\rangle\} \mid \mathcal{P}, \mathcal{C})$. The representation is a DSL with language features accommodating both the operation-centric program view $\langle\varphi\rangle$ and the product-flow-centric program view $\langle\omega\rangle$. The domain-specific corpus $\mathcal{C} = \{\Phi_1, \Phi_2, \ldots, \Phi_{|\mathcal{C}|}\}$ consists of **existing** protocols published in top-quality journals within the corresponding experimental domain. The source and profiles of $\mathcal{C}$ of each domain is detailed in Appx. E.1. We can obtain instance action with attributes based on $\mathcal{C}$ in a straightforward way through NL information extraction (see Appx. D.3 for implementation details). The prior knowledge of operations and products, $p(\varphi)$ and $p(\omega)$, including the basic syntax of the key-value structures and the elementary taxonomies, is derived according to the general common-sense of experimental sciences, as aforementioned in Sec. 2. Specifically, the problem essentially aims to fit the joint distribution models $p(\varphi, \phi, \varphi^{\text{prec}}, \varphi^{\text{post}}, \varphi^{\text{exec}})$ and $p(\omega, \omega^{\text{pred}}, \omega^{\text{succ}}, \omega^{\text{prop}})$ with domain-specific corpus $\mathcal{C}$ given prior knowledge $p(\varphi)$ and $p(\omega)$.

## 3.2 AUTOMATIC FUNCTION ABSTRACTION

The key challenge of encapsulating the operation-centric view is to aggregate all possible execution contexts for an operation, and then generalize the contexts to the interface. If we keep each of the use case as one single instance of the interface, which can be in thousands regarding one operation, the generalization is meaningless. Since there is no prior knowledge about the interface in advance, we develop the algorithm following the idea of non-parametric modeling, *i.e.*, Dirichlet Process Mixture Model (DPMM), resulting in flexible identification of interface instances.

**Hierarchical non-parametric modeling** As we must handle information coming in different granularities, from interface structures to values of parameters, we choose to model the operations in a hierarchical fashion. Compared with the flatten spectral clustering approach developed by Shi et al. (2024a), which compresses all information of an operation into a embedding vector, our modeling is competent for considering information at different levels comprehensively. We carefully adopt the prerequisite that the interface is generated subject to the operation, preconditions, postconditions, and execution configurations are generated subject to the interface, and the value of configuration parameters are generated subject to their corresponding keys. Thus, we have the model:

$$
\begin{aligned}
&p(\varphi, \phi, \varphi^{\text{prec}}, \varphi^{\text{post}}, \varphi^{\text{exec}}, \varphi^{\text{exec-v}}) \\
&= p(\varphi^{\text{exec-v}} \mid \varphi, \phi, \varphi^{\text{exec}}) p(\varphi^{\text{exec}} \mid \varphi, \phi) p(\varphi^{\text{prec}} \mid \varphi, \phi) p(\varphi^{\text{post}} \mid \varphi, \phi) p(\phi \mid \varphi) p(\varphi),
\end{aligned}
\tag{1}
$$

where $\varphi^{\text{exec-v}}$ denotes the values of configuration parameters. Within each iteration of the DPMM process, we sample the variables level-by-level. Since the structures of preconditions, postconditions, and the selection of devices and configuration parameters are discrete, we sample them directly from the Dirichlet Process. As permissible values of parameters can be discrete, *e.g.*, an array of specific values, common in acidity preparation; continuous, *e.g.*, an interval with minimum and maximum values, common in temperature setting; or mixed, *e.g.*, an array of specific values with random perturbations around the mean, common in timing control, we conduct the sampling by integrating Gaussian Process with Dirichlet Process $\varphi^{\text{exec-v}} \mid \varphi, \phi, \varphi^{\text{exec}} \sim DP(\alpha, H(\varphi^{\text{exec}}), \phi, \varphi) \times GP(m, K)$, where $\alpha$, $H$, $m$, and $K$ are corresponding hyperparameters.

**Unification of the interface** While clustering similar interface instances encapsulates operations, there may remain redundant interfaces due to minor discrepancies. These discrepancies often arise from differences in parameter values or naming conventions that do not fundamentally alter the operation's functionality. To alleviate such redundancies, we implement a unification process for the interfaces. Specifically, interface instances associated with the same operation are considered equivalent if they have the same number of slots and emits and share the same keys in their execution configuration parameters. By abstracting away differences in parameter values and names, we unify these interfaces into a single, generalized interface, akin to the algorithm proposed by Martelli & Montanari (1982). Unification enhances the generality of the operation-centric view by consolidating functionally-identical interfaces, maintaining a concise and representative set of operations.

**Results** Function abstraction converges on the domains respectively, as shown by the likelihood curve yielded by non-parametric model in Fig. 2B. In the DSL of Genetics, there are 304 operations in total, with an average of 7.9 interface instances per operation; for Medical, these two quantities are 269 and 6.9; for Bioengineering, they are 196 and 7.8; and for Ecology, they are 100 and 3.5. We find that a majority of operations with high occurrence frequency are unique to one domain, such as `Pipette` to Medical and `Lyse` to Genetics (see Fig. 2D). There are also common operations across domains, such as `Concentrate` and `Culture`. Take `Concentrate` for an example, its interface captures the instances with different devices according to input phases, *e.g.*, use `Bench-top_centrifuge` for `Liquid` while `Isotope_separation_centrifuge` for `Gas`, and also instances with different emits, *e.g.*, selecting `Supernatant` or `Suspension` as the product to keep.

## 3.3 Automatic Model Abstraction

The key challenge of encapsulating the product-flow-centric view is to select proper descriptive properties of a flow unit component. There exists false positive cases, where properties are attributed to components with the same semantic identifier but in different phases, *e.g.*, we consider ethanol with the property volume when it comes in liquid and with the property pressure when it comes in gas. There also exists false negative cases, where exact same components are regarded as different ones due to different reference names, *e.g.*, Acetylsalicylic Acid, ASA, and Aspirin refer to the same thing. To alleviate false positive and false negative results, we discard the design choice of the interface in the operation-centric view, which tends to cover the possibly richest context, and thereby have the non-parametric model:

$$
\begin{aligned}
&p(\omega, \omega^{\text{pred}}, \omega^{\text{succ}}, \omega^{\text{prop}}, \omega^{\text{prop-v}}) \\
&= p(\omega^{\text{prop-v}} \mid \omega^{\text{prop}}, \omega) p(\omega^{\text{prop}} \mid \omega) p(\omega^{\text{pred}} \mid \omega) p(\omega^{\text{succ}} \mid \omega) p(\omega),
\end{aligned}
\tag{2}
$$

where $\omega^{\text{prop-v}}$ denotes the values of property parameters.

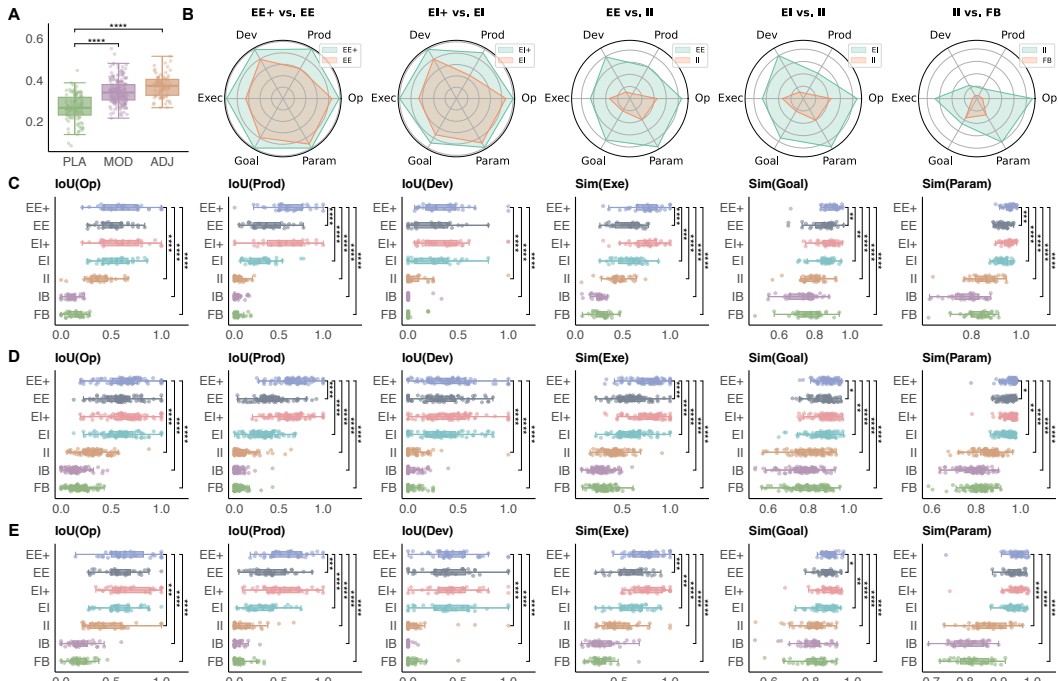

Figure 3: **Results of protocol design.** **(A)** Profile of text-level similarity between testing sets of the three tasks. **(B)** Pairwise comparison between the capabilities of different machine designers across the six dimensions. **(C-E)** Performances of the seven machine designers on the planning (C), modification (D), and adjustment (E) tasks across the six dimensions (index by column).

**Results** Model abstraction converges on the domains respectively, as shown by the likelihood curve in Fig. 2C. In the DSL of Genetics, there are $17,190$ model states, *i.e.*, product flow unit as product status, in total; the quantity is $12,472$ for Medical; $11,418$ for Bioengineering; and $2,205$ for Ecology. We find that most components of product flow units with high occurrence frequency are unique to one domain, such as RNA to Genetics and HCC to Medical (see Fig. 2E/F). Take Ethanol for example, the model captures its possible concentrations in liquid rather than in gas.

## 4 EXPERIMENTS AND DISCUSSION

In this section, we report and discuss the results of our experiments. We start from describing our realistic novel protocol design tasks (Sec. 4.1), along with the metrics to measure the consistency between the designed protocol and the groundtruth protocol (Sec. 4.2). Afterwards, we introduce the alternative representations and machine designers used for comparison (Sec. 4.3). Finally, we report and analyze the experimental results both quantitatively and qualitatively (Sec. 4.4).

### 4.1 PROTOCOL DESIGN TASKS

Generating unverified experimental objectives and their corresponding protocols specially for our protocol design tasks is impractical because those experiments which have not been peer-reviewed and published can be problematic regarding the contents themselves. To maintain both reality and scale of the testing set, for each domain we **filter out** a small subset of protocols which significantly differ from the remaining major part of the protocol set and **exclude** this subset from the corpora for automatic representation generation (Appx. E.1). This selected subset form the groundtruth of the testing set.

Table 1: **Statistics of the testing set.** Each cell presents the total number of protocols $m$ and experimental steps $n$ in the form $m\ (n)$.

|  | Genetics | Medical | Bioengineering | Ecology |
|---|---|---|---|---|
| Planning | 10 (130) | 7 (96) | 12 (157) | 2 (25) |
| Modification | 37 (442) | 15 (225) | 16 (210) | 6 (59) |
| Adjustment | 23 (219) | 5 (87) | 2 (26) | 5 (81) |

We exploit quantitative indicators to assist testing set selection, which follows the convention of measuring a protocol's *novelty* in experimental sciences (Schwab & Held, 2020). We comprehensively consider three indicators: (i) similarity between the text embedding of the NL-based description of purpose of protocols, employing the evaluation model in O'Donoghue et al. (2023); (ii) Intersection over Union (IoU) between the instance actions of protocols; (iii) similarity between the execution sequence of protocols, implemented through the Sequence Alignment (SA) algorithm (Smith et al., 1981). To note, indicators (ii) and (iii) are calculated upon the protocols pre-processed by the workflow described in Appx. D.3. Indicator (i) captures the high-level idea of protocol design, indicator (ii) is correlated to the implementation of the protocol design, while indicator (iii) captures the low-level information of protocol execution.

In response to the three purposes of protocol design introduced in Sec. 1, we specify the planning, modification, and adjustment tasks of protocol design. Candidate planning tasks, which are the confirmation of unverified experimental goals, come with relatively low scores (within the $20\%$ lowest) on indicators (i) and (ii). Candidate modification tasks come with fair scores (around the $40\%$ lowest) on indicators (i) and (ii) and relative low score on indicator (iii). Candidate adjustment tasks come with relatively high scores (within the $40\%$ highest) on all of the three indicators.

We obtain the final testing set through a human-machine collaborative workflow. We first detect the outliers of the original protocol corpus of each domain under the metrics above, thereby forming a candidate set. Afterwards, experts of the corresponding domain (holding at least a Master's degree majoring in that domain) manually check the applicability of protocols in the candidate set with cross-validation, discarding the misclassified ones, requesting for more candidate protocols, and refining the groundtruth file when necessary. The testing set includes 140 new protocols and 1757 steps in total, across the domains of Genetics, Medical, Bioengineering, and Ecology, with $23\%$ for planning, $52\%$ for modification, and $25\%$ for adjustment (see Tab. 1 and Fig. 3A for details).

## 4.2 INTER-PROTOCOL CONSISTENCY METRICS

Evaluating the consistency between a designed protocol and the groundtruth is not like comparing between two plain strings (O'Donoghue et al., 2023). Based on the corresponding commonground in experimental sciences (Bartley et al., 2023), we design six-dimensional metrics to comprehensively cover all of the major factors without biased weighting and composition. The six dimensions include: (i) **IoU on operations**, $\text{IoU(Op)} = \text{IoU}(\{\varphi_{1...|\Phi|}\}, \{\varphi'_{1...|\Phi'|}\})$, IoU between instance actions of the designed protocol $\Phi$ and the groundtruth $\Phi'$; (ii) **IoU on reagents and intermediate products**, $\text{IoU(Prod)} = \text{IoU}(\{\omega_{0...|\Phi|}\}, \{\omega'_{0...|\Phi'|}\})$; (iii) **IoU on devices**, $\text{IoU(Dev)} = \text{IoU}(\{\varphi(\text{Dev})_{1...|\Phi|}\}, \{\varphi(\text{Dev})'_{1...|\Phi'|}\})$, where $\varphi(\text{Dev})_t$ denotes the exact device for conducting the instance action $\varphi_t$; (iv) **Similarity between the execution sequences**, $\text{Sim(Exec)} = \text{SeqAlign}(\langle\varphi_{0...|\Phi|}\rangle, \langle\varphi'_{0...|\Phi'|}\rangle)$, where $\text{SeqAlign}(\cdot, \cdot)$ denotes the ordered sequence similarity score calculation by the SA algorithm; (v) **Similarity between experimental objectives**, $\text{Sim(Goal)} = \text{Cos}(S(\rho), S(\rho'))$, where $S(\cdot)$ represents the serialization operation on structural representations of protocols; (vi) **Similarity between complete protocols at parameter-wise level**, $\text{Sim(Param)} = \text{Cos}(S(\Phi), S(\Phi'))$. These six dimensions capture protocol information from low to high granularities, and also measure the consistency of both ingredient knowledge and procedural knowledge, offering a relatively objective evaluation standard.

## 4.3 MACHINE DESIGNERS

We implement an array of designers by combining different representations with different LLM-based automatic designers under tractable computing load (see Appx. D.7). We investigate four types of representations, including the original NL-based protocol representation (`Flatten`) and the three levels of encapsulation described in Sec. 2, *i.e.*, instance actions with attributes (`Instance`), operation-centric view only (`Encapsulated`), and the dual representation with operation- and product-flow-centric views (`Encapsulated+`). We consider three types of LLM-based protocol designers: (i) `Baseline`, a pure LLM-based approach with Retrieval-Augmented Generation (RAG) on the corresponding corpora (Appx. D.4); (ii) `Internal`, which takes the specific representation as part of the prompt of an LLM, requesting it to output the protocol under the constraint of the given representation (Appx. D.5); (iii) `External`, where the representation serves as an external constraint layer for the output of an LLM, verifying and refining the designed protocols (Appx. D.6). Notably, the external verifier is part of the resulting DSL as our proposed representation for protocol design.

The combination of representation and designer does not span a Cartesian space due to the intrinsic limitations of `Flatten` and `Baseline`. Therefore, we implement seven machine designers, including: (i) `Flatten-Baseline(FB)`, LLM with RAG on original protocol corpora; (ii) `Instance-Baseline(IB)`, LLM retrieval on the protocol corpora translated into instance actions; (iii) `Instance-Internal(II)`, prompting LLM with the Instruction Set Assembly (ISA) of instance actions, following the implementation of the currently state-of-the-art method O'Donoghue et al. (2023); (iv) `Encapsulated-Internal(EI)`, prompting LLM with the DSL with operation-centric view; (v) `Encapsulated-External(EE)`, LLM equipping with the external verifier provided by the DSL with operation-centric view; (vi) `Encapsulated-Internal+(EI+)`, prompting LLM with the DSL with the dual representation; and (vii) `Encapsulated-External+(EE+)`, LLM equipping with the external verifier provided by the DSL with the dual representation.

## 4.4 PROTOCOL DESIGN RESULTS

The complete quantitative results across the four domains, the three tasks, and the six dimensions of evaluation metrics are presented at Appx. B. Through paired samples t-test, we find that EE+ and EI+ significantly outperform other alternative approaches (EE+ outperforms EE: $t(278) = 8.007, \mu_d < 0, p < .0001$; EI+ outperforms EI: $t(278) = 8.397, \mu_d < 0, p < .0001$; EE+ outperforms II: $t(278) = 24.493, \mu_d < 0, p < .0001$; EI+ outperforms II: $t(278) = 23.855, \mu_d < 0, p < .0001$; see Fig. 3C-E). These comparisons demonstrate the suitability of our desired representation for protocol design. Similarly, we find that approaches equipping with a relatively higher-level representation significantly outperforms their counterparts with a relatively lower-level representation (EE outperforms II: $t(278) = 16.315, \mu_d < 0, p < .0001$; EI outperforms II: $t(278) = 15.259, \mu_d < 0, p < .0001$; II outperforms FB: $t(278) = 8.340, \mu_d < 0, p < .0001$; see Fig. 3B).

## 4.5 DISCUSSION

This work proposes a hierarchically encapsulated representation for the conceptual knowledge in experimental sciences, including instance actions with attributes, sequential representation of operations with function abstraction, and continuous representation of product-flows with model abstraction, to fully elicit LLMs' capability on protocol design as an auxiliary module. The following discussions on results reveal the design rationality, scalability, and generality of the representation.

**Contributions of the building blocks** The encapsulated representation approaches with dual views outperform their counterparts without dual views by enhancing both intra-step and inter-step details. At the intra-step level, `EI` and `EE` offer richer semantic information than `IB` and `II`, leveraging protocol-centric view to capture detailed configuration each operation. This feature accounts for their satisfactory performance on `IoU(Op)`. At the inter-step level, `EI+` and `EE+` treat each step as a `FlowUnit`, incorporating both preceding and succeeding step contexts, leading to notable improvements in `Sim(Exec)` and `IoU(Prod)`. This creates a *double assurance* mechanism (Shi et al., 2024c): the first assurance comes from internal input/output checks within each instruction, and the second from the input/output characteristics inferred from neighboring instructions. Namely, we estimate the output of the preceding operation and check its alignment with the current step's input. This design enhances step linkage, verification, and overall coherence, ensuring higher consistency and robustness in complex protocol workflows. Please refer to Appx. I.1 for the case study.

**Handling different task complexities** The overall performance aligns with the trend in complexity across the three tasks (Fig. 3A); however, the dual-view encapsulated representations, `EI+` and `EE+`, demonstrate superior performance compared to their counterparts. In planning, these methods consider all necessary components, enabling creative yet structured protocol generation. For modification tasks, they provide feedback on parameter changes, detecting inconsistencies that their counterparts might fail to capture. In adjustment tasks, `EE+`'s external verifier maintains protocol integrity by identifying component relationships. Please refer to Appx. I.2 for the case study.

**Generality across domains** Our DSL-based approaches offer a unified, modular representation with generalizability across scientific domains (see domain-indexed results at Appx. B.2). The dual-view approach abstracts experimental processes into operations and flow units, capturing essential details while remaining applicable across fields. By representing dependencies between steps and tracking product flow, the replication of experiments could be enhanced. The framework captures cross-domain commonalities while allowing domain-specific content like specialized operations and reagents. This unified representation standardizes protocols and enables researchers to adopt experimental protocols from multiple fields, fostering interdisciplinary collaboration and innovation. Please refer to Appx. I.3 for the case study. Limitations on generality are discussed at Appx. G.

ACKNOWLEDGEMENTS

This work was partially supported by the National Natural Science Foundation of China under Grants 52475001. Q. Xu is a visiting student at Peking University from University of Science and Technology of China. The authors would like to thank Haofei Hou for his earlier works regarding domain-specific representations, and also Jiawen Liu for her assistance in figure drawings.

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

# A  ADDITIONAL REMARKS

## A.1  RATIONALE OF THE OVERALL DESIGN CHOICE

It seems that we can formulate the protocol design problem in the fashion of Markov Decision Process (MDP) and solve it by heuristic-based planning methods or Hierarchical Reinforcement Learning (HRL) approaches. However, although the formulation itself is feasible, solving the problem may not be practical. Consider solving the problem through an HRL approach designed for heterogeneous action space with parameters (as the protocol is required to decide both the key properties of an operation and the corresponding values). This hierarchical agent may be trained to converge on a fine-grained environment with a clearly designed reward function, or on a large dataset with trajectories for offline learning. Unfortunately, we have access to neither an interactive environment simulating the experiments nor sufficient data to support offline training (Pateria et al., 2021).

Treating the experimental procedures as a white box and creating digital twins for experiments can be an elegant solution and thereby facilitate various applications other than protocol design. This effort requires elaborated design of simulation granularity, exhaustive collection of primitive principles of the system, efficient implementation of rule production, and define precise metrics for evaluating the distance between current and objective states (serving as a reward function), which can be labor-intensive and is far out of the scope of this work. On the other hand, viewing those published protocols as trajectories for offline training, the scale of the offline dataset and the density of the reward function are much too insufficient to support training to convergence. Augmenting the data, synthesizing realistic trajectories, or enhancing the accessibility of protocols, are out of the scope of this work. Given the current obstacles, we choose not to formulate the problem in an MDP fashion. Though an MDP-style formulation can be more precise and elegant, it may misguide the readers to some extent. Instead, we decide to leverage the rich domain-specific knowledge provided by knowledge-based agents such as LLMs, where knowledge may complement the lack of data and dense reward function. This design choice is also in line with the initial attempts on automatic experiment design (Boiko et al., 2023; M. Bran et al., 2024).

In summary, our design choice of formulation is a compromise based on currently limited resources and restricted scope. Nonetheless, the exploration of more precise and elegant formulations represents a promising avenue for future research.

## A.2  INTUITION BEHIND THE INTERFACE

Interface is a concept of functional abstraction (Abelson & Sussman, 1996). Interface disentangles the abstract functionality on the semantics level and its corresponding implementation details on the execution level. This approach encapsulates the implementation of an operation into a *black-box*, so the users of the operation would only need to consider its input and output. Therefore, with such encapsulated representation for protocol design, we only need to care about the consistency between the output of the predecessor operation and the input of the successor operation, without caring about their implementation details.

This is the idea behind operationalization. Operationalization makes the interface an abstract function over all relative instance actions. The interface is abstracted from the execution contexts of all instance actions with the same reference name, *i.e.*, the same purpose, and can be instantiated to an instance action given a specific execution context. A specific context can be the predecessor operation, the successor operation, the precondition, or the postcondition of the considered operation. An instance action configures a specific implementation for a specific execution context. For the operation *"Homogenization"*, the implementation of one instance action can be *"using an ultrasonic homogenizer"* if the precondition, namely, the execution context, has intermediate product *"cell suspension"* available; the implementation of another instance action can be *"using a bead mill"* if the precondition contains tissue. This example demonstrates the relationship between interface and instance actions of an operation: the interface is abstracted from the set of instance actions and can be instantiated to instance actions.

Here we also give a more intuitive example to enhance the reader's comprehension. Consider the culinary scenario with the actions *"frying the egg"*, *"frying the fish"*, and *"frying the steak"*.These are different instance actions coming with the same purpose *"to fry something"*. Therefore, we can abstract the interface from these instance actions to operationalize the operation *"fry"*. The input of *"fry"* should be something raw and its output should be something fried. Given different preconditions with available eggs or pieces of steak, the abstract semantic operation *"fry"* can be grounded

to instance actions *"frying the egg"* or *"frying the steak"* respectively, through the instantiation of the interface. In summary, an interface serves as the bridge between the semantics level and the execution level.

## A.3 VALUES OF MANUAL PROTOCOL CERTIFICATION

Certification is always one of the central focuses in the engineering practices of automation. In our practice, we only automate the process of protocol design, which is the primary objective of this work, and keep the manual certification part. On one hand, relieving experimental scientists from the labour-intensive protocol design tasks, thereby allowing them more time for high-level thinking, is a sufficiently significant improvement so far. On the other hand, engineering practices such as lab automation and manufacturing are in high demand for preciseness. This leads to the requirement of manual certification. Domain experts handle subtle cases through their tacit domain-specific knowledge and are responsible for their decisions (Wang et al., 2023b). According to these considerations and the standard operating processes of experimental sciences, we choose to certify the designed protocols by domain experts.

Our current choice is a compromise on the limitation of techniques and the demand for preciseness. In future work, we can conduct investigations on how to build digital twins of self-driving laboratories. Such digital twins support prediction, explanation, and counterfactual analysis of unseen behaviors of the experiments, which may facilitate machine-based protocol certification. Grounding these blue-sky thoughts necessitates addressing the challenging problems regarding the decision of simulation granularity, the implementation of data-efficient simulation model construction, and the injection of tacit domain-specific knowledge. In summary, the exploration of generated-protocol-certification by machines represents a promising avenue for future research.

## A.4 LIMITATIONS OF AUTOMATIC PROTOCOL CERTIFICATION

LLMs can be much too uncontrollable for engineering practices such as lab automation, which may lead to unpredictable dangerous situations (Wang et al., 2023b). There comes a dilemma—we try to exploit the capability of reasoning over knowledge of LLMs, while we try to alleviate the drawbacks brought up by the uncontrollable nature of LLMs. Our proposed representation is dedicated to resolving the dilemma. The representations not only elicit LLMs' potential on protocol design through structural knowledge representation, but also serve as a guardrail for LLMs. Since the generated protocols are represented as corresponding DSL programs, the permissible output space is much more confined compared with that of pure LLMs, serving as constraints upon the LLM-generated protocols. Thanks to the verification mechanisms provided by DSLs, the correctness of the generated protocols can be checked to some extent. Therefore, by equipping LLMs with an auxiliary constraint layer, we may approach a balance between knowledge utilization and preciseness.

However, the current verification on the level of DSL programs is far from sufficient for serving as a certification. Certification is a serious process, where any possibilities of reporting false positive cases are required to be eliminated. Some cases can be highly long-tailed distributed, which may not be detected by data-driven and knowledge-driven machine certifiers. In this context, human domain experts are responsible for coming up with these potential risks through their experiences and tacit knowledge. Therefore, we are not likely to move human experts out of the loop, except that we can efficiently build up appropriate digital twins for self-driving laboratories. In current practices, the automation of protocol design puts human experts into a larger loop without focusing on the low-level details of experiments. As a result, they are allowed more time for high-level thoughts on things like values, which are not likely to be alternated by machines. In summary, it is neither practical nor necessary to totally move human experts out of the loop of automatic scientific discovery. The investigation of human-machine coordination in protocol certification represents a promising avenue for future research.

## A.5 RATIONALE FOR THE REAGENT CONSUMPTION MODEL

We treat the instantiation and the consumption of reagents a *one-time deal* without considering the exact volume of consumption and the corresponding remainder. The rationale for such design choice comes from both the current Standard Operating Process (SOP) of experimental sciences and the properties of self-driving laboratories (Bartley et al., 2023).

In the current SOP for manually conducted experiments, experimenters are required to use prefabricated sets of reagents. Similarly, experimenters use specific containers with predefined capacities to transfer intermediate products. Therefore, one pack of reagents or one container of intermediate products is only used once for an operation, without considering the remainder. This results in a more succinct representation where reagents are regarded as discrete elements rather than continuous volumes.

For self-driving laboratories, this is deliberately designed for efficient variable management following the corresponding principles in computer system design (Abelson & Sussman, 1996). In computer systems, not removing used variables would cause out-of-memory errors, let alone in physical automation systems, where the physical memory slots are much harder than the virtual memory slots in computer systems to manage. Hence, we exploit this variable management mechanism to enhance the execution efficiency of self-driving laboratories.

### A.6 RELATION TO LLM REASONING

We would like to clarify that our objective is not to alternate Chain-of-Thought (CoT) reasoning. According to recent studies on the properties of CoT, LLMs with CoT may generate coherent but unprofessional text in expertise-intensive application scenarios (Xiao et al., 2023). Therefore, our proposed representation serves as an auxiliary guardrail module for LLMs with reasoning techniques such as CoT, enhancing LLMs' reasoning capability from two aspects: (i) the representation constrain the scope of reasoning into a close set of entities, such as available operations, reagents, and devices commonly used in the domain; and (ii) the representation provides fine-grained injection of domain-specific knowledge for LLMs, resulting in not only coherent but also expertise-compatible generated content.

### A.7 APPLICABILITY TO DOMAINS BEYOND SCIENTIFIC EXPERIMENT

In theory, our framework can be applied to any field that requires adherence to specific protocols and has a need for automated execution. Let us consider an automated kitchen controlled by a computer as an example.

Assuming the automated kitchen's computer is already programmed to prepare "braised pork ribs" and "steamed sea bass":

```
1  Braised Pork Ribs:
2
3  1.Select pork ribs as the main ingredient.
4  2.Heat a pan over high heat.
5  3.Add the ribs to the pan and fry for about 5 minutes until they are
       browned.
6  4.Add seasonings: soy sauce and sugar.
7  5.Reduce the heat to medium.
8  6.Simmer the ribs for 30 minutes until tender.
9  7.Serve hot.
10
11 START
12 SELECT ingredient: ribs
13 ACTION: fry, temperature: high, time: 5 min
14 ADD seasoning: soy sauce, sugar
15 ACTION: simmer, temperature: medium, time: 30 min
16 END
17
18 Steamed Sea Bass:
19
20 1.Select a whole sea bass as the main ingredient.
21 2.Prepare a steamer and heat it to high temperature.
```

```
22 3.Place the sea bass in the steamer.
23 4.Steam the fish for about 15 minutes until fully cooked.
24 5.Add seasonings: ginger slices and chopped scallions.
25 6.Serve immediately with the garnish.
26
27 START
28 SELECT ingredient: sea bass
29 ACTION: steam, temperature: high, time: 15 min
30 ADD seasoning: ginger, scallion
31 END
```

Next, we can derive the corresponding DSL. For instance:

```
 1 {
 2   "cooking_methods": {
 3     "braise": {
 4       "steps": [
 5         {"type": "fry", "temperature": "high", "time": "5 min"},
 6         {"type": "simmer", "temperature": "medium", "time": "30 min"}
 7       ],
 8       "seasoning": ["soy sauce", "sugar"]
 9     },
10     "steam": {
11       "steps": [
12         {"type": "steam", "temperature": "high", "time": "15 min"}
13       ],
14       "seasoning": ["ginger", "scallion"]
15     }
16   },
17   "ingredients": {
18     "ribs": {
19       "category": "meat",
20       "default_braise_time": "30 min"
21     },
22     "sea_bass": {
23       "category": "fish",
24       "default_braise_time": "20 min",
25       "default_steam_time": "15 min"
26     }
27   }
28 }
```

Now, let us create a new recipe for Braised Sea Bass by combining the braising technique with sea bass as the main ingredient.

```
1 START
2 SELECT ingredient: sea bass
3 ACTION: fry, temperature: high, time: 5 min
4 ADD seasoning: soy sauce, sugar
5 ACTION: simmer, temperature: medium, time: 20 min
6 END
```

# B  COMPLETE RESULTS

## B.1  TASK-INDEXED COMPLETE RESULTS

Table A1: **Complete quantitative results on protocol design, specifically the planning task.** Each cell represents the machine designer's average score (at the top of the cell) on all testing samples across the four domains and the corresponding standard error of mean (at the bottom of the cell). For each dimension, we highlight the results of both **the best** and **the second best** ones.

|      | IoU(Op) | IoU(Prod) | IoU(Dev) | Sim(Exec) | Sim(Goal) | Sim(Param) |
|------|---------|-----------|----------|-----------|-----------|------------|
| FB   | 0.143   | 0.040     | 0.020    | 0.282     | 0.766     | 0.826      |
|      | (0.087) | (0.055)   | (0.060)  | (0.090)   | (0.096)   | (0.059)    |
| IB   | 0.109   | 0.036     | 0.019    | 0.242     | 0.735     | 0.781      |
|      | (0.067) | (0.053)   | (0.074)  | (0.065)   | (0.090)   | (0.069)    |
| II   | 0.382   | 0.050     | 0.084    | 0.452     | 0.788     | 0.851      |
|      | (0.154) | (0.062)   | (0.191)  | (0.134)   | (0.074)   | (0.062)    |
| EI   | 0.542   | 0.305     | 0.259    | 0.572     | 0.849     | 0.926      |
|      | (0.160) | (0.181)   | (0.211)  | (0.152)   | (0.066)   | (0.026)    |
| EI+  | **0.603** | **0.555** | **0.357** | **0.737** | **0.875** | **0.949**  |
|      | (0.208) | (0.260)   | (0.237)  | (0.172)   | (0.057)   | (0.023)    |
| EE   | 0.524   | 0.370     | 0.252    | 0.558     | 0.846     | 0.928      |
|      | (0.151) | (0.198)   | (0.206)  | (0.148)   | (0.078)   | (0.025)    |
| EE+  | **0.607** | **0.605** | **0.355** | **0.744** | **0.893** | **0.951**  |
|      | (0.211) | (0.235)   | (0.242)  | (0.179)   | (0.056)   | (0.021)    |

Table A2: **Complete quantitative results on protocol design, specifically the modification task.** Each cell represents the machine designer's average score (at the top of the cell) on all testing samples across the four domains and the corresponding standard error of mean (at the bottom of the cell). For each dimension, we highlight the results of both **the best** and **the second best** ones.

|      | IoU(Op) | IoU(Prod) | IoU(Dev) | Sim(Exec) | Sim(Goal) | Sim(Param) |
|------|---------|-----------|----------|-----------|-----------|------------|
| FB   | 0.181   | 0.050     | 0.038    | 0.304     | 0.796     | 0.809      |
|      | (0.102) | (0.071)   | (0.071)  | (0.102)   | (0.090)   | (0.060)    |
| IB   | 0.150   | 0.038     | 0.039    | 0.281     | 0.772     | 0.788      |
|      | (0.100) | (0.065)   | (0.076)  | (0.100)   | (0.089)   | (0.060)    |
| II   | 0.331   | 0.101     | 0.061    | 0.416     | 0.802     | 0.851      |
|      | (0.143) | (0.131)   | (0.135)  | (0.127)   | (0.087)   | (0.059)    |
| EI   | 0.593   | 0.318     | 0.336    | 0.602     | 0.866     | 0.937      |
|      | (0.186) | (0.158)   | (0.235)  | (0.164)   | (0.066)   | (0.030)    |
| EI+  | **0.648** | **0.626** | **0.413** | **0.765** | **0.883** | **0.952**  |
|      | (0.210) | (0.188)   | (0.256)  | (0.170)   | (0.055)   | (0.031)    |
| EE   | 0.588   | 0.403     | 0.332    | 0.601     | 0.873     | 0.940      |
|      | (0.185) | (0.192)   | (0.228)  | (0.164)   | (0.053)   | (0.028)    |
| EE+  | **0.640** | **0.661** | **0.410** | **0.757** | **0.893** | **0.953**  |
|      | (0.213) | (0.179)   | (0.253)  | (0.170)   | (0.043)   | (0.032)    |

Table A3: **Complete quantitative results on protocol design, specifically the adjustment task.** Each cell represents the machine designer's average score (at the top of the cell) on all testing samples across the four domains and the corresponding standard error of mean (at the bottom of the cell). For each dimension, we highlight the results of both **the best** and **the second best** ones.

|  | IoU(Op) | IoU(Prod) | IoU(Dev) | Sim(Exec) | Sim(Goal) | Sim(Param) |
|---|---|---|---|---|---|---|
| FB | 0.192 (0.100) | 0.077 (0.104) | 0.051 (0.094) | 0.319 (0.103) | 0.811 (0.078) | 0.823 (0.051) |
| IB | 0.197 (0.131) | 0.039 (0.063) | 0.006 (0.021) | 0.337 (0.141) | 0.802 (0.082) | 0.810 (0.049) |
| II | 0.453 (0.208) | 0.115 (0.161) | 0.091 (0.211) | 0.508 (0.184) | 0.805 (0.081) | 0.873 (0.056) |
| EI | 0.587 (0.190) | 0.328 (0.186) | 0.400 (0.265) | 0.623 (0.165) | 0.863 (0.055) | 0.944 (0.027) |
| EI+ | **0.668** (0.208) | **0.545** (0.259) | **0.449** (0.247) | **0.775** (0.152) | **0.883** (0.056) | **0.950** (0.040) |
| EE | 0.581 (0.184) | 0.404 (0.205) | 0.395 (0.261) | 0.616 (0.162) | 0.875 (0.039) | 0.946 (0.026) |
| EE+ | **0.650** (0.220) | **0.589** (0.229) | **0.441** (0.248) | **0.758** (0.160) | **0.893** (0.033) | **0.950** (0.042) |

## B.2 DOMAIN-INDEXED COMPLETE RESULTS

Table A4: **Complete quantitative results on protocol design, specifically the Genetics domain.** Each cell represents the machine designer's average score (at the top of the cell) on all testing samples across the three tasks and the corresponding standard error of mean (at the bottom of the cell). For each dimension, we highlight the results of both **the best** and **the second best** ones.

|  | IoU(Op) | IoU(Prod) | IoU(Dev) | Sim(Exec) | Sim(Goal) | Sim(Param) |
|---|---|---|---|---|---|---|
| FB | 0.179 (0.113) | 0.065 (0.082) | 0.037 (0.080) | 0.301 (0.116) | 0.795 (0.091) | 0.805 (0.066) |
| IB | 0.157 (0.129) | 0.042 (0.060) | 0.022 (0.059) | 0.297 (0.137) | 0.793 (0.070) | 0.789 (0.062) |
| II | 0.379 (0.200) | 0.120 (0.158) | 0.079 (0.160) | 0.457 (0.180) | 0.807 (0.083) | 0.850 (0.072) |
| EI | 0.599 (0.189) | 0.332 (0.177) | 0.353 (0.243) | 0.619 (0.164) | 0.862 (0.055) | 0.941 (0.026) |
| EI+ | **0.691** (0.198) | **0.606** (0.252) | **0.429** (0.283) | **0.803** (0.151) | **0.882** (0.054) | **0.954** (0.033) |
| EE | 0.592 (0.189) | 0.415 (0.206) | 0.351 (0.241) | 0.615 (0.163) | 0.870 (0.052) | 0.943 (0.025) |
| EE+ | **0.677** (0.210) | **0.653** (0.228) | **0.425** (0.280) | **0.791** (0.161) | **0.888** (0.045) | **0.955** (0.034) |

Table A5: **Complete quantitative results on protocol design, specifically the Medical domain.**
Each cell represents the machine designer's average score (at the top of the cell) on all testing
samples across the three tasks and the corresponding standard error of mean (at the bottom of the
cell). For each dimension, we highlight the results of both **the best** and **the second best** ones.

|      | IoU(Op) | IoU(Prod) | IoU(Dev) | Sim(Exec) | Sim(Goal) | Sim(Param) |
|------|---------|-----------|----------|-----------|-----------|------------|
| FB   | 0.174   | 0.048     | 0.030    | 0.312     | 0.796     | 0.839      |
|      | (0.085) | (0.070)   | (0.067)  | (0.075)   | (0.087)   | (0.043)    |
| IB   | 0.139   | 0.029     | 0.023    | 0.264     | 0.721     | 0.795      |
|      | (0.054) | (0.038)   | (0.063)  | (0.045)   | (0.123)   | (0.070)    |
| II   | 0.373   | 0.081     | 0.091    | 0.424     | 0.776     | 0.871      |
|      | (0.093) | (0.087)   | (0.205)  | (0.072)   | (0.097)   | (0.041)    |
| EI   | 0.604   | 0.322     | 0.309    | 0.594     | 0.861     | 0.932      |
|      | (0.167) | (0.146)   | (0.253)  | (0.148)   | (0.079)   | (0.031)    |
| EI+  | **0.615** | **0.574** | **0.400** | **0.758** | 0.871   | **0.952**  |
|      | (0.196) | (0.242)   | (0.198)  | (0.149)   | (0.060)   | (0.021)    |
| EE   | 0.591   | 0.373     | 0.298    | 0.583     | **0.873** | 0.936      |
|      | (0.158) | (0.166)   | (0.234)  | (0.149)   | (0.054)   | (0.030)    |
| EE+  | **0.615** | **0.613** | **0.390** | **0.756** | **0.891** | **0.955**  |
|      | (0.197) | (0.210)   | (0.202)  | (0.151)   | (0.040)   | (0.019)    |

Table A6: **Complete quantitative results on protocol design, specifically the Ecology domain.**
Each cell represents the machine designer's average score (at the top of the cell) on all testing
samples across the three tasks and the corresponding standard error of mean (at the bottom of the
cell). For each dimension, we highlight the results of both **the best** and **the second best** ones.

|      | IoU(Op) | IoU(Prod) | IoU(Dev) | Sim(Exec) | Sim(Goal) | Sim(Param) |
|------|---------|-----------|----------|-----------|-----------|------------|
| FB   | 0.155   | 0.030     | 0.021    | 0.297     | 0.781     | 0.807      |
|      | (0.085) | (0.035)   | (0.048)  | (0.088)   | (0.096)   | (0.056)    |
| IB   | 0.162   | 0.006     | 0.030    | 0.275     | 0.763     | 0.788      |
|      | (0.118) | (0.015)   | (0.058)  | (0.105)   | (0.090)   | (0.063)    |
| II   | 0.386   | 0.043     | 0.027    | 0.448     | 0.788     | 0.856      |
|      | (0.176) | (0.062)   | (0.062)  | (0.131)   | (0.065)   | (0.044)    |
| EI   | **0.458** | 0.259   | **0.351** | 0.514     | 0.879     | 0.933      |
|      | (0.171) | (0.134)   | (0.195)  | (0.142)   | (0.048)   | (0.028)    |
| EI+  | 0.411   | **0.569** | **0.359** | **0.586** | **0.888** | **0.945**  |
|      | (0.134) | (0.133)   | (0.175)  | (0.127)   | (0.052)   | (0.023)    |
| EE   | **0.458** | 0.347   | **0.351** | 0.507     | 0.874     | 0.934      |
|      | (0.171) | (0.151)   | (0.195)  | (0.138)   | (0.048)   | (0.029)    |
| EE+  | 0.414   | **0.581** | 0.346    | **0.586** | **0.910** | **0.944**  |
|      | (0.142) | (0.141)   | (0.177)  | (0.131)   | (0.035)   | (0.024)    |

Table A7: **Complete quantitative results on protocol design, specifically the Bioengineering domain.** Each cell represents the machine designer's average score (at the top of the cell) on all testing samples across the three tasks and the corresponding standard error of mean (at the bottom of the cell). For each dimension, we highlight the results of both **the best** and **the second best** ones.

|      | IoU(Op) | IoU(Prod) | IoU(Dev) | Sim(Exec) | Sim(Goal) | Sim(Param) |
|------|---------|-----------|----------|-----------|-----------|------------|
| FB   | 0.176   | 0.048     | 0.050    | 0.300     | 0.790     | 0.826      |
|      | (0.085) | (0.089)   | (0.081)  | (0.084)   | (0.090)   | (0.042)    |
| IB   | 0.149   | 0.050     | 0.038    | 0.286     | 0.767     | 0.797      |
|      | (0.077) | (0.087)   | (0.091)  | (0.078)   | (0.083)   | (0.046)    |
| II   | 0.352   | 0.062     | 0.066    | 0.443     | 0.810     | 0.860      |
|      | (0.151) | (0.090)   | (0.187)  | (0.125)   | (0.073)   | (0.045)    |
| EI   | 0.565   | 0.307     | 0.310    | 0.603     | 0.851     | 0.930      |
|      | (0.164) | (0.186)   | (0.249)  | (0.169)   | (0.072)   | (0.033)    |
| EI+  | **0.657** | **0.577** | **0.394** | **0.743** | **0.888** | **0.944**  |
|      | (0.209) | (0.177)   | (0.241)  | (0.179)   | (0.056)   | (0.041)    |
| EE   | 0.558   | 0.392     | 0.303    | 0.598     | 0.855     | 0.933      |
|      | (0.162) | (0.214)   | (0.246)  | (0.165)   | (0.076)   | (0.030)    |
| EE+  | **0.653** | **0.614** | **0.401** | **0.742** | **0.900** | **0.945**  |
|      | (0.206) | (0.172)   | (0.246)  | (0.176)   | (0.046)   | (0.041)    |

## C  ETHICS STATEMENT

### C.1  HUMAN EXPERT PARTICIPANTS

The testing set selection and groundtruth checking tasks conducted by human experts in this work has been approved by the Institutional Review Board (IRB) of Peking University. We have been committed to upholding the highest ethical standards in conducting this study and ensuring the protection of the rights and welfare of all participants. We paid the domain experts a wage of $22.5/h for their work in this study.

We have obtained informed consent from all human experts, including clear and comprehensive information about the purpose of the study, the procedures involved, the risks and benefits, and the right to withdraw at any time without penalty. Participants were also assured of the confidentiality of their information. Any personal data collected (including name, age, and gender) was handled in accordance with applicable laws and regulations.

### C.2  CORPORA COLLECTION

We carefully ensure that all protocols included in our corpora strictly comply with open access policies under the Creative Commons license. This strategy guarantees adherence to copyright and intellectual property laws, thereby preventing any potential infringement or unauthorized use of protected materials. By exclusively employing resources that are freely accessible and legally distributable, we maintain the highest standards of ethical research conduct, promoting transparency and respect for the intellectual property rights of others. This commitment ensures that our work advances the frontiers of knowledge in a manner that is both legally sound and ethically responsible.

## D  IMPLEMENTATION DETAILS

### D.1  PRIOR MODEL OF PRODUCT FLOW-CENTRIC VIEW

```
1  <ProductFlow> ::= <Pred> <FlowUnit> <Succ>
```

```
 2
 3 <Pred> ::= <Operation.UniqueName>
 4
 5 <Succ> ::= <Operation.UniqueName>
 6
 7 <FlowUnit> ::= <Component> <ComponentType> <RefName> <Vol> <Container> *<
     Cond>
 8 <Component> ::= <STR>
 9 <ComponentType> ::= Gas | Liquid | Solid | Semi-Solid | Mixture |
     ChemicalCompound | BiologicalMaterial | Reagent | PhysicalObject |
     File/Data | ... [Known component types]
10 <RefName> ::= <Component> <Index>
11 <UnitArgType> ::= MAT | PROD
12 <Vol> ::= <REAL> <MEAS>
13 <Container> ::= Tube | Flask | Pipette | ... [Known container types]
14 <Cond> ::= <ArgKey> <ArgValue>
15 <ArgKey> ::= Temperature | Pressure | Acidity | Lighting | ... [Known
     conditional keys]
16 <ArgValue> ::= <REAL> <MEAS>
```

## D.2 PRIOR MODEL OF OPERATION-CENTRIC VIEW

```
 1 <Operation> ::= <UniqueName> *<Pattern>
 2
 3 <UniqueName> ::= <STR>
 4
 5 <Pattern> ::= <Precond> <Execution> <Postcond> *<Example>
 6
 7 <Precond> ::= <SlotArgNum> *<SlotArg>
 8 <SlotArgNum> ::= <INT>
 9 <SlotArg> ::= <ProductFlow.FlowUnit.ComponentType>
10
11 <Postcond> ::= <EmitArgNum> *<EmitArg>
12 <EmitArgNum> ::= <INT>
13 <EmitArg> ::= <ProductFlow.FlowUnit.ComponentType>
14
15 <Example> ::= <STR>
16
17 <Execution> ::= <DeviceType> <Capacity> *<Config>
18 <DeviceType> ::= Incubator | Autoclave | Centrifuge | ... [Known device
     types]
19 <Capacity> ::= <REAL> <MEAS>
20 <Config> ::= <ArgKey> <ArgValue>
21 <ArgKey> ::= Duration | Pace | Power | Quantity | ... [Known device
     configuration items]
22 <ArgValue> ::= <REAL> <MEAS>
```

### D.3 Pre-processing of the protocols

The protocol pre-processing steps begin by reading all JSON files of the protocols. Each protocol is then splitted sentence-by-sentence using Spacy[1], with the constraint that every sentence is longer than ten characters. Due to the large volume of data, sentence splitting is handled in parallel. Afterwards, deeper sentence splitting is performed based on specific conditions for further refinement, such as the presence of `"and/then/and then"` followed by a verb[2]. We then parse sentences into root verbs and purpose clauses, which are identified using `token.dep_ == "ROOT"` for root verbs and `prepositional/adverbial/modals` for purpose clauses. Lastly, we merge phrases based on punctuation, and their classification into valid sentences or decorative phrases depends on whether they contain a root verb or lack a purpose clause.

The first verb in each sentence is extracted as an opcode, again utilizing parallel processing for efficiency. Opcode frequency is filtered to exclude stopwords, which are recorded in a separate text file. Then we categorize these opcodes into high-level operation classes using a GPT model (gpt-4o mini), where each opcode is classified into categories like `Transfer Operations`, `Transformation Operations`, or `Data Operations`.

Once operation classification is complete, entity recognition is performed (also using gpt-4o mini) to identify entities like `devices`, `input_flow_units`, `output_flow_units`, and `total_time`. Each flow unit is further categorized (also using gpt-4o mini) with a high-level classification composed of a phase, *i.e.*, `Gas`, `Liquid`, `Solid`, *etc.*; and a type, *i.e.*, `Chemical Compound`, `Biological Material`, *etc*. When both phase and type are successfully labeled, phase is preferred as the feature of the flow unit. If phase labeling fails, we use type the feature of the flow unit. If neither phase nor type is successfully labeled, the corresponding feature is set to `None`. Part of the rationale is that there are non-reagent components in the general sense, *i.e.*, `data`, `files`, `obscure or undefined substances`, *etc*. Therefore, we apply this strategy to maximize the possibility that there is a meaningful upper class labeling of the components without any redundancy.

Finally, we conduct a synonym merge process on the devices, which starts by using `transformers AutoTokenizer`[3] to get an embedding for each device name. Afterwards, we use `sklearn`[4] to identify potentially similar entity pairs by calculating the cosine similarity of the candidate entities, and then passing these entity pairs to the GPT model for synonym detection, thereby merging devices belonging to the same type. The reference names of these combined devices will be one of the features.

### D.4 Pure LLM-based designer

The pure LLM-based designer employs RAG to retrieve similar protocols from the corresponding corpora for representation, following the design choice of the baseline in O'Donoghue et al. (2023). Specifically, in the FB approach, three similar protocols are first retrieved from the original protocol corpora using RAG, and then, along with the title and description of the target protocol, they are provided to the LLM to generate a NL plan. The LLM subsequently translates the NL plan into Python pseudocode. In the IB approach, three similar protocols' instance actions (like Python pseudofunctions definitions) are first retrieved from the corpora, and after randomizing their order, they are provided to the LLM along with the title and description of the target protocol to generate a plan in the form of Python pseudocode.

```
1 [Prompt for retrieving similar protocols from corpora]
2 You are an expert in biology and you are very familiar with the
      experiment protocols.
3 I would like to make a protocol for {title}.
4 I will give you some related protocols in the database.
5 Could you find me the most three similar and relevant protocols for
      reference in the given range?
```

---

[1] https://spacy.io/api/sentencizer

[2] https://spacy.io/api/matcher#_title

[3] https://huggingface.co/docs/transformers/v4.45.1/en/model_doc/auto#transformers.AutoTokenizer

[4] https://scikit-learn.org/stable/modules/generated/sklearn.metrics.pairwise.cosine_similarity.html#cosine-similarity

```
 6
 7 Please output id of your selected protocols, separating with a comma. Don
       't output any other information.
 8 [Output format]
 9 id_1,id_2,id_3
10
11 [Related protocols]
12 {context}
13
14 Answer:
```

```
 1 [Prompt for generating NL plan]
 2 Your goal is to generate steps for a biology protocol.
 3 These protocol steps must accurately describe a complete scientific
       protocol to obtain a result.
 4 Steps of some similar protocols will be provided as a reference for you
       to generate the new one.
 5 Output should only contain the steps without any other information.
 6
 7 Here is an example of how to generate steps for a biology protocol.
 8
 9 EXAMPLE:
10
11 {example protocol title}
12
13 Here are some extra details about the protocol:
14
15 {example protocol description}
16
17 example steps:
18
19 {example protocol steps}
20
21 YOUR TASK:
22 Generate steps for a protocol for {title}.
23
24 Here are some extra details about the protocol:
25
26 {details}
27
28 Here are some similar protocols' steps for reference:
29
30 {steps}
31
32 your steps:
```

```
 1 [Prompt for translating NL plan to pseudocode]
 2 Your goal is to convert biology protocols into python pseudocode.
 3
 4 EXAMPLE
```

```
 5 Here is an example of how to convert a protocol for {example protocol
      title} into python pseudocode
 6
 7 {example protocol}
 8
 9 {example python pseudocode}
10
11 YOUR TASK:
12 Here is a biology protocol entitled '{title}' The protocol steps are as
      follows:
13
14 {protocol}
15
16 Please convert this protocol into python pseudocode.
17
18 python pseudocode:
```

```
 1 [Prompt for generating plan in pseudocode]
 2 Your goal is to generate python pseudocode for biology protocols.
 3
 4 Here is an example of how to generate pseudocode for a biology protocol.
 5
 6 EXAMPLE:
 7
 8 {example protocol title}
 9
10 Here are some extra details about the protocol:
11
12 {example protocol description}
13
14 example pseudocode:
15
16 {example pseudocode}
17
18 YOUR TASK:
19 Generate pseudocode for a protocol for {title}.
20
21 Here are some extra details about the protocol:
22
23 {details}
24
25 You may only make use of the following python pseudocode functions:
26
27 {psuedofunctions}
28
29 your pseudocode:
```

## D.5 INTERNAL DESIGNER

The internal designer incorporates the specific representation as part of the prompt for an LLM, asking it to output the protocol while adhering to the given representation constraints, echoing the

idea of Wang et al. (2023a). Specifically, in II, the instance actions retrieved from the corpora via RAG and the pseudofunctions definitions of the target protocol are shuffled and then provided together to the LLM, constraining it to generate a plan in the form of Python pseudocode using the given pseudofunctions definitions. In EI and EI+, relevant DSL instructions are selected from a domain-specific operation-centric view DSL and product-flow-centric view DSL, respectively. These instructions and the target protocol's title and description are provided to the LLM, prompting it to output the corresponding plan as instantiated DSL instructions.

```
1 [Protocol for generating plan in DSL program using operation-centric view
     DSL]
2 Your goal is to generate plan in domain specific language (DSL) for
     biology protocols.
3 The DSL specifications related to the operations involved in the
     experiment are provided. The DSL specification of each operation
     consists of multiple patterns, each pattern is an operation execution
      paradigm.
4
5 Here is an example of how to generate plan in DSL for a biology protocol.
6
7 EXAMPLE:
8
9 {example protocol title}
10
11 Here are some extra details about the protocol:
12
13 {example protocol description}
14
15 example plan in DSL:
16
17 {example plan}
18
19 [Requirements]
20 1. Design the experiment with finer granularity, incorporating more steps
      to complete the experiment in a more rigorous, complex, and
     comprehensive manner.
21 2. There are some missing parameters in the DSL specification. You should
      generate each step of the DSL program as detailed as possible based
     on your understanding of the protocol plan.
22 3. In Precond and Postcond, use formal name of the component to represent
      the SlotArg and EmitArg of each step. The component name should
     clearly describe the content of the component.
23
24 YOUR TASK:
25 Generate plan in DSL for a protocol for {title}.
26
27 Here are some extra details about the protocol:
28
29 {details}
30
31 You can choose to instantiate the following DSL specification to
     construct the DSL program:
32
```

```
33 {DSL}
34
35 Your plan in DSL program:
```

```
 1 [Protocol for generating plan in DSL program using dual representation]
 2 Your goal is to generate plan in domain specific language (DSL) for
       biology protocols.
 3 Two perspectives of the DSL specification are provided: the specification
        for experimental operations and the specification for experimental
       products.
 4 The DSL specification of each operation or product consists of multiple
       patterns, each pattern is an operation execution paradigm or a
       product flow paradigm.
 5 Output every operation of the plan in the form of an operation DSL
       program and every product of the plan in the form of a product DSL
       program.
 6
 7 Here is an example of how to generate plan in DSL for a biology protocol.
 8
 9 EXAMPLE:
10
11 {example protocol title}
12
13 Here are some extra details about the protocol:
14
15 {example protocol description}
16
17 example plan in DSL:
18
19 {example plan}
20
21 YOUR TASK:
22 Generate plan in DSL for a protocol for {title}.
23
24 Here are some extra details about the protocol:
25
26 {details}
27
28 You can choose to instantiate the following DSL specifications to
       construct the DSL program:
29
30 Operation-view DSL specification:
31 {Operation-DSL}
32
33 Product-view DSL specification:
34 {Product-DSL}
35
36 Your plan in DSL program:
```

### D.6 EXTERNAL DESIGNER

The external designer combines (i) deductive verification through DSL; and (ii) self-improvement by the LLM (Madaan et al., 2023). In EE, the external verifier is provided by the operation-centric view DSL and performs checks on two main aspects: (i) whether the precondition of each operation is an intermediate product of a previous step rather than appearing from nowhere; and (ii) whether the postcondition of each operation is used in subsequent steps rather than being omitted. Similarly, in EE+, the external verifier is provided by the DSL with a dual representation, focusing on cross-verifying the parallel dual tracks (the two perspectives of the DSL program). It checks whether the corresponding operation causes each *status transition* of the product: (i) whether the product in each product-view program is the output of its preceding operation; and (ii) whether the product in each product-view program is the input for its succeeding operation. If a mismatch occurs, the verifier generates corresponding error messages, such as *"Error: The product {product} required by operation {operation} at step {i} is not available from previous steps."* These error messages are then fed into the feedback-refine loop as feedback for the LLM to revise the plan. The loop terminates when the program passes the verification or reaches the maximum number of iterations, and the best result is retained based on the verification information.

```
1 [Prompt for refining the plan according to the feedback vertified by
      operation-centric view
2 DSL]
3 Your task is to improve a biology experimental protocol plan represented
      in domain-specific language (DSL) based on provided feedback.
4 The input plan in DSL consists of multiple DSL programs, each
      representing one step in the experimental protocol planning process,
      arranged in top-down order to indicate the execution sequence of
      operations.
5 Each DSL program has the following format:
6 {
7     "Operation": ,    // Operation verb
8     "Precond": {      // Precondition for this step
9         "SlotArgNum": ,   // Number of arguments for the precondition
10        "SlotArg":        // Input product for this step
11    },
12    "Execution": {
13        "DeviceType": ,   // Execution device for the operation
14        "Config": {       // dict of execution arguments - values
15            Argkey: Argvalues
16        }
17    },
18    "Postcond": {     // Postcondition for this step
19        "EmitArgNum":,    // Number of arguments for the postcondition
20        "EmitArg":        // Output product for this step
21    }
22 }
23
24 The provided feedback indicates errors that occurred when compiling the
      DSL programs. You need to correct the program to ensure that the
      product is properly transferred between each step, i.e., the input
      product of each step must be the output from a previous step (except
      for the first step), and verify whether the output of each step is
      used as the input for subsequent steps (except for the final step).
```

```
25 If you believe the error in a particular step is due to the step
      preparing reagents rather than using a previous intermediate product,
       you can ignore this error.
26
27 Output your refined plan in DSL, returning a JSON block without any
      additional information or comments.
28
29 YOUR TASK:
30 Refine the plan in DSL for a protocol for {title}.
31
32 Here are some extra details about the protocol:
33
34 {details}
35
36 Refine the following plan:
37
38 {plan}
39
40 Here is the feedback of the plan:
41
42 {feedback}
43
44 Your refined plan in DSL:
```

```
1 [Prompt for refining the plan according to the feedback vertified by DSL
      with dual representation]
2 Your task is to improve a Biology experimental protocol plan represented
      in domain-specific language (DSL) based on provided feedback.
3 The input plan in DSL consists of multiple DSL programs from two
      perspectives: operation-view and product-view. The DSL programs from
      these two perspectives alternate and constrain each other.
4
5 This is the format of a product-view DSL program:
6 // Each product view DSL program represents the state of the product at
      that moment.
7 {
8    Pred: <Operation>,      // Pred represents the operation that
     precedes the creation of this product, need to align to the operation
      name in the operation view DSL program. If the product is in its
     initial state, return "".
9    FlowUnit: {     // FlowUnit defines the properties of the product
     being processed.
10       Component: ,    // Component represents the actual product or
     material being processed, need to be the formal name of the component
     .
11       ComponentType: Gas|Liquid|Solid|Semi-Solid|Mixture|
     ChemicalCompound|BiologicalMaterial|Reagent|PhysicalObject|File/Data,
           // ComponentType describes the type of the component, which
     can be one of the following: Gas, Liquid, Solid, Semi-Solid, Mixture,
```

```
        ChemicalCompound, BiologicalMaterial, Reagent, PhysicalObject, or
     File/Data.
12      RefName: ,       // RefName is the reference name used to uniquely
     identify this component, need to align to the operation-view program
13      UnitArgType: MAT | PROD,    // UnitArgType specifies whether this
     is a material (MAT) or a product (PROD).
14      Vol: ,       // Vol represents the volume or quantity of the
     component.
15      Container: ,    // Container indicates the type of container or
     storage used for this component. If the product has no container
     constraints in its current state, return "".
16      Cond: {          // Cond defines the specific conditions under
     which the operation is carried out, which is expressed as key-value
     pairs.
17          ArgKey: ArgValues
18      }
19    },
20    Succ: <Operation>      // Succ represents the operation that follows
     the creation of this product. If the product is in its final state,
     return "".
21 }
22
23 This is the format of an operation-view DSL program:
24 // Each operation view DSL program represents a sequence of operations
     that alters the state of the product.
25 {
26    Operation: ,    // Operation verb
27    Precond: {      // Precondition
28        SlotArgNum: ,   // Number of arguments for the precondition
29        SlotArg:        // SlotArg represents the input product or
     material required for this operation, using formal component names
     from the product perspective DSL program, with serial numbers to
     distinguish repeated components in different states.
30    },
31    Execution: {
32        DeviceType: ,   // Execution device for the operation
33        Config: {       // dict of execution arguments - values
34            ArgKey: ArgValues
35        }
36    },
37    Postcond: {     // Postcondition
38        EmitArgNum: ,    // Number of arguments for the postcondition
39        EmitArg:         // EmitArg represents the output product or
     material resulting from the operation, using formal component names
     from the product perspective DSL program, with serial numbers to
     distinguish repeated components in different states.
40    }
41 }
42
```

```
43 The provided feedback indicates errors that occurred when compiling the
      DSL programs. You need to correct the program to ensure that the
      state changes of each product's RefName in the Product-view are
      caused by the corresponding operations in the Operation-view.
44 If you believe the error in a particular step is due to a mismatch in
      product names between the two perspectives rather than an actual
      error, you can ignore this error.
45
46 Output your refined plan in DSL, returning a JSON block without any
      additional information or comments.
47
48 YOUR TASK:
49 Refine the plan in DSL for a protocol for {title}.
50
51 Here are some extra details about the protocol:
52
53 {details}
54
55 Refine the following plan:
56
57 {plan}
58
59 Here is the feedback of the plan:
60
61 {feedback}
62
63 Your refined plan in DSL:
```

### D.7 COMPUTING LOAD OF THE MACHINE DESIGNERS

For automated representation generation, we primarily used GPT-4o mini with OpenAI's Batch API[5] for preprocessing, incurring a cost of approximately \$60 across four domains. The design of the DSLs was executed on a MacBook with an M2 chip, running 1,000 iterations to ensure convergence. This process required an average of 55 seconds per iteration for the operation-centric view DSL and an average of 2 seconds per iteration for the product-centric view DSL. For the machine designer, we primarily utilized GPT-4o mini combined with RAG for design, with a total cost of approximately \$10 (7 methods, 140 protocols). In summary, the overall computational load is relatively low, highlighting the accessibility of our machine designers when utilizing the proposed representations and the corresponding automatic representation generation modules.

## E  DATA COLLECTION

### E.1 CORPORA SOURCES

The corpora $\mathcal{C}$ for the automatic generation of representations (Sec. 3.1) and the corpora for selecting the testing set (Sec. 4.1) are both retrieved from open-sourced websites run by top-tier publishers, including Nature's Protocolexchange[6], Cell's Star-protocols[7], Bio-protocol[8], Wiley's Current Pro-

---

[5] https://platform.openai.com/docs/guides/batch/batch-api
[6] https://protocolexchange.researchsquare.com/
[7] https://star-protocols.cell.com/
[8] https://bio-protocol.org/en

tocols[9], and Jove[10]. These sources compile a dataset of 15,837 experimental protocols across four domains: Genetics (8794 protocols), Medical (7351), Ecology (812), and Bioengineering (3597), with minimal overlap between them. We aggregated the corpora and analyzed the themes of the protocols according to the first- and second-level labels attached to them. We adopt measures to ensure that $\mathcal{C}$ is mutually exclusive with the testing set.

Other domains, such as Physics and Chemistry, are also representative domains of experimental sciences, besides Biology, Medical, and Ecology. The preliminary factor that restricts our current scope is data accessibility. Due to the higher cost of accessing the corpora of protocols for conducting physics and chemistry experiments, for example, mining the protocol from the "method" section of relevant published papers, we leave the application to Physics and Chemistry for future work.

### E.2 ELIMINATING THE RISK OF DATA LEAKING

We employ the broadly accepted standard operating process to empirically verify that LLMs have not memorized the data we use. We adopt the methodology outlined in Section 5.2 of *Skywork* (Wei et al., 2023) and draw upon recent studies on detecting memorization in LLMs (Carlini et al., 2021; 2022). Specifically, we use gpt-4o mini to synthesize data resembling the style of steps from novel protocols, and then calculate the perplexity on the test set and reference set. Since the reference set is newly generated, we consider it clean, not belonging to any training set of any model.

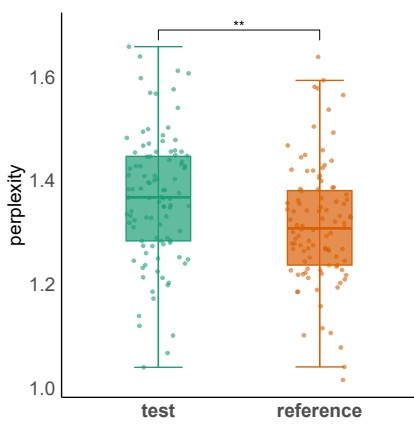

Figure A1: **Comparison between the perplexity of the test set and the reference set**

We randomly sample 100 sequences each from the test set and the reference set of the novel protocols. Each sequence corresponds to a single procedural step described in NL. We truncate the final 50 tokens of each sequence, retaining the prefixes. These prefixes are then used as prompts for the LLM to predict the next 50 tokens, for which we calculate the perplexity. If the perplexity of the test set is significantly lower than that of the reference set, the test set might have appeared in the model's training phase.

The results indicate that the LLM's average perplexity on the test set is significantly higher than that on the reference set ($t(198) = 3.040, \mu_d < 0, p < .05$; see Fig. A1), suggesting that the LLM encounters greater uncertainty with the novel protocols in the test set. This finding implies that for a published, widely accepted, and standardized operating process, there is no evidence to suggest that the LLM has memorized the data.

### E.3 ON THE DIVERSITY OF NOVEL PROTOCOLS

Assessing diversity among novel protocols is both informative and meaningful. To further support our analysis, we incorporate a t-SNE visualization of the experimental objectives (described in natural language) for the novel protocols we select, as shown at Fig. A2. The results demonstrate a well-dispersed distribution, indicating a sufficient level of diversity among the protocols.

### E.4 SHOWCASES

```
1 [Protocol 1 - Bioengineering]
2 Preparation of lysates
3 1. Harvest approximately 1 x 10^7 cells by centrifugation at 2000 RPM for
      5 min. Aspirate media and resuspend cell pellet with 1 mL of ice-
      cold PBS and transfer to a 1 mL centrifuge tube. Microcentrifuge at
      2000 RPM for 5 min at 4 °C.
```

---

[9]https://currentprotocols.onlinelibrary.wiley.com/
[10]https://www.jove.com/

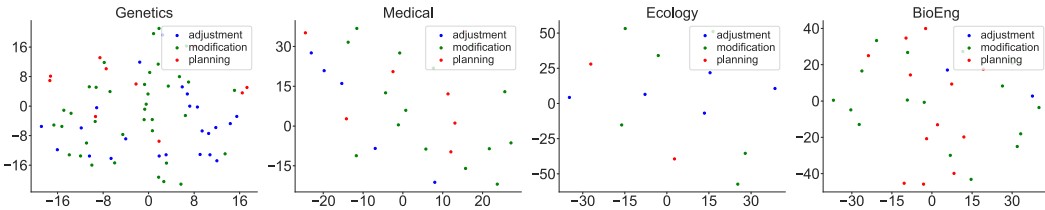

Figure A2: **Visualization of diversity between novel protocols**

```
 4  2. Aspirate PBS, and then add Hypotonic Buffer (supplemented with 1%
       Triton X-100, to disrupt membrane and cytoskeleton-bound MEKK1
       fractions).
 5  3. Cell lysates are homogenized by passing through 22-gauge needles, and
       tubes are put on ice for 15 min to complete the lysis. Crude extracts
        are then centrifuged at 2500 RPM for 5 min. Supernatants are
       transferred to fresh centrifuge tubes, and cold 5 M NaCl is added to
       each sample to make a salt concentration of between 0.7-1.0 M to
       disrupt protein-protein interactions.
 6  4. Spin the crude extracts by ultracentrifugation at 55000 RPM to
       properly pellet residual insoluble proteins from the extract.
       Transfer supernatants into fresh centrifuge tubes.
 7  Immunoprecipitation
 8  5. Rinse Protein A beads in Hypotonic Buffer and place on ice until ready
        for use.
 9  6. Take a volume of cell lysates (prepared as described above), and
       dilute with Hypotonic Buffer to 250-500 mM salt to enable protein-
       protein interactions.
10  7. Add 2 µg of preclearing antibody to the diluted lysate (e.g., anti-Myc
        or anti-p65), vortex, add 50 µL of Protein A beads, and rock for 45
       min.
11  8. Touchspin samples, and transfer supernatant to a fresh tube.
12  9. Add 2 µg of polyclonal anti-MEKK1 to the lysates, and rock for 1 h.
       After this period, add 50 µL of Protein A beads and rock tubes at 4 °
       C for 1 h.
13  10. Touchspin beads, wash beads with hypotonic buffer (supplemented with
       NaCl to a concentration of 300 mM), vortex, and rock for 10 min. In
       total, 3-5 washes of the beads are performed.
14  11. Finally, wash once with Hypotonic Buffer, and resuspend in Kinase
       Assay Buffer. Purified MEKK1 may be stored by snap-freezing in liquid
        nitrogen and long-term storage at -80 °C. Kinase assay Following
       preparation of MEKK1 immunoprecipitates (as above), incubate with 7 µ
       g of JNKK1(K131M) along with 5 µCi of ATP in Kinase Assay Buffer for
       30 min at 30 °C."
15
16  [Protocol 2 - Genetics]
17  1. Note that everything is in DEPC water. Inoculate W303a cells
       expressing different TOR1-RR variants in 2 mL SC medium overnight.
18  2. Subculture the cells starting from OD600=0.1 in 10 mL SC media, shake
       vigorously at 30 °C, 300 RPM for around 4-6 h until OD600=0.4-0.5.
```

```
19 3. Collect the cells by spinning down without freezing on ice. Discard
      supernatant.
20 4. Re-suspend cells with 1 mL water and transfer to a 1.5 eppendorf tube,
       quickly spin down at 3,000 x g for 15 sec.
21 5. Re-suspend cell pellet in 400 μL of AE buffer at room temperature.
22 6. Add 40 μL 10% SDS (final around 1%) and vortex briefly at room
      temperature (RT).
23 7. Immediately add 500 μL hot phenol/AE (put in 65 °C for 10 min before
      use), vortex vigorously for 1 min. Incubate at 65 °C for 5 min.
      Briefly vortex every 30 sec.
24 8. Immediately freeze by dumping into liquid nitrogen. Wait to thaw at RT
       (put in 30 °C to thaw may crack the tube).
25 9. Centrifuge for 10 min on a standard laboratory microfuge at 20,000 x g
       at RT.
26 10. Transfer around 400 μL supernatant to a new eppendorf tube. Recycle
      the lower phenol fraction carefully following the chemical safety
      protocol in your laboratory.
27 11. Add equal volume (400 μL) phenol: CHCl3/AE-Na. Vortex vigorously for
      1 min at RT.
28 12. Spin down at 20,000 x g for 5 min in a standard laboratory microfuge.
29 13. Transfer supernatant (around 350 μL) to a fresh 1.5 mL eppendorf tube
      . Add CHCl3: isoamyl alcohol (24:1). Vortex vigorously for 1 min at
      RT.
30 14. Transfer aqueous supernatant to a fresh 1.5 mL microfuge tube. If
      white cloudy precipitate is observed between the aqueous phase and
      organic phase, repeat steps 17-18.
31 15. Add 1/10 volume of 3 M NaOAc (pH 5) and vortex vigorously. Add 2.5
      volumes of ethanol. Vortex again.
32 16. Place at -20 °C for at least 30 min.
33 17. Spin down in the microfuge at 20,000 x g, 15 min at 4 °C. RNA pellet
      is usually visible.
34 18. Add ice-cold 75% EtOH, place at 4 °C for around 10 min. Vortex and
      spin down on microfuge 20,000 x g, 15 min at 4 °C. Discard
      supernatant. Suck out the liquid droplets in the tube. The white RNA
      pellet will turn clear when it dries out. Add 30-50 μL ddH2O (DEPC)
      immediately after it becomes clear. Do not let the RNA over-dry,
      which will make it difficult to dissolve. If RNA pellet is over-dry,
      dissolve RNA at 37 °C for 30 min. Store RNAs at -80 °C for more than
      2 months."
35
36 [Protocol 3 - Medical]
37 1. Passage through a 45 μm filter. Add 100 μL/well of 100 μg/mL salmon
      sperm DNA to a 96-well Microtest assay plate.
38 2. Wrap the plate with plastic wrap and incubate at 4 °C overnight.
39 3. Discard the coating antibody solution and wash the plate with 1x PBS-
      Tween 6 times.
40 4. Dry the plate and add 100 μL of blocking solution per well to the
      plate.
41 5. Incubate the plate at room temperature (RT) for 1.5 h.
```

42 6. Discard the blocking solution and wash the plate with 1x PBS-Tween 5
      times.
43 7. Dry the plate and keep it at 4 °C for later use.
44 8. Harvest the spleen and create a single-cell suspension by gently
      smashing spleen pieces with the frosted surface of a pair of
      microscope slides in 5 mL of DMEM.
45 9. Transfer the cells into 50 mL conical tubes and spin down the cells at
       300 RCF for 5 min at 4 °C.
46 10. Discard the supernatant with aspiration without disturbing the pellet
      .
47 11. Re-suspend the cells with 5 mL of 0.17 M ammonium chloride and keep
      the cells on ice for 5 min.
48 12. Add 15 mL DMEM to the cells and spin at 300 RCF for 5 min at 4 °C.
49 13. Discard the supernatant and re-suspend the cells with 20 mL of DMEM
      and count the cells.
50 14. Re-suspend 2 x 10^7 cells in 2 mL of 10% DMEM and make a three-fold
      serial dilution (a total of 8 dilutions) with 10% DMEM.
51 15. Add 50 $\mu$L/well of the serial dilutions on the DNA-coated plate and
      centrifuge at 300 RCF for 5 min at 4 °C.
52 16. Incubate the cells at 30 °C for 2 h in a cell-culture incubator with
      6% CO2.
53 17. Add 50 $\mu$L/well of biotin-conjugated anti-IgM or anti-IgG (1:350 in
      10% DMEM) to the cells.
54 18. Centrifuge the cells at 300 RCF for 5 min at 4 °C and incubate the
      cells overnight in a cell-culture incubator with 6% CO2.
55 19. Discard the cells and wash the plates 10 times with 10x PBS-Tween 20.
56 20. Dry the plates and add 50 $\mu$L of streptavidin alkaline phosphatase
      (1:1,000 in 1% BSA/PBS) to the plate.
57 21. Incubate the plate at RT for 1 h and wash the plate 10 times with 10x
       PBS-Tween 20.
58 22. Dry the plate and add 50 $\mu$L/well of 1 mg/mL BCIP in AMP buffer to
      develop the plate.
59 23. When the spots are clearly visible under a dissecting microscope,
      stop the development by discarding the BCIP solution and rinsing the
      plate with tap water thoroughly.
60 24. Spots can be counted using a dissecting microscope or using an
      ELISpot reader."

## F    REPRODUCIBILITY

The project page with supplementary files for reproducing the results of this paper will be available at `https://autodsl.org/procedure/papers/iclr25shi.html`.

## G    LIMITATIONS

As a representation designed for a relatively new problem, the design and evaluation of the proposed framework come with limitations, leading to further investigations:

- Overall, our method achieves promising results across the four domains. Specifically, it performs best in experimental design for Genetics, shows comparable effectiveness in Medical and Bio-engineering, but is less effective in Ecology. Notably, the Genetics corpus is the largest among

the four domains, while the Ecology corpus is significantly smaller than the others. These observations suggest a potential positive correlation between the size of the domain-specific corpus and the "quality" of the resulting DSL. In other words, a larger corpus may lead to a "better" representation, thereby influencing the outcomes of protocol design. This hypothesis necessitates further investigation through rigorously designed experiments and carefully defined metrics for evaluating what constitutes a "better" representation.

- We majorly consider the imperative programming DSLs as the implementation of representation in this work. This raises the question of whether incorporating objective-oriented programming paradigms could enhance the representation of complex entities within protocols, particularly the properties of reagents and intermediate products. If we are able to make the DSLs model the fine-grained reactions between different components and automate the design of those DSLs based on a broader source of data, such as the Wikipedia pages, we can ultimately manage to build up a symbolic digital twin for a domain-specific system, such as the cell cultivation environment. Such simulation systems may greatly benefit protocol design with their power of prediction, explanation, and counterfactual analysis.

- Can we explicitly extend our proposed representation to a hierarchical graph, thereby establishing the foundation for employing the advanced algorithms on graph routing and graph optimization? Results on the hierarchical graph can also serve as a external heuristic and constraint for LLM-based protocol designers. This hybrid approach may combine both the advantages of LLMs, *i.e.*, exploitation of background knowledge, and those of classical algorithms, *i.e.*, white-boxed properties with high explainability.

- Can we apply the representation and the automatic representation generator to other critical domains with a high demand for automating procedure design, such as designing product route sheets for advanced manufacturing?

With many questions unanswered, we hope to explore more on automated protocol design for self-driving laboratories and beyond.

## H   THE AUTOMATICALLY GENERATED REPRESENTATIONS

### H.1   OPERATION-CENTRIC VIEW DSL

```
1  {
2      "Operation": "Precipitate",
3      "pattern_0": {
4          "Precond": {
5              "SlotArgNum": 2,
6              "SlotArg": ["Liquid", "Solid"]
7          },
8          "Execution": {
9                  "DeviceType": "falcon tube",
10                 "Capacity": "15 mL",
11                 "Config": {}
12         },
13         "Postcond": {
14             "EmitArgNum": 1,
15             "EmitArg": ["Liquid"]
16         },
17         "Example": [
18             "Precipitate RNA by adding 600 μL of 100 % EtOH, 20 μL of 3 M
    NaOAc ( pH 5.5 ), and 3 μL of glycogen .",
19             "Precipitate the DNA in each tube by adding 20 μl of 3 M
    sodium acetate ( pH 5.2 ) and 550 μl of 100 % ethanol .",
20             "Ethanol precipitate the RNA by adding 5 μl 3 M sodium
    acetate ( pH 5.2 ) ,2 μl of glycogen ( 20 mg / ml ) ,and 171 μl of
    100 % ethanol .",
```

```
21          ]
22      },
23      "pattern_1": {
24          "Precond": {
25              "SlotArgNum": 4,
26              "SlotArg": ["Liquid", "Liquid", "Liquid", "Liquid"]
27          },
28          "Execution": {
29                  "DeviceType": "centrifuge",
30                  "capacity": "1.5 ml",
31                  "Config": {
32                      "time": "10-15 min",
33                      "speed": ["12,000 × g", "20,000 × g"],
34                      "temperature": "4 °C",
35                  }
36          },
37          "Postcond": {
38              "EmitArgNum": 1,
39              "EmitArg": ["Solid"]
40          },
41          "examples": [
42              "Precipitate the cell debris in the lysate by centrifugation
    at 20,000 × g for 10-15 min at 4 ° C.",
43              "precipitate DNA with 13.5 μL of following mixture (1 μL of
    20 mg / ml Glycogen , 12.5 μL of 3 M NaOAc [pH 5.3]) and 340 μL
    ethanol.",
44              "precipitate the total RNA by centrifuging at 12,000 × g for
    15 min at 4 ° C."
45          ]
46      }
47 },
48 {
49      "Operation": "Spin",
50      "pattern_0": {
51          "Precond": {
52              "SlotArgNum": 2,
53              "SlotArg": ["Liquid", "Liquid"]
54          },
55          "Execution": {
56              "DeviceType": "spin plate",
57              "Config": {
58                  "time": ["1 min"]
59              }
60          },
61          "Postcond": {
62              "EmitArgNum": 1,
63              "EmitArg": ["Physical Object"]
64          },
65          "Example": [
```

```
66              "Nuclei washing and tagmentation: Spin down nuclei at 600 g
        for 10 mins at 4 ° C , resuspended with 50 μL Complete Buffer.",
67              "Spin the sample at 4,000 × g at 4 ° C until the volume
        reduces to about 1 mL. Quantify protein concentration as described in
         step 60.",
68              "Spin down the 15 mL tubes at 2,500 ×g and 4 ° C for 20 min
        .",
69              "Spin for 2 min at 1,000 x g. Save a few μL of concentrated
        sample to run on an agarose gel later.",
70              "Spin the tube for 30 sec at 12,000 x g to consolidate the
        gel at the bottom of the tube.",
71              "Spin plate at 300×g for 1 min to collect liquid at the
        bottom of the wells.",
72              "Once NXT PCR program is complete, quick spin the sample tube
         then place it on the magnet for 1 min. Transfer the supernatant
        containing the amplified mRNA-seq library into a new PCR tube.",
73              "Spin down 2 mill nuclei at 600×g for 5 min (whole liver
        nuclei) or use a magnet (bead-bound nuclei)."
74          ]
75      },
76      "pattern_1": {
77          "Precond": {
78              "SlotArgNum": 1,
79              "SlotArg": ["Mixture"]
80          },
81          "Execution": {
82              "DeviceType": "microcentrifuge",
83              "Config": {}
84          },
85          "Postcond": {},
86          "Example": [
87              "Small volumes, 1-3 mL should be spun in a small tube where
        these fewer EVPs can more readily be collected.",
88              "Briefly spin down the bead-lysate mixture.",
89              "Spin down the mix tube to eliminate bubbles/air in a bench
        microcentrifuge. Add 19 μL of the mix to each well."
90          ]
91      },
92      "pattern_2": {
93          "Precond": {
94              "SlotArgNum": 1,
95              "SlotArg": ["Liquid"]
96          },
97          "Execution": {
98              "DeviceType": "centrifuge",
99              "Config": {
100                 "speed": ["800 g"],
101                 "time": ["7 min"]
102             }
103         },
```

```
104        "Postcond": {
105            "EmitArgNum": 1,
106            "EmitArg": ["Liquid"]
107        },
108        "Example": [
109            "Spin lysate at 14 krcf for 10 min at 4 ° C; transfer cleared
        lysate to new tube.",
110            "Spin down the beads for 60 s at 2,000 x g. Discard the
        supernatant by carefully pipetting out the buffer.",
111            "Spin at 12,000 × g until the total volume in both filters is
         reduced to 120 μL (<=30 min). Keep aside 5 μL of purified labeled
        histone for SDS-PAGE analysis.",
112            "Quickly spin the FACS tube to allow the cell suspension to
        pass through the filter to remove undigested large tissue debris.",
113            "Spin once for 7 min at 800 g. Use the BD cytofix/cytoperm
        kit according to the manufacturer's instructions and thereafter add
        antibodies for intracellular detection of IFN and TNF."
114        ]
115    }
116 },
117 {
118    "Operation": "Sonicate",
119    "pattern_0": {
120        "Precond": {
121            "SlotArgNum": 1,
122            "SlotArg": ["Liquid"]
123        },
124        "Execution": {
125            "DeviceType": "sonicator",
126            "Config": {
127                "time": ["20 - 30 s"]
128            }
129        },
130        "Postcond": {
131            "EmitArgNum": 1,
132            "EmitArg": ["Semi-Solid"]
133        },
134        "Example": [
135            "Sonicate the pellet suspension on ice under a 50 % duty
        cycle for 5 min.",
136            "Agarose gel of sonicated Arabidopsis chromatin.",
137            "Sonicate proteoliposomes for 20 - 30 s or 3 times for 10 s,
        placing on ice in between sonication, if necessary.",
138            "The lipid suspension is sonicated to form small unilamellar
        vesicles (SUVs)."
139        ]
140    },
141    "pattern_1": {
142        "Precond": {
143            "SlotArgNum": 2,
```

```
144                "SlotArg": ["Liquid", "Solid"]
145            },
146            "Execution": [
147                {
148                    "DeviceType": "bransonic",
149                    "Config": {
150                        "temperature": ["60 ° C"],
151                        "time": ["90 min"]
152                    }
153                },
154                {
155                    "DeviceType": "sonicator",
156                    "Config": {}
157                }
158            ],
159            "Postcond": {},
160            "Example": [
161                "Sonicate 10 μg BAC DNA or 50 μg genomic DNA in total (you
        will recover 10 % DNA after sonication and size selection).",
162                "Sonicated chromatin is immunoprecipitated with the chosen
        antibodies and non-enriched chromatin washed with a series of washing
         buffers.",
163                "If the herring sperm DNA has not been sufficiently sonicated
         or too much has been used, the DNA pellet might not adhere to the
        microfuge tube and can be lost with the ethanol.",
164                "Sonicate the lipid tube to dissolve lipids with the mineral
        oil for 90 min at 60 ° C by using Bransonic."
165            ]
166        }
167 }
```

## H.2 PRODUCT-FLOW-CENTRIC VIEW DSL

```
1 {
2     "Pred": "Modification Operations",
3     "FlowUnit": {
4         "Component": "FBS",
5         "ComponentType": "Liquid",
6         "UnitArgType": "MAT"
7         "Vol": ["0.1 mL", "0.5 mL", "1 mL", "1.5 mL", "2 mL", "3 mL", "5
    mL", "10 mL", "25 mL", "50 mL", "400 μL", "500 μL", "500 mL"],
8         "Container": "Tube",
9         "Cond": {
10             "Concentration": ["0.5%", "1%", "2%", "2.5%", "5%", "10%",
    "15%", "20%", "30%", "50%", "90%", "100%"],
11             "Temperature": ["-150°C", "4°C", "18°C-26°C", "37°C", "56°C
    "],
12             "State": "heat-inactivated"
13         }
14     "Succ": "Transfer Operations"
```

```
15 },
16 {
17     "Pred": "Detection and Measurement Operations",
18     "FlowUnit": {
19         "Component": "ethidium bromide",
20         "ComponentType": "Solid",
21         "UnitArgType": "MAT",
22         "Vol": ["0.25 μL/mL", "0.5 μL/mL", "2 - 3 μL", "10 μL", "15 μL",
    "10 μg/mL", "0.5 μg/μL"],
23         "Container": "Flask",
24         "Cond": {
25             "Concentration": ["0.0024%", "0.3 - 10 μg/mL", "1.5% (w/v)",
    "5 μM", "1:1000"],
26             "Temperature": ["25°C", "room temperature"],
27             "State": ["toxic", "carcinogenic"],
28             "Charge": ["positively charged"]
29         }
30     },
31     "Succ": "Modification Operations"
32 },
33 {
34     "Pred": "Transfer Operations",
35     "FlowUnit": {
36         "Component": "gel",
37         "ComponentType": "Semi-Solid",
38         "UnitArgType": "MAT",
39         "Vol": ["0.5 mL"],
40         "Container": ["Gel Cassette", "Tank", "Tube"],
41         "Cond": {
42             "Impedance": ["under 20 kOhm"],
43             "Size": ["50-250 nt"]
44         }
45     },
46     "Succ": "Transfer Operations"
47 }
```

# I CASE STUDIES

## I.1 CASE STUDY: CONTRIBUTIONS OF THE BUILDING BLOCKS

Part of protocol designed by EE+:

```
1 {
2     "Pred": "",
3     "FlowUnit": {
4         "Component": "Lysis solution",
5         "ComponentType": "Liquid",
6         "RefName": "Lysis_solution-1",
7         "UnitArgType": "MAT",
8         "Vol": "50 μL",
9         "Container": "",
```

```
10          "Cond": {
11              "State": "Liquid"
12          }
13      },
14      "Succ": "Pipette"
15 },
16
17 {
18      "Operation": "Pipette",
19      "Precond": {
20          "SlotArgNum": 1,
21          "SlotArg": [
22              "Lysis_solution-1"
23          ]
24      },
25      "Execution": {
26          "DeviceType": "Pipette",
27          "Config": {
28              "time": "10 times",
29              "volume": "50 μL"
30          }
31      },
32      "Postcond": {
33          "EmitArgNum": 1,
34          "EmitArg": [
35              "Lysis_solution-2"
36          ]
37      }
38 }
```

Part of protocol designed by EE:

```
1 {
2      "Operation": "Add",
3      "Precond": {
4          "SlotArgNum": 1,
5          "SlotArg": [
6              "Triton-X"
7          ]
8      },
9      "Execution": {
10          "DeviceType": "8-channel pipette",
11          "Config": {
12              "Volume": "1% solution"
13          }
14      },
15      "Postcond": {
16          "EmitArgNum": 1,
17          "EmitArg": [
18              "Triton_X_Solution"
19          ]
```

```
20       }
21  }
22
23  {
24       "Operation": "Run",
25       "Precond": {
26           "SlotArgNum": 1,
27           "SlotArg": [
28               "Cell_Lysis_Mixture"
29           ]
30       },
31       "Execution": {
32           "DeviceType": "Thermal Cycler",
33           "Config": {
34               "Temperature": "70 °C",
35               "Time": "15 min"
36           }
37       },
38       "Postcond": {
39           "EmitArgNum": 1,
40           "EmitArg": [
41               "cDNA_Reverse_Transcription"
42           ]
43       }
44  }
```

Part of protocol designed by EI:

```
1  {
2       "Operation": "Run",
3       "Precond": {
4           "SlotArgNum": 1,
5           "SlotArg": [
6               "Cell_Lysis_Mixture"
7           ]
8       },
9       "Execution": {
10           "DeviceType": "Thermal Cycler",
11           "Config": {
12               "Temperature": "70 °C",
13               "Time": "15 min"
14           }
15       },
16       "Postcond": {
17           "EmitArgNum": 1,
18           "EmitArg": [
19               "cDNA_Reverse_Transcription"
20           ]
21       }
22  }
```

Part of protocol designed by II:

```
1  "reverse_transcribe": {
2      "muscs": "cells",
3      "buffer": "RT buffer",
4      "enzyme": "reverse transcriptase",
5      "incubation_time": "60 minutes",
6      "temperature": "42"
7  }
8
9  "prepare_single_cell_suspension": {
10     "input_cells": "lysed cells"
11 }
```

Part of protocol designed by IB:

```
1  "reverse_transcribe": {
2      "muscs": "RNA",
3      "buffer": "reverse transcription buffer",
4      "enzyme": "reverse transcriptase",
5      "incubation_time": "60 minutes",
6      "temperature": "42"
7  }
8
9  "prepare_single_cell_suspension": {
10     "input_cells": "single-cell samples"
11 }
```

Part of protocol designed by FB:

```
1  "sort_single_cell": {
2      "plate": "PCR plate",
3      "nozzle_size": "100 μm",
4      "mode": "single-cell purity"
5  }
```

## I.2 CASE STUDY: HANDLING DIFFERENT TASK COMPLEXITIES

Part of protocol designed in Planning:

```
1  {
2      "Operation": "Obtain",
3      "Precond": {
4          "SlotArgNum": 1,
5          "SlotArg": [
6              "File/Data"
7          ]
8      },
9      "Execution": {
10         "DeviceType": "QIAGEN Blood & Cell Culture DNA Maxi Kit",
11         "Config": {}
12     },
13     "Postcond": {
```

```
14            "EmitArgNum": 1,
15            "EmitArg": [
16                "HMW genomic DNA"
17            ]
18        }
19 }
```

Part of protocol designed in Modification:

```
1 {
2      "Operation": "Centrifuge",
3      "Precond": {
4          "SlotArgNum": 1,
5          "SlotArg": [
6              "Serum_Plasma_in_PBS-1"
7          ]
8      },
9      "Execution": {
10         "DeviceType": "Ultracentrifuge",
11         "Config": {
12             "speed": [
13                 "12,000 × g"
14             ],
15             "time": [
16                 "20 min"
17             ],
18             "temperature": [
19                 "4 °C"
20             ]
21         }
22     },
23     "Postcond": {
24         "EmitArgNum": 1,
25         "EmitArg": [
26             "Pellet-1"
27         ]
28     }
29 }
```

Part of protocol designed in Adjustment:

```
1 {
2      "Operation": "Incubate",
3      "Precond": {
4          "SlotArgNum": 2,
5          "SlotArg": [
6              "Washed sections with 1st antibody-1",
7              "2nd antibody mixture-1"
8          ]
9      },
10     "Execution": {
11         "DeviceType": "Moistening box",
```

```
12        "Config": {
13            "temperature": "37 C",
14            "time": "1 h"
15        }
16    },
17    "Postcond": {
18        "EmitArgNum": 1,
19        "EmitArg": [
20            "Sections with 2nd antibody-1"
21        ]
22    }
23 }
```

### I.3  CASE STUDY: GENERALITY ACROSS DOMAINS

Part of protocol designed for Bioengineering:

```
1 {
2      {
3          "Operation": "Incubate",
4          "Precond": {
5              "SlotArgNum": 2,
6              "SlotArg": [
7                  "Lysis Mixture-2",
8                  "Stop Buffer-1"
9              ]
10         },
11         "Execution": {
12             "DeviceType": "Thermocycler",
13             "Config": {
14                 "temperature": "65 °C",
15                 "time": "30 min"
16             }
17         },
18         "Postcond": {
19             "EmitArgNum": 1,
20             "EmitArg": [
21                 "Neutralized Mixture-1"
22             ]
23         }
24     },
25     {
26         "Pred": "Incubate",
27         "FlowUnit": {
28             "Component": "Neutralized Mixture",
29             "ComponentType": "Mixture",
30             "RefName": "Neutralized Mixture-1",
31             "UnitArgType": "PROD",
32             "Vol": "60 μL",
33             "Container": "0.2-ml PCR tube",
34             "Cond": {
```

```
35                      "State": "Neutralized"
36                   }
37                },
38                "Succ": "Mix"
39             },
40             {
41                "Operation": "Elute",
42                "Precond": {
43                   "SlotArgNum": 1,
44                   "SlotArg": [
45                      "Washed Beads-1"
46                   ]
47                },
48                "Execution": {
49                   "DeviceType": "Centrifuge",
50                   "Config": {
51                      "time": "1 min"
52                   }
53                },
54                "Postcond": {
55                   "EmitArgNum": 1,
56                   "EmitArg": [
57                      "Eluted Product-1"
58                   ]
59                }
60             },
61             {
62                "Pred": "Elute",
63                "FlowUnit": {
64                   "Component": "Eluted Product",
65                   "ComponentType": "BiologicalMaterial",
66                   "RefName": "Eluted Product-1",
67                   "UnitArgType": "PROD",
68                   "Vol": "50 μL",
69                   "Container": "0.2-ml PCR tube",
70                   "Cond": {
71                      "State": "Eluted"
72                   }
73                },
74                "Succ": "Incubate Clear"
75             },
76             {
77                "Operation": "Quantify",
78                "Precond": {
79                   "SlotArgNum": 1,
80                   "SlotArg": [
81                      "Clear Eluted Solution-1"
82                   ]
83                },
84                "Execution": {
```

```
 85              "DeviceType": "Nanodrop",
 86              "Config": {}
 87          },
 88          "Postcond": {
 89              "EmitArgNum": 1,
 90              "EmitArg": [
 91                  "Quantified Sample-1"
 92              ]
 93          }
 94      },
 95      {
 96          "Pred": "Quantify",
 97          "FlowUnit": {
 98              "Component": "Quantified Sample",
 99              "ComponentType": "Liquid",
100              "RefName": "Quantified Sample-1",
101              "UnitArgType": "PROD",
102              "Vol": "50 μL",
103              "Container": "0.2-ml PCR tube",
104              "Cond": {
105                  "State": "Quantified",
106                  "Concentration": "150 ng/μL",
107                  "A260/A280": 1.85,
108                  "A260/A230": 2.1
109              }
110          },
111          "Succ": "Dilute"
112      }
113 }
```

Part of protocol designed for Ecology:

```
 1 {
 2      {
 3          "Operation": "Grow",
 4          "Precond": {
 5              "SlotArgNum": 1,
 6              "SlotArg": [
 7                  "Watered_Rice_Plants-1"
 8              ]
 9          },
10          "Execution": {
11              "DeviceType": "Environmental growth chamber",
12              "Config": {
13                  "Temperature": "24 °C",
14                  "LightCycle": "12h light/12h dark"
15              }
16          },
17          "Postcond": {
18              "EmitArgNum": 1,
19              "EmitArg": [
```

```
20                    "Mature rice plants"
21                ]
22            }
23        },
24        {
25            "Pred": "Grow",
26            "FlowUnit": {
27                "Component": "Mature rice plants",
28                "ComponentType": "BiologicalMaterial",
29                "RefName": "Mature_Rice_Plants-1",
30                "UnitArgType": "PROD",
31                "Vol": "N/A",
32                "Container": "Plastic pot",
33                "Cond": {
34                    "State": "Mature",
35                    "Height": "50-60 cm"
36                }
37            },
38            "Succ": "Anesthetize"
39        },
40        {
41            "Operation": "Collect",
42            "Precond": {
43                "SlotArgNum": 2,
44                "SlotArg": [
45                    "Monitored_Aphid-1",
46                    "Mature_Rice_Plants-1"
47                ]
48            },
49            "Execution": {
50                "DeviceType": "Microcapillary tube",
51                "Config": {}
52            },
53            "Postcond": {
54                "EmitArgNum": 1,
55                "EmitArg": [
56                    "Phloem sap"
57                ]
58            }
59        },
60        {
61            "Pred": "Collect",
62            "FlowUnit": {
63                "Component": "Phloem sap",
64                "ComponentType": "Liquid",
65                "RefName": "Phloem_Sap-1",
66                "UnitArgType": "PROD",
67                "Vol": "1-2 μL",
68                "Container": "Microcapillary tube",
69                "Cond": {
```

```
70                    "State": "Collected",
71                    "Appearance": "Clear, slightly viscous"
72                }
73            },
74            "Succ": "Dilute"
75        },
76        {
77            "Operation": "Centrifuge",
78            "Precond": {
79                "SlotArgNum": 1,
80                "SlotArg": [
81                    "Diluted_Phl_Sap -1"
82                ]
83            },
84            "Execution": {
85                "DeviceType": "Centrifuge",
86                "Config": {
87                    "speed": "6000 rpm",
88                    "temperature": "4 °C",
89                    "time": "10 min"
90                }
91            },
92            "Postcond": {
93                "EmitArgNum": 1,
94                "EmitArg": [
95                    "Extracellular vesicles"
96                ]
97            }
98        },
99        {
100            "Pred": "Centrifuge",
101            "FlowUnit": {
102                "Component": "Extracellular vesicles",
103                "ComponentType": "Mixture",
104                "RefName": "EVs -1",
105                "UnitArgType": "PROD",
106                "Vol": "N/A",
107                "Container": "200-µl microtube",
108                "Cond": {
109                    "State": "Purified",
110                    "Appearance": "Small, almost invisible pellet"
111                }
112            },
113            "Succ": ""
114        }
115 }
```

