# OpenReview forum: "Hierarchically Encapsulated Representation for Protocol Design in Self-Driving Labs"
_ICLR.cc/2025/Conference — ICLR 2025 Poster_

### Official Review · Reviewer_Xeav · 2024-11-02

**Soundness:** 3
**Presentation:** 3
**Contribution:** 3
**Rating:** 8
**Confidence:** 4

**Summary:**

The work focuses on the automated design of self-driving labs' protocols. Protocols specify how robots in self-driving labs conduct experiments. In many cases, conservative protocols are not sufficient because they do not allow for significant scientific discovery. However, designing new protocols is a cumbersome and error-prone process. To tackle these challenges, the authors suggest LLMs for the generation of protocols.

**Strengths:**

- The problem is relevant and timely. Automation is much-needed and of high added value.

- The problem is open without sufficient solutions in the area. The approach seems to be novel and contributes to the state of the art.

- The contribution seems to be sound and the evaluation is convincing.

- Clear presentation and discussion.

**Weaknesses:**

- It is unclear why LLMs are chosen for this purpose. This is essentially a big design-space exploration problem with the added opportunity of using a physical actor in purposeful experimentation. A similarly apt approach could be using digital twins with traditional learning methods, e.g., reinforcement learning and obtain a formally modeled protocol.

- I find it somewhat concerning that a costly and potentially hazardous endeavor such as robotized experimentation with chemicals and biologically active materials is approached at the level of LLMs and the safety concerns are not discussed. For example, certification of protocols through formal verification might not be possible in this approach.

**Questions:**

- Why LLMs and not some other AI method? This would allow for a better positioning of the work.

- How can the generated protocols be certified? This would be a much-appreciated, brief discussion point.

---

> ### Author Response · Authors · 2024-11-25
> **Response to reviewer #Xeav - 1**
>
> > Why LLMs and not some other AI method? This would allow for a better positioning of the work.
>
> This is a very good question. The same consideration was evaluated during the decision-making process of designing our representation. It seems that we can formulate the protocol design problem in the fashion of Markov Decision Process (MDP) and solve it by heuristic-based planning methods or Hierarchical Reinforcement Learning (HRL) approaches. However, although the formulation itself is feasible, solving the problem may not be practical. Consider solving the problem through an HRL approach designed for heterogeneous action space with parameters (as the protocol is required to decide both the key properties of an operation and the corresponding values). This hierarchical agent may be trained to converge on a fine-grained environment with a clearly designed reward function, or on a large dataset with trajectories for offline learning. Unfortunately, we have access to neither an interactive environment simulating the experiments nor sufficient data to support offline training [1].
>
> Treating the experimental procedures as a white box and creating digital twins for experiments can be an elegant solution and thereby facilitate various applications other than protocol design. This effort requires elaborated design of simulation granularity, exhaustive collection of primitive principles of the system, efficient implementation of rule production, and define precise metrics for evaluating the distance between current and objective states (serving as a reward function), which can be labor-intensive and is far out of the scope of this work. On the other hand, viewing those published protocols as trajectories for offline training, the scale of the offline dataset and the density of the reward function are much too insufficient to support training to convergence. Augmenting the data, synthesizing realistic trajectories, or enhancing the accessibility of protocols, are out of the scope of this work. Given the current obstacles, we choose not to formulate the problem in an MDP fashion. Though an MDP-style formulation can be more precise and elegant, it may misguide the readers to some extent. Instead, we decide to leverage the rich domain-specific knowledge provided by knowledge-based agents such as Large Language Models (LLMs), where knowledge may complement the lack of data and dense reward function. This design choice is also in line with the initial attempts on automatic experiment design [2, 3].
>
> In summary, our design choice of formulation is a compromise based on currently limited resources and restricted scope. Nonetheless, the exploration of more precise and elegant formulations represents a promising avenue for future research, and we appreciate the reviewer's insightful suggestion in this regard. We have made the revisions to convey these insights.
>
> References:
>
> [1] Pateria, S., Subagdja, B., Tan, A. H., & Quek, C. (2021). Hierarchical reinforcement learning: A comprehensive survey. ACM Computing Surveys (CSUR), 54(5), 1-35.
>
> [2] Boiko, D. A., MacKnight, R., Kline, B., & Gomes, G. (2023). Autonomous chemical research with large language models. Nature, 624(7992), 570-578.
>
> [3] M. Bran, A., Cox, S., Schilter, O.,  Baldassari, C., White, A. D., & Schwaller, P. (2024). Augmenting  large language models with chemistry tools. Nature Machine Intelligence, 1-11.

---

> > ### Comment · Reviewer_Xeav · 2024-11-28
> >
> > Thank you, this part of the response clarifies things:
> > > However, although the formulation itself is feasible, solving the problem may not be practical.
> >
> > I recommend explaining the design choice in the camera-ready version of the paper.

---

> > > ### Author Response · Authors · 2024-11-28
> > >
> > > Thank you for your suggestion. We have incorporated the discussion on design choice into the 'Additional Remarks' section. We are continuously working to further elaborate on this part.

---

> ### Author Response · Authors · 2024-11-25
> **Response to reviewer #Xeav - 2**
>
> > How can the generated protocols be certified? This would be a much-appreciated, brief discussion point.
>
> This is a very good question. Certification is always one of the central focuses in the engineering practices of automation. In our practice, we only automate the process of protocol design, which is the primary objective of this work, and keep the manual certification part. On one hand, relieving experimental scientists from the labour-intensive protocol design tasks, thereby allowing them more time for high-level thinking, is a sufficiently significant improvement so far. On the other hand, engineering practices such as lab automation and manufacturing are in high demand for preciseness. This leads to the requirement of manual certification. Domain experts handle subtle cases through their tacit domain-specific knowledge and are responsible for their decisions [1]. According to these considerations and the standard operating processes of experimental sciences, we choose to certify the designed protocols by domain experts.
>
> Our current choice is a compromise on the limitation of techniques and the demand for preciseness. In future work, we can conduct investigations on how to build digital twins of self-driving laboratories. Such digital twins support prediction, explanation, and counterfactual analysis of unseen behaviors of the experiments, which may facilitate machine-based protocol certification. Grounding these blue-sky thoughts necessitates addressing the challenging problems regarding the decision of simulation granularity, the implementation of data-efficient simulation model construction, and the injection of tacit domain-specific knowledge. In summary, the exploration of generated-protocol-certification by machines represents a promising avenue for future research, and we appreciate the reviewer's insightful suggestion in this regard. We have extended the discussion accordingly.
>
> References:
>
> [1] Wang, H., Fu, T., Du, Y., Gao, W.,  Huang, K., Liu, Z., ... & Zitnik, M. (2023). Scientific discovery in  the age of artificial intelligence. Nature, 620(7972), 47-60.

---

> > ### Comment · Reviewer_Xeav · 2024-11-28
> >
> > Thank you for the clarification. I recommend elaborating on this aspect in the discussion of the work in the camera-ready version.

---

> > > ### Author Response · Authors · 2024-11-28
> > >
> > > Thank you for your suggestion. We have incorporated the discussion on protocol certification into the 'Additional Remarks' section. We are continuously working to further elaborate on this part.

---

> ### Author Response · Authors · 2024-11-25
> **Response to reviewer #Xeav - 3**
>
> > It is unclear why LLMs are chosen for this purpose. This is essentially a big design-space exploration problem with the added opportunity of using a physical actor in purposeful experimentation. A similarly apt approach could be using digital twins with traditional learning methods, e.g., reinforcement learning and obtain a formally modeled protocol.
>
> This is a very good question. The same consideration was evaluated during the decision-making process of designing our representation. It seems that we can formulate the protocol design problem in the fashion of Markov Decision Process (MDP) and solve it by heuristic-based planning methods or Hierarchical Reinforcement Learning (HRL) approaches. However, although the formulation itself is feasible, solving the problem may not be practical. Consider solving the problem through an HRL approach designed for heterogeneous action space with parameters (as the protocol is required to decide both the key properties of an operation and the corresponding values). This hierarchical agent may be trained to converge on a fine-grained environment with a clearly designed reward function, or on a large dataset with trajectories for offline learning. Unfortunately, we have access to neither an interactive environment simulating the experiments nor sufficient data to support offline training [1].
>
> Treating the experimental procedures as a white box and creating digital twins for experiments can be an elegant solution and thereby facilitate various applications other than protocol design. This effort requires elaborated design of simulation granularity, exhaustive collection of primitive principles of the system, efficient implementation of rule production, and define precise metrics for evaluating the distance between current and objective states (serving as a reward function), which can be labor-intensive and is far out of the scope of this work. On the other hand, viewing those published protocols as trajectories for offline training, the scale of the offline dataset and the density of the reward function are much too insufficient to support training to convergence. Augmenting the data, synthesizing realistic trajectories, or enhancing the accessibility of protocols, are out of the scope of this work. Given the current obstacles, we choose not to formulate the problem in an MDP fashion. Though an MDP-style formulation can be more precise and elegant, it may misguide the readers to some extent. Instead, we decide to leverage the rich domain-specific knowledge provided by knowledge-based agents such as Large Language Models (LLMs), where knowledge may complement the lack of data and dense reward function. This design choice is also in line with the initial attempts on automatic experiment design [2, 3].
>
> In summary, our design choice of formulation is a compromise based on currently limited resources and restricted scope. Nonetheless, the exploration of more precise and elegant formulations represents a promising avenue for future research, and we appreciate the reviewer's insightful suggestion in this regard. We have made the revisions to convey these insights.
>
> References:
>
> [1] Pateria, S., Subagdja, B., Tan, A. H., & Quek, C. (2021). Hierarchical reinforcement learning: A comprehensive survey. ACM Computing Surveys (CSUR), 54(5), 1-35.
>
> [2] Boiko, D. A., MacKnight, R., Kline, B., & Gomes, G. (2023). Autonomous chemical research with large language models. Nature, 624(7992), 570-578.
>
> [3] M. Bran, A., Cox, S., Schilter, O.,  Baldassari, C., White, A. D., & Schwaller, P. (2024). Augmenting  large language models with chemistry tools. Nature Machine Intelligence, 1-11.

---

> ### Author Response · Authors · 2024-11-25
> **Response to reviewer #Xeav - 4**
>
> > I find it somewhat concerning that a costly and potentially hazardous endeavor such as robotized experimentation with chemicals and biologically active materials is approached at the level of LLMs and the safety concerns are not discussed. For example, certification of protocols through formal verification might not be possible in this approach.
>
> Thanks for the insightful comment. We totally agree with the reviewer, in particular on the observation that LLMs can be much too uncontrollable for engineering practices such as lab automation, which may lead to unpredictable dangerous situations [1]. There comes a dilemma --- we try to exploit the capability of reasoning over knowledge of LLMs, while we try to alleviate the drawbacks brought up by the uncontrollable nature of LLMs. Our proposed representation is dedicated to resolving the dilemma. The representations not only elicit LLMs' potential on protocol design through structural knowledge representation, but also serve as a guardrail for LLMs. Since the generated protocols are represented as corresponding DSL programs, the permissible output space is much more confined compared with that of pure LLMs, serving as constraints upon the LLM-generated protocols. Thanks to the verification mechanisms provided by DSLs, the correctness of the generated protocols can be checked to some extent. Therefore, by equipping LLMs with an auxiliary module of the constraints, we may approach a balance between knowledge utilization and preciseness.
>
> However, the current verification on the level of DSL programs is far from sufficient for serving as a certification. Certification is a serious process, where any possibilities of reporting false positive cases are required to be eliminated. Some cases can be highly long-tailed distributed, which may not be detected by data-driven and knowledge-driven machine certifiers. In this context, human domain experts are responsible for coming up with these potential risks through their experiences and tacit knowledge. Therefore, we are not likely to move human experts out of the loop, except that we can efficiently build up appropriate digital twins for self-driving laboratories. In current practices, the automation of protocol design puts human experts into a larger loop without focusing on the low-level details of experiments. As a result, they are alloweds more time for high-level thoughts on things like values, which are not likely to be alternated by machines. In summary, it is neither practical nor necessary to totally move human experts out of the loop of automatic scientific discovery. The investigation of human-machine coordination in protocol certification represents a promising avenue for future research, and we appreciate the reviewer's insightful suggestion in this regard. We have extended the discussion accordingly.
>
> References:
>
> [1] Zhang, Y., Li, Y., Cui, L., Cai, D.,  Liu, L., Fu, T., ... & Shi, S. (2023). Siren's song in the AI ocean: a survey on hallucination in large language models. arXiv preprint arXiv:2309.01219.

---

### Official Review · Reviewer_LMMv · 2024-11-03

**Soundness:** 3
**Presentation:** 2
**Contribution:** 3
**Rating:** 6
**Confidence:** 2

**Summary:**

The authors propose a hierarchical structured representation of experiment protocols to improve the reliability of novel protocols generated by LLMs with RAG.
* The representation is actually a dual representation over operation-centric and flow-centric views of a protocol.
* This works because the representation serves as a domain-specific language to generate protocols in a bipartite graph-like structure, which enables verification of both the "nodes" and "edges" of the protocol, improving its reliability.
* However, to use this structured representation requires extracting the entities, which is domain-specific. To do this, they use a Dirichlet Process Mixture Model to extract entities from natural language protocols, which they then aggregate for functions but not for components.
* To demonstrate the improved viability of protocols generated by this approach, they subselect a test set of 140 protocols from their database with the intent of evaluating across 3 different protocol design tasks and 4 domains. They evaluate generated protocols against the ground truth using six distinct metrics. They compare seven different design approaches which consider different elements of their proposed representation, with and without verification, showing how the different aspects of their approach contribute to its success.

**Strengths:**

The authors make three contributions:
1. They present the protocol design problem setting, as well as the hierarchical representation of protocols.
2. The propose a method to automatically determine the elements of this structured representation from existing protocols.
3. They perform experiments assessing the value of their representation by testing different methods based on this representation on a varied set of tasks.

# originality

The proposed representation and method appear quite novel, and the choice of metrics to assess the protocol designers is interesting. However, there is no mention of the extensive literature on problem decomposition for LLMs, e.g. Chain-of-Thought reasoning (Wei et al. (2023)). This makes it difficult to ascertain exactly how novel such a decomposition really is. it would be helpful to compare and contrast to existing methods in the literature both for the applied problem of protocol design (e.g. O’Donoghue et al. (2023)) and the solution of task decomposition (e.g. Wei et al. (2023)). It would also be helpful to explicitly state in the contributions whether the is a new problem formulation, application of an existing method to a new domain, or both.

# quality

The conclusions drawn in section 4.5 are the following:
1. Their proposed approaches outperform naive counterparts.
2. Dual-view hierarchical representations excel at protocol design tasks.
3. The approach generalizes across scientific domains.

The first claim is sound considering the extensive metrics used to assess different approaches. However, it is also shallow compared to the amount of information in Figure 3. On average, it seems that the approach put forward performs well, but there is significantly more variability in their performance. Furthermore, different metrics seem to behave different to the new metrics, with IoU metrics increasing the tail of good results (e.g. IoU metrics), and Sim metrics decreasing the tail of bad results. It is surprising that this is not discussed in more detail, and that there is no estimator of uncertainty (e.g. std, stderr, variance) in the tables in Appendix A.

The second claim is hard to interpret, as there are no baselines from the literature that are evaluated. As such, it is unclear what is meant by "excel" in this case. Given the highly variable performance values in Figure 3 and the lack of baselines from the literature, it is not possible to conclude this from the available results. You could alternatively state that the dual-view approaches outperform their counterparts across all tasks on average across all metrics.

The third claim is well supported by the experiments across four domains and the aggregate results. That being said, an additional set of results in the appendix that shows results by scientific domain would also be helpful in supporting this claim, especially if these results were to show that the proposed methods performed best across all domains separately (i.e., tables like those in appendix A, but partitioned by domain instead of task).

# clarity

As a non-expert on protocol design, I found the problem motivation well done. The task is well situated within its broader context. Figure 1A is enlightening, but Figure 1B is slightly confusing, as the relationships between the different elements could be made clearer. The problem setting is well presented with a judicious use of notation and examples throughout. The experiments are well explained as well.

# significance

The motivation put forward in the introduction accurately captures the significance of this work. However, it could be improved even more by relating it to existing work that creates structured workflows using LLMs, and relating it to the more general problem of getting LLMs to structure their outputs.

**Weaknesses:**

# Presentation

* Figure 3 Caption is not standalone, very small font sizes for xticks and y ticks. The x-axis is not mentioned, which makes interpretation impossible.

* Figure 2 labels and ticks are too small, the caption is largely uninformative without the main text

* Fig 1 B is unclear

* Limitations section is empty (Appendix G)

# Soundness
* Line 361: A reference to Fig 2B is insufficient for the claim that function abstraction works. Idem for Fig 2C. The results must be interpreted (see the "Quality" section in strengths for more details).

* The testing set is susceptible for memorization because you are using LLMs. It would be good to have a set of protocols that are after the cutoff dates of the LLM training.

* it would be good to introduce the SA algorithm in more detail so that we understand what the similarity score actually represents.

* Lack of baselines from the literature

# Contribution
* Very limited discussion of the results.

* Unclear whether they consider the problem setting to be a contribution. It would be important to contrast the problem formulation with that of prior work, such as that of O’Donoghue et al. (2023). In particular, whether their formulation of the protocol design problem (not the representation itself) differs in any meaningful way. This is important because the identification of the protocol design problem is cited as a contribution on line 147.

**Questions:**

* Line 44: What is a "protocol portal"? Such terminology, if not generally entrenched in the suggested domain, should be introduced.

* Line 199: "The total amount of the instance actions can be extremely high, i.e., about 150K per domain," how do you arrive at this number?

* How do you track quantities of reagents? The algorithm seems to treat them as a set and remove them if used once without any notion of quantity.

* Line 504: what it t( ) ?

---

> ### Author Response · Authors · 2024-11-25
> **Response to reviewer #LMMv - 1**
>
> > Line 44: What is a "protocol portal"? Such terminology, if not generally entrenched in the suggested domain, should be introduced.
>
> Thanks for pointing this out. The term "protocol portal" refers to the open-sourced websites run by top-tier publishers, including Nature's [Protocolexchange](https://protocolexchange.researchsquare.com/), Cell's [Star-protocols](https://star-protocols.cell.com/), [Bio-protocol](https://bio-protocol.org/en), Wiley's [Current Protocols](https://currentprotocols.onlinelibrary.wiley.com/), and [Jove](https://www.jove.com/). As these websites are essentially protocol databases, we changed the expression to "protocol databases", which seems to be more common, to enhance the clarity of the paper.
>
> > Line 199: "The total amount of the instance actions can be extremely high, i.e., about 150K per domain," how do you arrive at this number?
>
> Thanks for the question. We randomly sampled approximately 10% of the protocols from the large corpus in each domain and used the method of BioPlanner to generate corresponding pseudofunctions and pseudocode [1]. Each unique function name represents an instance action. We then counted the number of instance actions in the sampled subset and extrapolated the total number of instance actions for each domain within the large corpus. Since the corpus we used encompasses the majority of protocols in each domain, these estimates are considered to be fairly accurate. We have made the revisions to clarify this point.
>
> References:
>
> [1] O’Donoghue, O., Shtedritski, A.,  Ginger, J., Abboud, R., Ghareeb, A., & Rodriques, S. (2023,  December). BioPlanner: Automatic Evaluation of LLMs on Protocol Planning  in Biology. In Proceedings of the 2023 Conference on Empirical Methods in Natural Language Processing (pp. 2676-2694).
>
> > How do you track quantities of reagents? The algorithm seems to treat them as a set and remove them if used once without any notion of quantity.
>
> This is an insightful question. The same concern was considered in the decision-making process of formulating the desired representation. The rationale for such design choice comes from both the current Standard Operating Processes (SOP) of experimental sciences and the properties of self-driving laboratories [1].
>
> In the current SOP for manually conducted experiments, experimenters are required to use prefabricated sets of reagents. Similarly, experimenters use specific containers with predefined capacities to transfer intermediate products. Therefore, one pack of reagents or one container of intermediate products is only used once for an operation, without considering the remainder. This results in a more succinct representation where reagents are regarded as discrete elements rather than continuous volumes.
>
> For self-driving laboratories, this is deliberately designed for efficient variable management following the corresponding principles in computer system design [2]. In computer systems, not removing used variables would cause out-of-memory errors, let alone in physical automation systems, where the physical memory slots are much harder than the virtual memory slots in computer systems to manage. Hence, we exploit this variable management mechanism to enhance the execution efficiency of self-driving laboratories.
>
> We appreciate the reviewer for pointing this out. We have revised the paper to enhance the clarity of this design choice.
>
> References:
>
> [1] Bartley, B., Beal, J., Rogers, M.,  Bryce, D., Goldman, R. P., Keller, B., ... & Weston, M. (2023).  Building an open representation for biological protocols. ACM Journal on Emerging Technologies in Computing Systems, 19(3), 1-21.
>
> [2] Abelson, H., & Sussman, G. J. (1996). Structure and interpretation of computer programs (p. 688). The MIT Press.

---

> ### Author Response · Authors · 2024-11-25
> **Response to reviewer #LMMv - 2**
>
> > Line 504: what it t( ) ?
>
> Thanks for the question. In our study, we used a statistical method called the paired t-test, represented by the symbol $t()$, to determine whether there is a significant difference in performance between machine designers using our proposed representation (EE+ and EI+) and the baseline methods. This test helps us verify our hypothesis that our proposed methods significantly outperform the baseline methods.
>
> The formulation $t(\cdot)=\dots, μ_d<0, p<.01$ in our results indicates the outcome of the t-test. Here, $μ_d$ represents the average difference in performance between the two groups being compared. If $μ_d$ is less than zero, it suggests that the performances of the proposed methods are generally better than the performances of the baselines. However, the crucial part of this result is the p-value, represented by $p$. The p-value is a statistical measure that helps us determine the probability of observing the results we did if the null hypothesis were true. The null hypothesis in this context is that there is no difference in performance between the two methods.
>
> As our results show that the p-value is less than 0.01, it indicates that there is less than a 1% chance that the observed difference in performance could occur if there were actually no differences (i.e., if the null hypothesis were true). This low probability leads us to reject the null hypothesis, thus concluding that the difference in performance is statistically significant. This means we have sufficient evidence to support our claim that the proposed representations (EE+ and EI+) outperform the baselines in a meaningful way. We have made the revisions to clarify this concept.
>
> References:
>
> [1] Student. (1908). The probable error of a mean. Biometrika, 1-25.
>
> > Line 361: A reference to Fig 2B is insufficient for the claim that function abstraction works. Idem for Fig 2C. The results must be interpreted (see the "Quality" section in strengths for more details).
>
> Thanks for the suggestion. We have extended both the caption of Fig 2 and the corresponding text for reference to Fig 2.

---

> ### Author Response · Authors · 2024-11-25
> **Response to reviewer #LMMv - 3**
>
> > The testing set is susceptible for memorization because you are using LLMs. It would be good to have a set of protocols that are after the cutoff dates of the LLM training.
>
> Thanks for the comment. The same concern was considered during the development of these machine designers. We made the design choice from the following two aspects. On one hand, the benchmark used for evaluation, namely, the groundtruth DSL programs as generated protocols are created and verified by our team of domain experts. This indicates that the benchmark has never been publicly released at the time LLMs were collecting training data. On the other hand, although the high-level experimental objectives can be duplicated on the Internet, the performance of LLM-based protocol designer evaluated in previous work reveals that LLMs cannot exploit such unstructured knowledge to generate fine-grained experimental procedures [1].
>
> We also employ the broadly accepted standard operating process to empirically verify that LLMs did not memorize the data. We adopted the methodology outlined in Section 5.2 of *Skywork*  [2] and drew upon recent studies on detecting memorization in large language models (LLMs) [3, 4]. Specifically, we used gpt-4o mini to synthesize data resembling the style of steps from novel protocols, and then calculated the perplexity on the test set and reference set. Since the reference set was newly generated, we considered it clean, not belonging to any training set of any model.
>
> We randomly sampled 100 sequences each from the test set and the reference set of the novel protocols. Each sequence corresponds to a single procedural step described in natural language. We truncated the final 50 tokens of each sequence, retaining the prefixes. These prefixes were then used as prompts for the LLM to predict the next 50 tokens, for which we calculated the perplexity. If the test set’s perplexity is significantly lower than the reference set’s, the test set might have appeared in the model’s training phase.
>
> The perplexity results for the test set and reference set are as follows:
> |               | average_ppl (std, var, stderr) |
> | ------------- | ------------------------------ |
> | test set      | 1.366 (0.123, 0.015, 0.012)    |
> | reference set | 1.315 (0.116, 0.014, 0.012)    |
>
> [Comparison between the perplexity of the test set and the reference set](https://anonymous.4open.science/api/repo/AutoDSL-Planning-Figure-0DFE/file/ppl.png?v=2263a5a4)
>
> The results indicate that the LLM’s average perplexity on the test set is significantly higher than that on the reference set ($t(198)=3.040, \mu_d<0, p<.05$; see the figure above), suggesting that the LLM encounters greater uncertainty with the novel protocols in the test set. This finding implies that for a published, widely accepted, and standardized operating process, there is no evidence to suggest that the LLM has memorized the data.
>
> References:
>
> [1] O’Donoghue, O., Shtedritski, A.,  Ginger, J., Abboud, R., Ghareeb, A., & Rodriques, S. (2023,  December). BioPlanner: Automatic Evaluation of LLMs on Protocol Planning in Biology. In Proceedings of the 2023 Conference on Empirical Methods in Natural Language Processing (pp. 2676-2694).
>
> [2] Wei T, Zhao L, Zhang L, et al. Skywork: A more open bilingual foundation model. arXiv preprint arXiv:2310.19341, 2023.
>
> [3] Carlini N, Ippolito D, Jagielski M, et al. Quantifying memorization across neural language models. arXiv preprint arXiv:2202.07646, 2022.
>
> [4] Carlini N, Tramer F, Wallace E, et al. Extracting training data from large language models//30th USENIX Security Symposium (USENIX Security 21). 2021: 2633-2650.

---

> ### Author Response · Authors · 2024-11-25
> **Response to reviewer #LMMv - 4**
>
> > it would be good to introduce the SA algorithm in more detail so that we understand what the similarity score actually represents.
>
> Thanks for the suggestion. We treat the operation execution order of each protocol as an ordered sequence of varying lengths. The SA metric evaluates the similarity between two such sequences, accounting for both the similarity of individual operations and the consistency of their execution order. We have made the revisions accordingly.
>
> [Algorithm for Sequence Alignment (SA) metric](https://anonymous.4open.science/api/repo/AutoDSL-Planning-Figure-0DFE/file/SA.png?v=d162a8df)
>
> [Needleman-Wunsch Algorithm](https://anonymous.4open.science/api/repo/AutoDSL-Planning-Figure-0DFE/file/Needleman-Wunsch.png?v=db1c2685)
>
> [Smith-Waterman Algorithm](https://anonymous.4open.science/api/repo/AutoDSL-Planning-Figure-0DFE/file/Smith-Waterman.png?v=dbeab0fb)
>
> | GroundTruth                                       | EE+                                                          | EE                                                           | II                                                           | IB                                                           | FB                                                           |
> | ------------------------------------------------- | ------------------------------------------------------------ | ------------------------------------------------------------ | ------------------------------------------------------------ | ------------------------------------------------------------ | ------------------------------------------------------------ |
> | ["Dissolve", "Add", "Add", "Incubate", "Analyze"] | ["Dissolve", "Add", "Add", "Incubate", "Analyze"] **SA = 1.0** | ["Dissolve", "Add", "Incubate", "Add", "Dissolve", "Analyze"] **SA = 0.75** | ["Add", "Incubate", "Collect", "Analyze"] **SA = 0.57**      | ["Incubate", "Lyse", "Add", "Run", "Wash", "Extract", "Incubate", "Perform", "Detect", "Analyze"] **SA = 0.34** | ["Prepare", "Add", "Terminate", "Cool", "Purify", "Wash", "Elute", "Concentrate", "Prepare", "Dissolve", "Setup", "Inject", "Perform", "Analyse", "Compile"] **SA = 0.23** |
> | ["Discharge", "Place", "Sit", "Stain"]            | ["Make", "Place", "Let", "Stain"] **SA = 0.63**              | ["Resuspend", "Place", "Place", "Let", "Stain", "Let", "Dispose"] **SA = 0.47** | ["Prepare", "Glow", "Apply", "Wick", "Adsorb", "Allow", "Stain", "Image"] **SA = 0.25** | ["Cut", "Dip", "Prepare", "Mix", "Allow", "Transfer", "Collect", "Post", "Stain", "Image"] **SA = 0.22** | ["Prepare", "Place", "Dilute", "Pipette", "Incubate", "Remove", "Rinse", "Fix", "Remove", "Wash", "Strain", "Remove", "Rinse", "Complete", "Observe"] **SA = 0.15** |
>
> > Lack of baselines from the literature
>
> Thanks for the question. Automating the design of experiments is a relatively new domain, which was initially introduced by recent works in 2023 [1, 2]. In the previous literature search, we only find the current state-of-the-art work BioPlanner [3], which explicitizes the originally implicit experiment design process in previous works [1, 2]. As we have mentioned in the paper, our baselines are developed based on the methods proposed by these previous works. The Instance-Internal (II) designer is developed based on the state-of-the-art method of BioPlanner. The Flatten-Baseline (FB) and Instance-Baseline (IB) designers are developed based on the baselines being evaluated in [3].
>
> We appreciate the reviewer for pointing this out. We have revised the paper to enhance the links between the introduction of these baseline methods in the subsection "Machine designers" and our citations of these previous works in the section "Introduction".
>
> References:
>
> [1] Boiko, D. A., MacKnight, R., Kline, B., & Gomes, G. (2023). Autonomous chemical research with large language models. Nature, 624(7992), 570-578.
>
> [2] M. Bran, A., Cox, S., Schilter, O.,  Baldassari, C., White, A. D., & Schwaller, P. (2024). Augmenting  large language models with chemistry tools. Nature Machine Intelligence, 1-11.
>
> [3] O’Donoghue, O., Shtedritski, A.,  Ginger, J., Abboud, R., Ghareeb, A., & Rodriques, S. (2023,  December). BioPlanner: Automatic Evaluation of LLMs on Protocol Planning  in Biology. In Proceedings of the 2023 Conference on Empirical Methods in Natural Language Processing (pp. 2676-2694).
>
> > Figure 3 Caption is not standalone, very small font sizes for xticks and y ticks. The x-axis is not mentioned, which makes interpretation impossible.
>
> Thanks for pointing this out. We have made the revisions accordingly to enhance the readability of the figure.
>
> > Figure 2 labels and ticks are too small, the caption is largely uninformative without the main text
>
> Thanks for pointing this out. We have made the revisions accordingly to enhance the readability of the figure.
>
> > Fig 1 B is unclear
>
> Thanks for pointing this out. We have made the revisions accordingly to enhance the readability of the figure.

---

> ### Author Response · Authors · 2024-11-25
> **Response to reviewer #LMMv - 5**
>
> > Limitations section is empty (Appendix G)
>
> Thanks for pointing this out. Appendix G was accidentally removed from the version of the submission due to version control issues. We have recovered it in the revised version.
>
> > However, there is no mention of the extensive literature on problem decomposition for LLMs, e.g. Chain-of-Thought reasoning (Wei et al. (2023)). This makes it difficult to ascertain exactly how novel such a decomposition really is. it would be helpful to compare and contrast to existing methods in the literature both for the applied problem of protocol design (e.g. O’Donoghue et al. (2023)) and the solution of task decomposition (e.g. Wei et al. (2023)). It would also be helpful to explicitly state in the contributions whether the is a new problem formulation, application of an existing method to a new domain, or both.
>
> Thanks for the comment. We would like to clarify that our objective is not to alternate Chain-of-Thought (CoT) reasoning. According to recent studies on the properties of CoT, LLMs with CoT may generate coherent but unprofessional text in expertise-intensive application scenarios [1]. Therefore, our proposed representation serves as an auxiliary guardrail module for LLMs with reasoning techniques such as CoT, enhancing LLMs' reasoning capability from two aspects: (i) the representation constrain the scope of reasoning into a close set of entities, such as available operations, reagents, and devices commonly used in the domain; and (ii) the representation provides fine-grained injection of domain-specific knowledge for LLMs, resulting in not only coherent but also professionality-compatible generated content.
>
> One of our compared baseline approaches, II, is originated from the BioPlanner approach [2], which is actually implemented based on CoT. BioPlanner equips CoT with a relatively naive representation, namely, the instance actions with attributes described in Sec 2.2 of our paper. Results show that our approach significantly outperforms the approaches with CoT and the instance actions with attributes representation (EE+ vs. II: $t(278) = 24.493, \mu_d < 0, p < .0001 $; EI+ vs. II: $t(278) = 23.855, \mu_d < 0, p < .0001 $; also see the figures below), demonstrating the capability of our proposed representation. Also, this implies that representation may be dominant for LLMs' performances in this task. We have made the revisions to clarify this point.
>
> [Comparison between the capabilities of our approach and II across the six dimensions](https://anonymous.4open.science/api/repo/AutoDSL-Planning-Figure-0DFE/file/radar_1.png?v=928256b1)
>
> References:
>
> [1] Xiao, Z., Zhang, D., Wu, Y., Xu, L.,  Wang, Y. J., Han, X., ... & Chen, G. (2023). Chain-of-Experts: When  LLMs Meet Complex Operations Research Problems. In The Twelfth International Conference on Learning Representations.
>
> [2] O’Donoghue, O., Shtedritski, A.,  Ginger, J., Abboud, R., Ghareeb, A., & Rodriques, S. (2023,  December). BioPlanner: Automatic Evaluation of LLMs on Protocol Planning  in Biology. In Proceedings of the 2023 Conference on Empirical Methods in Natural Language Processing (pp. 2676-2694).

---

> ### Author Response · Authors · 2024-11-25
> **Response to reviewer #LMMv - 6.1**
>
> > However, it is also shallow compared to the amount of information in Figure 3. On average, it seems that the approach put forward performs well, but there is significantly more variability in their performance. Furthermore, different metrics seem to behave different to the new metrics, with IoU metrics increasing the tail of good results (e.g. IoU metrics), and Sim metrics decreasing the tail of bad results. It is surprising that this is not discussed in more detail, and that there is no estimator of uncertainty (e.g. std, stderr, variance) in the tables in Appendix A.
>
> Thanks for the comment. Though we didn't show the results of uncertainty, we performed significance tests on the effects between methods, which is shown in Figure 3 (C-E), the bar plots with error bars.
>
> **planning**
> | Method | IoU(Op) mean(std, var, stderr) | IoU(Prod)                   | IoU(Dev)                    | Sim(Exec)                   | Sim(Goal)                   | Sim(Param)                  |
> | ------ | ------------------------------ | --------------------------- | --------------------------- | --------------------------- | --------------------------- | --------------------------- |
> | FB     | 0.143 (0.087, 0.008, 0.016)    | 0.04 (0.055, 0.003, 0.01)   | 0.02 (0.06, 0.004, 0.011)   | 0.282 (0.09, 0.008, 0.016)  | 0.766 (0.096, 0.009, 0.017) | 0.826 (0.059, 0.004, 0.011) |
> | IB     | 0.109 (0.067, 0.004, 0.012)    | 0.036 (0.053, 0.003, 0.01)  | 0.019 (0.074, 0.005, 0.013) | 0.242 (0.065, 0.004, 0.012) | 0.735 (0.09, 0.008, 0.016)  | 0.781 (0.069, 0.005, 0.012) |
> | II     | 0.382 (0.154, 0.024, 0.028)    | 0.05 (0.062, 0.004, 0.011)  | 0.084 (0.191, 0.037, 0.034) | 0.452 (0.134, 0.018, 0.024) | 0.788 (0.074, 0.006, 0.013) | 0.851 (0.062, 0.004, 0.011) |
> | EI     | 0.542 (0.16, 0.026, 0.029)     | 0.305 (0.181, 0.033, 0.032) | 0.259 (0.211, 0.045, 0.038) | 0.572 (0.152, 0.023, 0.027) | 0.849 (0.066, 0.004, 0.012) | 0.926 (0.026, 0.001, 0.005) |
> | EI+    | 0.603 (0.208, 0.043, 0.037)    | 0.555 (0.26, 0.068, 0.047)  | 0.357 (0.237, 0.056, 0.043) | 0.737 (0.172, 0.03, 0.031)  | 0.875 (0.057, 0.003, 0.01)  | 0.949 (0.023, 0.001, 0.004) |
> | EE     | 0.524 (0.151, 0.023, 0.027)    | 0.37 (0.198, 0.039, 0.036)  | 0.252 (0.206, 0.043, 0.037) | 0.558 (0.148, 0.022, 0.027) | 0.846 (0.078, 0.006, 0.014) | 0.928 (0.025, 0.001, 0.004) |
> | EE+    | 0.607 (0.211, 0.044, 0.038)    | 0.605 (0.235, 0.055, 0.042) | 0.355 (0.242, 0.059, 0.044) | 0.744 (0.179, 0.032, 0.032) | 0.893 (0.056, 0.003, 0.01)  | 0.951 (0.021, 0.0, 0.004)   |
>
> **modification**
> | Method | IoU(Op) mean(std, var, stderr) | IoU(Prod)                   | IoU(Dev)                    | Sim(Exec)                   | Sim(Goal)                   | Sim(Param)                  |
> | ------ | ------------------------------ | --------------------------- | --------------------------- | --------------------------- | --------------------------- | --------------------------- |
> | FB     | 0.181 (0.102, 0.01, 0.012)     | 0.05 (0.071, 0.005, 0.008)  | 0.038 (0.071, 0.005, 0.008) | 0.304 (0.102, 0.01, 0.012)  | 0.796 (0.09, 0.008, 0.01)   | 0.809 (0.06, 0.004, 0.007)  |
> | IB     | 0.15 (0.1, 0.01, 0.012)        | 0.038 (0.065, 0.004, 0.008) | 0.039 (0.076, 0.006, 0.009) | 0.281 (0.1, 0.01, 0.012)    | 0.771 (0.089, 0.008, 0.01)  | 0.788 (0.06, 0.004, 0.007)  |
> | II     | 0.331 (0.143, 0.021, 0.017)    | 0.101 (0.131, 0.017, 0.015) | 0.061 (0.135, 0.018, 0.016) | 0.416 (0.127, 0.016, 0.015) | 0.802 (0.087, 0.008, 0.01)  | 0.851 (0.059, 0.004, 0.007) |
> | EI     | 0.593 (0.186, 0.035, 0.022)    | 0.318 (0.158, 0.025, 0.018) | 0.336 (0.235, 0.055, 0.027) | 0.602 (0.164, 0.027, 0.019) | 0.866 (0.066, 0.004, 0.008) | 0.937 (0.03, 0.001, 0.003)  |
> | EI+    | 0.648 (0.21, 0.044, 0.024)     | 0.626 (0.188, 0.035, 0.022) | 0.413 (0.256, 0.065, 0.03)  | 0.765 (0.17, 0.029, 0.02)   | 0.883 (0.055, 0.003, 0.006) | 0.952 (0.031, 0.001, 0.004) |
> | EE     | 0.588 (0.185, 0.034, 0.022)    | 0.403 (0.192, 0.037, 0.022) | 0.332 (0.228, 0.052, 0.027) | 0.601 (0.164, 0.027, 0.019) | 0.873 (0.053, 0.003, 0.006) | 0.94 (0.028, 0.001, 0.003)  |
> | EE+    | 0.64 (0.213, 0.045, 0.025)     | 0.661 (0.179, 0.032, 0.021) | 0.41 (0.253, 0.064, 0.029)  | 0.757 (0.17, 0.029, 0.02)   | 0.893 (0.043, 0.002, 0.005) | 0.953 (0.032, 0.001, 0.004) |

---

> ### Author Response · Authors · 2024-11-25
> **Response to reviewer #LMMv - 6.2**
>
> **adjustment**
> | Method | IoU(Op) mean(std, var, stderr) | IoU(Prod)                   | IoU(Dev)                    | Sim(Exec)                   | Sim(Goal)                   | Sim(Param)                  |
> | ------ | ------------------------------ | --------------------------- | --------------------------- | --------------------------- | --------------------------- | --------------------------- |
> | FB     | 0.192 (0.1, 0.01, 0.017)       | 0.077 (0.104, 0.011, 0.018) | 0.051 (0.094, 0.009, 0.016) | 0.319 (0.103, 0.011, 0.017) | 0.811 (0.078, 0.006, 0.013) | 0.823 (0.051, 0.003, 0.009) |
> | IB     | 0.197 (0.131, 0.017, 0.022)    | 0.039 (0.063, 0.004, 0.011) | 0.006 (0.021, 0.0, 0.004)   | 0.337 (0.141, 0.02, 0.024)  | 0.802 (0.082, 0.007, 0.014) | 0.81 (0.049, 0.002, 0.008)  |
> | II     | 0.453 (0.208, 0.043, 0.035)    | 0.115 (0.161, 0.026, 0.027) | 0.091 (0.211, 0.045, 0.036) | 0.508 (0.184, 0.034, 0.031) | 0.805 (0.081, 0.007, 0.014) | 0.873 (0.056, 0.003, 0.01)  |
> | EI     | 0.587 (0.19, 0.036, 0.032)     | 0.328 (0.186, 0.034, 0.031) | 0.4 (0.265, 0.07, 0.045)    | 0.623 (0.165, 0.027, 0.028) | 0.863 (0.055, 0.003, 0.009) | 0.944 (0.027, 0.001, 0.005) |
> | EI+    | 0.668 (0.208, 0.043, 0.035)    | 0.545 (0.259, 0.067, 0.044) | 0.449 (0.247, 0.061, 0.042) | 0.775 (0.152, 0.023, 0.026) | 0.883 (0.056, 0.003, 0.009) | 0.95 (0.04, 0.002, 0.007)   |
> | EE     | 0.581 (0.184, 0.034, 0.031)    | 0.404 (0.205, 0.042, 0.035) | 0.395 (0.261, 0.068, 0.044) | 0.616 (0.162, 0.026, 0.027) | 0.875 (0.039, 0.002, 0.007) | 0.946 (0.026, 0.001, 0.004) |
> | EE+    | 0.65 (0.22, 0.048, 0.037)      | 0.589 (0.229, 0.052, 0.039) | 0.441 (0.248, 0.062, 0.042) | 0.758 (0.16, 0.025, 0.027)  | 0.893 (0.033, 0.001, 0.006) | 0.95 (0.042, 0.002, 0.007)  |
>
> > As such, it is unclear what is meant by "excel" in this case. Given the highly variable performance values in Figure 3 and the lack of baselines from the literature, it is not possible to conclude this from the available results. You could alternatively state that the dual-view approaches outperform their counterparts across all tasks on average across all metrics.
>
> Thanks for the suggestion. As the reviewer has mentioned, analysis and discussion over results from multi-dimensional evaluation metrics are subtle. Therefore, we changed the term "excel" to a more objective expression following the reviewer's suggestion. We have made the revision accordingly.
>
> > That being said, an additional set of results in the appendix that shows results by scientific domain would also be helpful in supporting this claim, especially if these results were to show that the proposed methods performed best across all domains separately (i.e., tables like those in appendix A, but partitioned by domain instead of task).
>
> Thanks for the suggestion. Here we present the result of domain-indexed performance. We have made the revision accordingly.
>
> **Genetics**
> | Method | IoU(Op) mean (std, var, stderr) | IoU(Prod)                   | IoU(Dev)                    | Sim(Exec)                   | Sim(Goal)                   | Sim(Param)                  |
> | ------ | ------------------------------- | --------------------------- | --------------------------- | --------------------------- | --------------------------- | --------------------------- |
> | FB     | 0.179 (0.113, 0.013, 0.014)     | 0.065 (0.082, 0.007, 0.01)  | 0.037 (0.08, 0.006, 0.01)   | 0.301 (0.116, 0.014, 0.014) | 0.795 (0.091, 0.008, 0.011) | 0.805 (0.066, 0.004, 0.008) |
> | IB     | 0.157 (0.129, 0.017, 0.015)     | 0.042 (0.06, 0.004, 0.007)  | 0.022 (0.059, 0.003, 0.007) | 0.297 (0.137, 0.019, 0.016) | 0.793 (0.07, 0.005, 0.008)  | 0.789 (0.062, 0.004, 0.007) |
> | II     | 0.379 (0.2, 0.04, 0.024)        | 0.12 (0.158, 0.025, 0.019)  | 0.079 (0.16, 0.026, 0.019)  | 0.457 (0.18, 0.032, 0.022)  | 0.807 (0.083, 0.007, 0.01)  | 0.85 (0.072, 0.005, 0.009)  |
> | EI     | 0.599 (0.189, 0.036, 0.023)     | 0.332 (0.177, 0.031, 0.021) | 0.353 (0.243, 0.059, 0.029) | 0.619 (0.164, 0.027, 0.02)  | 0.862 (0.055, 0.003, 0.007) | 0.941 (0.026, 0.001, 0.003) |
> | EI+    | 0.691 (0.198, 0.039, 0.024)     | 0.606 (0.252, 0.064, 0.03)  | 0.429 (0.283, 0.08, 0.034)  | 0.803 (0.151, 0.023, 0.018) | 0.882 (0.054, 0.003, 0.006) | 0.954 (0.033, 0.001, 0.004) |
> | EE     | 0.592 (0.189, 0.036, 0.023)     | 0.415 (0.206, 0.042, 0.025) | 0.351 (0.241, 0.058, 0.029) | 0.615 (0.163, 0.027, 0.02)  | 0.87 (0.052, 0.003, 0.006)  | 0.943 (0.025, 0.001, 0.003) |
> | EE+    | 0.677 (0.21, 0.044, 0.025)      | 0.653 (0.228, 0.052, 0.027) | 0.425 (0.28, 0.078, 0.033)  | 0.791 (0.161, 0.026, 0.019) | 0.888 (0.045, 0.002, 0.005) | 0.955 (0.034, 0.001, 0.004) |

---

> ### Author Response · Authors · 2024-11-25
> **Response to reviewer #LMMv - 6.3**
>
> **Ecology**
> | Method | IoU(Op) mean (std, var, stderr) | IoU(Prod)                   | IoU(Dev)                    | Sim(Exec)                   | Sim(Goal)                   | Sim(Param)                  |
> | ------ | ------------------------------- | --------------------------- | --------------------------- | --------------------------- | --------------------------- | --------------------------- |
> | FB     | 0.155 (0.085, 0.007, 0.023)     | 0.03 (0.035, 0.001, 0.01)   | 0.021 (0.048, 0.002, 0.013) | 0.297 (0.088, 0.008, 0.024) | 0.781 (0.096, 0.009, 0.027) | 0.807 (0.056, 0.003, 0.015) |
> | IB     | 0.162 (0.118, 0.014, 0.033)     | 0.006 (0.015, 0.0, 0.004)   | 0.03 (0.058, 0.003, 0.016)  | 0.275 (0.105, 0.011, 0.029) | 0.763 (0.09, 0.008, 0.025)  | 0.788 (0.063, 0.004, 0.017) |
> | II     | 0.386 (0.176, 0.031, 0.049)     | 0.043 (0.062, 0.004, 0.017) | 0.027 (0.062, 0.004, 0.017) | 0.448 (0.131, 0.017, 0.036) | 0.788 (0.065, 0.004, 0.018) | 0.856 (0.044, 0.002, 0.012) |
> | EI     | 0.458 (0.171, 0.029, 0.047)     | 0.259 (0.134, 0.018, 0.037) | 0.351 (0.195, 0.038, 0.054) | 0.514 (0.142, 0.02, 0.039)  | 0.879 (0.048, 0.002, 0.013) | 0.933 (0.028, 0.001, 0.008) |
> | EI+    | 0.411 (0.134, 0.018, 0.037)     | 0.569 (0.133, 0.018, 0.037) | 0.359 (0.175, 0.03, 0.048)  | 0.586 (0.127, 0.016, 0.035) | 0.888 (0.052, 0.003, 0.015) | 0.945 (0.023, 0.001, 0.006) |
> | EE     | 0.458 (0.171, 0.029, 0.047)     | 0.347 (0.151, 0.023, 0.042) | 0.351 (0.195, 0.038, 0.054) | 0.507 (0.138, 0.019, 0.038) | 0.874 (0.048, 0.002, 0.013) | 0.934 (0.029, 0.001, 0.008) |
> | EE+    | 0.414 (0.142, 0.02, 0.039)      | 0.581 (0.141, 0.02, 0.039)  | 0.346 (0.177, 0.031, 0.049) | 0.586 (0.131, 0.017, 0.036) | 0.91 (0.035, 0.001, 0.01)   | 0.944 (0.024, 0.001, 0.007) |
>
> **Medical**
> | Method | IoU(Op) mean (std, var, stderr) | IoU(Prod)                   | IoU(Dev)                    | Sim(Exec)                   | Sim(Goal)                   | Sim(Param)                  |
> | ------ | ------------------------------- | --------------------------- | --------------------------- | --------------------------- | --------------------------- | --------------------------- |
> | FB     | 0.174 (0.085, 0.007, 0.016)     | 0.048 (0.07, 0.005, 0.013)  | 0.03 (0.067, 0.004, 0.013)  | 0.312 (0.075, 0.006, 0.014) | 0.796 (0.087, 0.007, 0.017) | 0.839 (0.043, 0.002, 0.008) |
> | IB     | 0.139 (0.054, 0.003, 0.01)      | 0.029 (0.038, 0.001, 0.007) | 0.023 (0.063, 0.004, 0.012) | 0.264 (0.045, 0.002, 0.009) | 0.721 (0.123, 0.015, 0.024) | 0.795 (0.07, 0.005, 0.013)  |
> | II     | 0.373 (0.093, 0.009, 0.018)     | 0.081 (0.087, 0.008, 0.017) | 0.091 (0.205, 0.042, 0.039) | 0.424 (0.072, 0.005, 0.014) | 0.776 (0.097, 0.009, 0.019) | 0.871 (0.041, 0.002, 0.008) |
> | EI     | 0.604 (0.167, 0.028, 0.032)     | 0.322 (0.146, 0.021, 0.028) | 0.309 (0.253, 0.064, 0.049) | 0.594 (0.148, 0.022, 0.029) | 0.861 (0.079, 0.006, 0.015) | 0.932 (0.031, 0.001, 0.006) |
> | EI+    | 0.615 (0.196, 0.039, 0.038)     | 0.574 (0.242, 0.059, 0.047) | 0.4 (0.198, 0.039, 0.038)   | 0.758 (0.149, 0.022, 0.029) | 0.871 (0.06, 0.004, 0.012)  | 0.952 (0.021, 0.0, 0.004)   |
> | EE     | 0.591 (0.158, 0.025, 0.03)      | 0.373 (0.166, 0.028, 0.032) | 0.298 (0.234, 0.055, 0.045) | 0.583 (0.149, 0.022, 0.029) | 0.873 (0.054, 0.003, 0.01)  | 0.936 (0.03, 0.001, 0.006)  |
> | EE+    | 0.615 (0.197, 0.039, 0.038)     | 0.613 (0.21, 0.044, 0.04)   | 0.39 (0.202, 0.041, 0.039)  | 0.756 (0.151, 0.023, 0.029) | 0.891 (0.04, 0.002, 0.008)  | 0.955 (0.019, 0.0, 0.004)   |

---

> ### Author Response · Authors · 2024-11-25
> **Response to reviewer #LMMv - 6.4**
>
> **BioEng**
> | Method | IoU(Op) mean (std, var, stderr) | IoU(Prod)                   | IoU(Dev)                    | Sim(Exec)                   | Sim(Goal)                   | Sim(Param)                  |
> | ------ | ------------------------------- | --------------------------- | --------------------------- | --------------------------- | --------------------------- | --------------------------- |
> | FB     | 0.176 (0.085, 0.007, 0.016)     | 0.048 (0.089, 0.008, 0.016) | 0.05 (0.081, 0.007, 0.015)  | 0.3 (0.084, 0.007, 0.015)   | 0.79 (0.09, 0.008, 0.016)   | 0.826 (0.042, 0.002, 0.008) |
> | IB     | 0.149 (0.077, 0.006, 0.014)     | 0.05 (0.087, 0.008, 0.016)  | 0.038 (0.091, 0.008, 0.017) | 0.286 (0.078, 0.006, 0.014) | 0.767 (0.083, 0.007, 0.015) | 0.797 (0.046, 0.002, 0.008) |
> | II     | 0.352 (0.151, 0.023, 0.028)     | 0.062 (0.09, 0.008, 0.016)  | 0.066 (0.187, 0.035, 0.034) | 0.443 (0.125, 0.016, 0.023) | 0.81 (0.073, 0.005, 0.013)  | 0.86 (0.045, 0.002, 0.008)  |
> | EI     | 0.565 (0.164, 0.027, 0.03)      | 0.307 (0.186, 0.034, 0.034) | 0.31 (0.249, 0.062, 0.045)  | 0.603 (0.169, 0.029, 0.031) | 0.851 (0.072, 0.005, 0.013) | 0.93 (0.033, 0.001, 0.006)  |
> | EI+    | 0.657 (0.209, 0.044, 0.038)     | 0.577 (0.177, 0.031, 0.032) | 0.394 (0.241, 0.058, 0.044) | 0.743 (0.179, 0.032, 0.033) | 0.888 (0.056, 0.003, 0.01)  | 0.944 (0.041, 0.002, 0.007) |
> | EE     | 0.558 (0.162, 0.026, 0.03)      | 0.392 (0.214, 0.046, 0.039) | 0.303 (0.246, 0.061, 0.045) | 0.598 (0.165, 0.027, 0.03)  | 0.855 (0.076, 0.006, 0.014) | 0.933 (0.03, 0.001, 0.005)  |
> | EE+    | 0.653 (0.206, 0.042, 0.038)     | 0.614 (0.172, 0.03, 0.031)  | 0.401 (0.246, 0.061, 0.045) | 0.742 (0.176, 0.031, 0.032) | 0.9 (0.046, 0.002, 0.008)   | 0.945 (0.041, 0.002, 0.007) |
>
> > However, it could be improved even more by relating it to existing work that creates structured workflows using LLMs, and relating it to the more general problem of getting LLMs to structure their outputs.
>
> Thanks for this insightful suggestion. In this work, our proposed representation structures the information into multiple granularities from coarse- to fine-grained, including operations, reagents, devices, and their corresponding parameters. By using machines' externalized language in parallel with humans' internalized language [1], the hierarchical structure of information can be precisely captured, resulting in at least rational output in the local context, namely, a value for the key at least lies in its permissible range of value. By contrast, end-to-end natural language representation flattens the hierarchical structure of information. Although humans are able to recognize them thanks to internalized language [2], machines may generate "not even wrong" irrelevant content with misinterpreted information structures [3]. We have revised the discussion sections to connect these ideas.
>
> References:
>
> [1] Chomsky, N. (2007). Approaching UG from below. Interfaces+ recursion= language, 89, 1-30.
>
> [2] Chomsky, N. (1956). Three models for the description of language. IRE Transactions on information theory, 2(3), 113-124.
>
> [3] Zhang, Y., Li, Y., Cui, L., Cai, D.,  Liu, L., Fu, T., ... & Shi, S. (2023). Siren's song in the AI ocean:  a survey on hallucination in large language models. arXiv preprint arXiv:2309.01219.
>
> > Very limited discussion of the results.
>
> Thanks for the comment. Our discussion covers three typical aspects of method evaluation: (i) discussion on the contributions of different building blocks to the performance of the proposed method; (ii) discussion on the scalability of the proposed method from relatively simple tasks to relatively complicated ones; and (iii) discussion on the generalizability of the proposed method towards different application domains. We acknowledge that due to space limitations, these aspects are not sufficiently extended. Furthermore, some other critical aspects, such as the influence of automated protocol design on human experts and the limitations of the current solution, are not covered. We have made the revisions extending the insightful discussions on these topics, to improve the comprehensibility of the paper.

---

> ### Author Response · Authors · 2024-11-25
> **Response to reviewer #LMMv - 7**
>
> > Unclear whether they consider the problem setting to be a contribution. It would be important to contrast the problem formulation with that of prior work, such as that of O’Donoghue et al. (2023). In particular,  whether their formulation of the protocol design problem (not the representation itself) differs in any meaningful way. This is important because the identification of the protocol design problem is cited as a  contribution on line 147.
>
> Thanks for the comment. As the reviewer has mentioned, the protocol design problem has been explored by previous works. Differently, our major contribution is identifying the problem of representation for protocol design, which was not explicitly considered by the previous work. The type of representation is the pivot variable we are controlling in the evaluation. In comparison with our compared baseline approaches, including the original natural language-based text representation, i.e., FB; the instance actions with attributes representation developed upon BioPlanner [1], i.e., IB and II; and the operation-centric view-only representation, i.e., EI and EE, the results indicate that our proposed approaches with the dual-representation of operation- and product-flow-centric views, i.e., EI+ and EE+, significantly outperform their counterparts with alternative representations (EE+ vs. EE: $t(278) = 8.007, \mu_d < 0, p < .0001$; EI+ vs. EI: $t(278) = 8.397, \mu_d < 0, p < .0001$; EE+ vs. II: $t(278) = 24.493, \mu_d < 0, p < .0001$; EI+ vs. II: $t(278) = 23.855, \mu_d < 0, p < .0001$; EE vs. II: $t(278) = 16.315, \mu_d < 0, p < .0001$; EI vs. II: $t(278) = 15.259, \mu_d < 0, p < .0001$; II vs. FB: $t(278) = 8.340, \mu_d < 0, p < .0001$; also see the figures below). This result suggests that representation can be a key factor to the extent we are able to elicit the potential of knowledge-based machine designers like LLMs on protocol design. Therefore, the problem of representation for protocol design should be a sufficiently significant contribution, as we have stated on line 147.
>
> [Comparison between the capabilities of different machine designers across the six dimensions](https://anonymous.4open.science/api/repo/AutoDSL-Planning-Figure-0DFE/file/redar_2.png?v=a97ddaac)
>
> To clarify, the protocol design problem part is not essentially distinct from that introduced in the previous work [1], and we did not identify it as a major contribution. Since there are only natural language-based empirical descriptions of the protocol design problem, we provide the symbolic notations of the protocol design problem, in order to set up the foundation of the problem formulation of representation for protocol design. We have made the revisions to make this point clearer.
>
> References:
>
> [1] O’Donoghue, O., Shtedritski, A.,  Ginger, J., Abboud, R., Ghareeb, A., & Rodriques, S. (2023,  December). BioPlanner: Automatic Evaluation of LLMs on Protocol Planning  in Biology. In Proceedings of the 2023 Conference on Empirical Methods in Natural Language Processing (pp. 2676-2694).

---

### Official Review · Reviewer_WJzq · 2024-11-04

**Soundness:** 3
**Presentation:** 2
**Contribution:** 3
**Rating:** 6
**Confidence:** 3

**Summary:**

This paper addresses the challenge of automated protocol design for conducting scientific experiments. While it is currently possible to automatically execute predefined protocols, it remains a challenge how to design new protocols or modify existing ones to achieve novel experimental objectives. The authors propose a hierarchical representation framework that enables automated protocol design by capturing both procedural and domain knowledge at multiple levels of abstraction. Specifically, the framework consists of three hierarchical levels: (1) a basic level that breaks down protocols into individual actions with their specific attributes (like timing, temperature, etc.), (2) an operation-centric view that groups and generalizes these actions based on their purpose, and (3) a product-flow-centric view that tracks how materials change and interact throughout the experiment. These three levels are implemented using Domain-Specific Languages (DSLs) that help verify and ensure the correctness of protocols. The framework includes an automated method to generate these representations for different scientific domains using (mostly) non-parametric modeling, allowing it to learn and adapt to different types of experiments.

**Strengths:**

1. The premise of the paper is extremely compelling - enabling AI agents (specifically LLMs) with a structured representation of expert knowledge that allows them to harness their generative capabilities to create novel experimental protocols.

2. Representing the task of protocol modeling protocols as conditional probabilities is particularly elegant, as it enables better control and parameter learning compared to black-box approaches like neural networks, primarily because this representation naturally captures the logical dependencies between experimental steps.

3. The dual verification system combining product and operation views is especially clever, as it allows simultaneous optimization of protocol operations while ensuring each step both builds on previous products and generates viable outputs for subsequent steps.

4.  The comprehensive evaluation across multiple scientific domains with real human experts demonstrates the practical utility and broad applicability of the method.

**Weaknesses:**

1. The language is often very abstract and abstruse. This is sometimes expected because the topic itself is very abstract. But making it simpler would enable the reader to appreciate the contributions more if there was more clarity in explanations.

2. The work suggests it uses LLMs for the protocol design and it is indeed mentioned where LLMs are used. There are no details, however
on how exactly LLMs are employed in the suggested framework (i.e. do they receive the input from the DSL? or is the DSL a step along the way). It is somewhat intuitive, but an explicit description would be helpful. I appreciated the details in the Appendix but this should be mentioned in the main text.

3. It is not clear how scalable this approach is for more difficult protocols. It is not also clear for someone outside of the tested areas if there even are more difficult protocols. This could be addressed.

Other points:
- It would be very helpful to start with a motivational example of some particular case of research design like the one in Figure 1. Currently, it reads very abstract. But the Figure is helpful.

- Lines 92-107 are supposed to briefly summarize the mechanism introduced by authors but it is very hard to comprehend. It would be good to correspond these to Figure 1B.

**Questions:**

1. Can you provide more intuition on what the interface \phi is? Is that simply a set of possible experiments for a given operation (such as the given “homogenization”)? Or a set of functions? Why is it operationalized the way it was operationalized?

2. The motivating examples and descriptions seem to be very heavily concerned with natural sciences. Is it possible to extend your framework to, say psychology or experimentation in computer science itself?

3. How scalable is the framework for more complex protocols?

---

> ### Author Response · Authors · 2024-11-25
> **Response to reviewer #WJzq - 1**
>
> > Can you provide more intuition on what the interface \phi is? Is that simply a set of possible experiments for a given operation (such as the given “homogenization”)? Or a set of functions? Why is it operationalized the way it was operationalized?
>
> This is a very good question. Interface is a concept of functional abstraction [1]. Interface disentangles the abstract functionality on the semantics level and its corresponding implementation details on the execution level. This approach encapsulates the implementation of an operation into a "black-box", so the users of the operation would only need to consider its input and output. Therefore, with such encapsulated representation for protocol design, we only need to care about the consistency between the output of the predecessor operation and the input of the successor operation, without caring about their implementation details.
>
> This is the idea behind operationalization. Operationalization makes the interface an abstract function over all relative instance actions. The interface is abstracted from the execution contexts of all instance actions with the same reference name, i.e., the same purpose, and can be instantiated to an instance action given a specific execution context. A specific context can be the predecessor operation, the successor operation, the precondition, or the postcondition of the considered operation. An instance action configures a specific implementation for a specific execution context. For the operation "homogenization", the implementation of one instance action can be "using an ultrasonic homogenizer" if the precondition, namely, the execution context, has intermediate product "cell suspension" available; the implementation of another instance action can be "using a bead mill" if the precondition contains tissue. This example demonstrates the relationship between interface and instance actions of an operation: the interface is abstracted from the set of instance actions and can be instantiated to instance actions.
>
> Here we also give a more intuitive example to enhance the reviewer's comprehension. Consider the culinary scenario with the actions "frying the egg", "frying the fish", and "frying the steak".These are different instance actions coming with the same purpose "to fry something". Therefore, we can abstract the interface from these instance actions to operationalize the operation "fry". The input of "fry" should be something raw and its output should be something fried. Given different preconditions with available eggs or pieces of steak, the abstract semantic operation "fry" can be grounded to instance actions "frying the egg" or "frying the steak" respectively, through the instantiation of the interface. In summary, an interface serves as the bridge between the semantics level and the execution level. We have made the revision to enhance the accessibility of the concept.
>
> References:
>
> [1] Abelson, H., & Sussman, G. J. (1996). Structure and interpretation of computer programs (p. 688). The MIT Press.

---

> > ### Comment · Reviewer_WJzq · 2024-11-30
> >
> > Thank you for this response, this makes the operationalization clear. I assure you however, that it is not clear while reading Section 2.3. The need for an interface is well-established there. But if possible -- Section 2.3 could really use a sentence conveying this intuition you gave above :
> >
> > "Consider the culinary scenario with the actions "frying the egg", "frying the fish", and "frying the steak".These are different instance actions coming with the same purpose "to fry something". Therefore, we can abstract the interface from these instance actions to operationalize the operation "fry"."
> >
> > (although frying itself may seem out of place with so many biologically-inspired examples).

---

> > > ### Author Response · Authors · 2024-11-30
> > >
> > > Thank you for your suggestion. We will revise Section 2.3 to integrate the intuition behind operationalization. Since you have mentioned the "out of place" issue, we will adapt this intuition to the biologically-inspired examples in the context. For example, the actions "homogenization of mouse liver tissue ...", "homogenization of bacterial cell suspension ...", and "homogenization of bacterial air samples ..." are different instance actions coming with the same purpose "to homogenize something", and can be operationalized to the operation "homogenization". We will add this one-sentence intuition to the main text, and retain the culinarily-inspired examples in the additional remarks for a more intuitive reference.

---

> ### Author Response · Authors · 2024-11-25
> **Response to reviewer #WJzq - 2**
>
> > The motivating examples and descriptions seem to be very heavily concerned with natural sciences. Is it possible to extend your framework to, say psychology or experimentation in computer science itself?
>
> Thanks for the question. In theory, our framework can be applied to any field that requires adherence to specific protocols and has a need for automated execution. As an example, consider an automated kitchen controlled by a computer, which we provide here for your reference:
> 1. Assuming the automated kitchen’s computer is already programmed to prepare “braised pork ribs” and “steamed sea bass”:
>   **Braised Pork Ribs**
> ```Plain
> Braised Pork Ribs
> 1.Select pork ribs as the main ingredient.
> 2.Heat a pan over high heat.
> 3.Add the ribs to the pan and fry for about 5 minutes until they are browned.
> 4.Add seasonings: soy sauce and sugar.
> 5.Reduce the heat to medium.
> 6.Simmer the ribs for 30 minutes until tender.
> 7.Serve hot.
> ```
> ```Plain
> START
> SELECT ingredient: ribs
> ACTION: fry, temperature: high, time: 5 min
> ADD seasoning: soy sauce, sugar
> ACTION: simmer, temperature: medium, time: 30 min
> END
> ```
>   **Steamed sea bass**
> ```Plain
> Steamed Sea Bass
> 1.Select a whole sea bass as the main ingredient.
> 2.Prepare a steamer and heat it to high temperature.
> 3.Place the sea bass in the steamer.
> 4.Steam the fish for about 15 minutes until fully cooked.
> 5.Add seasonings: ginger slices and chopped scallions.
> 6.Serve immediately with the garnish.
> ```
> ```Plain
> START
> SELECT ingredient: sea bass
> ACTION: steam, temperature: high, time: 15 min
> ADD seasoning: ginger, scallion
> END
> ```
> 2. Next, we can derive the corresponding DSL. For instance:
> ```JSON
> {
>   "cooking_methods": {
>     "braise": {
>       "steps": [
>         {"type": "fry", "temperature": "high", "time": "5 min"},
>         {"type": "simmer", "temperature": "medium", "time": "30 min"}
>       ],
>       "seasoning": ["soy sauce", "sugar"]
>     },
>     "steam": {
>       "steps": [
>         {"type": "steam", "temperature": "high", "time": "15 min"}
>       ],
>       "seasoning": ["ginger", "scallion"]
>     }
>   },
>   "ingredients": {
>     "ribs": {
>       "category": "meat",
>       "default_braise_time": "30 min"
>     },
>     "sea_bass": {
>       "category": "fish",
>       "default_braise_time": "20 min",
>       "default_steam_time": "15 min"
>     }
>   }
> }
> ```
> 3. Now, let’s create a new recipe for Braised Sea Bass by combining the braising technique with sea bass as the main ingredient.
> ```Plain
> START
> SELECT ingredient: sea bass
> ACTION: fry, temperature: high, time: 5 min
> ADD seasoning: soy sauce, sugar
> ACTION: simmer, temperature: medium, time: 20 min
> END
> ```
> > How scalable is the framework for more complex protocols?
>
> This is a very good question. Following the convention of experimental sciences, more complex protocols can be referred to longer protocols. Here we profile the length distribution of the groundtruth in our test set across the four domains. On this basis, we select one of the longest protocols to demonstrate the scalability of the framework.
>
> Number of steps (corresponding to the length of the ground-truth pseudocode program): min=2, avg=12.62, max=33
>
> [Steps of novel protocols across four domains](https://anonymous.4open.science/api/repo/AutoDSL-Planning-Figure-0DFE/file/len_count_program.png?v=7d71ce56)
>
> We select a complex protocol as an example to demonstrate the scalability of our framework. This protocol consists of 26 steps in its pseudocode program and 131 steps in its natural language procedure. Below, we present a fragment of the generated results from our best approach alongside several baseline methods:
>
> [Complex example](https://anonymous.4open.science/api/repo/AutoDSL-Planning-Figure-0DFE/file/example2.png?v=c742c1a1)
>
> 1. Overall, our framework demonstrates strong performance even when handling complex protocols;
> 2. When facing long and complex protocols, the effect will be influenced by the hallucination of LLM:
>    1. The machine designers may select devices different from those specified in the ground truth to complete the experiment;
>    2. The consistency of LLM-generated design results may decrease, underscoring the importance of validating LLM outputs through DSL.
>
> The more rigorous analysis of scalability represents a promising avenue for future research, and we appreciate the reviewer's insightful suggestion in this regard. We have included this additional discussion in the revised version.

---

> > ### Comment · Reviewer_WJzq · 2024-11-30
> >
> > This is great, thank you, I appreciate you added the first example to the appendix. It's already very long but I think it would be also helpful if you added the scalability analysis. Seems very promising, good job!

---

> > > ### Author Response · Authors · 2024-11-30
> > >
> > > Thank you for your suggestion. We will incorporate scalability analysis into the Appendix. We are continuously working to further elaborate on this part.

---

> ### Author Response · Authors · 2024-11-25
> **Response to reviewer #WJzq - 3.1**
>
> > The language is often very abstract and abstruse. This is sometimes expected because the topic itself is very abstract. But making it simpler would enable the reader to appreciate the contributions more if there was more clarity in explanations.
>
> Thanks for the comment. The same concern was raised when we were writing the paper. We have tried to make the expressions as concrete and intuitive as possible. However, as the reviewer has mentioned, the topic is intrinsically abstract because we are describing a scientific problem abstracted from real-world applications. We appreciate the reviewer for pointing this out. We are trying our best in the revision process to enhance the accessibility of the writing.
>
> > The work suggests it uses LLMs for the protocol design and it is indeed mentioned where LLMs are used. There are no details, however on how exactly LLMs are employed in the suggested framework (i.e. do they receive the input from the DSL? or is the DSL a step along the way). It is somewhat intuitive, but an explicit description would be helpful. I appreciated the details in the Appendix but this should be mentioned in the main text.
>
> Thanks for the question.
> 1. We begin by retrieving DSL instructions that are potentially relevant to the target protocol;
> 2. Next, we combine the title, description, and DSL instructions as a prompt for the LLM.
>
> Here is an example of a prompt used for EI and the initial stage of EE methods.
> ```
> Your goal is to generate plan in domain specific language (DSL) for biology protocols.
> The DSL specifications related to the operations involved in the experiment are provided. The DSL specification of each operation consists of multiple patterns, each pattern is an operation execution paradigm.
> Output each operation of the plan in the form of a DSL program. Each DSL program is a dictionary. The final plan consists of the program of each step and is returned in a json block, without any annotation.
>
> Here is an example of how to generate plan in DSL for a biology protocol.
>
> EXAMPLE:
>
> {example protocol title}
>
> Here are some extra details about the protocol:
>
> {example protocol description}
>
> example plan in DSL:
>
> {example plan}
>
> YOUR TASK:
> Generate plan in DSL for a protocol for High Molecular Weight genomic DNA from coral sperm.
>
> Here are some extra details about the protocol:
>
> This molecular biology protocol aims to extract high molecular weight genomic DNA from coral sperm using a method based on RNAse and ProteinaseK treatment, followed by phenol/chloroform extraction. The protocol prioritizes purity and minimal damage to the DNA, making it suitable for downstream genetic analyses.
>
> You can choose to instantiate the following DSL specification to construct the DSL program:
>
> {
>     "Grind": [
>         {
>             "pattern": {
>                 "Precond": {
>                     "SlotArgNum": 1,
>                     "SlotArg": [
>                         "Liquid"
>                     ]
>                 },
>                 "Execution": [
>                     {
>                         "DeviceType": "centrifuge",
>                         "Config": {
>                             "time": [
>                                 "3 - 5 sec"
>                             ]
>                         }
>                     },
>                     {
>                         "DeviceType": "mortar and pestle",
>                         "Config": {}
>                     }
>                 ],
>                 "Postcond": {
>                     "EmitArgNum": 1,
>                     "EmitArg": [
>                         "Solid"
>                     ]
>                 }
>             },
>             "examples": [
>                 "Grind again as in step B3 ( make sure the water is not frozen before grinding ) .",
>                 "grind tissue with a mortar and pestle in the presence of liquid nitrogen .",
>                 "Grind the tissue to a fine powder by using mortar and pestle using liquid nitrogen .",
>                 "The material before the grinding process ( Before ) and the fully grinded material ( After ) DNA extractionBriefly centrifuge the CTAB treated samples for 3 - 5 sec ."
>             ]
>         }
>     ],
>     ...
>     {The rest of Operation-view DSL specification}
> }
>
> Your plan in DSL program:
> ```

---

> > ### Comment · Reviewer_WJzq · 2024-11-30
> >
> > Thank you for this clarification. Please, add this 2-sentence information on how you specifically use the LLMs and the DSL to the appropriate section in the main text.

---

> > > ### Author Response · Authors · 2024-11-30
> > >
> > > Thank you for your suggestion. We will incorporate this two-sentence information into Section 4.3 "Machine designers".

---

> ### Author Response · Authors · 2024-11-25
> **Response to reviewer #WJzq - 3.2**
>
> Here is an example of a prompt used for EI+ and the initial stage of EE+ methods.
> ```
> Your goal is to generate plan in domain specific language (DSL) for biology protocols.
> Two perspectives of the DSL specification are provided: the specification for experimental operations and the specification for experimental products.
> The DSL specification of each operation or product consists of multiple patterns, each pattern is an operation execution paradigm or a product flow paradigm.
> Output every operation of the plan in the form of an operation DSL program and every product of the plan in the form of a product DSL program.
> Each DSL program is a dictionary. The final plan consists of the program of each step and product and is returned in a json block, without any annotation.
>
> Here is an example of how to generate plan in DSL for a biology protocol.
>
> EXAMPLE:
>
> {example protocol title}
>
> Here are some extra details about the protocol:
>
> {example protocol description}
>
> example plan in DSL:
>
> {example plan}
>
> YOUR TASK:
> Generate plan in DSL for a protocol for High Molecular Weight genomic DNA from coral sperm.
>
> Here are some extra details about the protocol:
>
> This molecular biology protocol aims to extract high molecular weight genomic DNA from coral sperm using a method based on RNAse and ProteinaseK treatment, followed by phenol/chloroform extraction. The protocol prioritizes purity and minimal damage to the DNA, making it suitable for downstream genetic analyses.
>
> You can choose to instantiate the following DSL specifications to construct the DSL program:
>
> Operation-view DSL specification:
> {
>     "Place": [
>         {
>             "pattern": {
>                 "Precond": {
>                     "SlotArgNum": 1,
>                     "SlotArg": [
>                         "Physical Object"
>                     ]
>                 },
>                 "Execution": [
>                     {
>                         "DeviceType": "pasteur pipet",
>                         "Config": {}
>                     }
>                 ],
>                 "Postcond": {}
>             },
>             "examples": [
>                 "Place the clipped cartridges into the grid box.",
>                 "( B ) The needle bevel is placed against the side of the 15 mL tube and the liquid gently layered on top of the previous density .",
>                 "Place a cover slide ( 24 x 60 mm ) on top of your samples ."
>             ]
>         }
>     ],
>     ...
>     {The rest of Operation-view DSL specification}
> }
>
> Product-view DSL specification:
> {
>     "suspended semen": {
>         "Pred": "Transfer Operations",
>         "FlowUnit": {
>             "Component": "suspended semen",
>             "ComponentType": "Liquid",
>             "Vol": [
>                 "one drop"
>             ],
>             "Container": [
>                 "slide"
>             ],
>             "Cond": {}
>         },
>         "Succ": "Detection and Measurement Operations"
>     },
>     ...
>     {The rest of Product-view DSL specification}
> }
>
> Your plan in DSL program:
> ```
> We have referred this to the main text in the revised version to enhance the accessibility of the paper.

---

> ### Author Response · Authors · 2024-11-25
> **Response to reviewer #WJzq - 4**
>
> > It is not clear how scalable this approach is for more difficult protocols. It is not also clear for someone outside of the tested areas if there even are more difficult protocols. This could be addressed.
>
> This is a very good question. Following the convention of experimental sciences, more complex protocols can be referred to longer protocols. Here we profile the length distribution of the groundtruth in our test set across the four domains. On this basis, we select one of the longest protocols to demonstrate the scalability of the framework.
>
> Number of steps (corresponding to the length of the ground-truth pseudocode program): min=2, avg=12.62, max=33
>
> [Steps of novel protocols across four domains](https://anonymous.4open.science/api/repo/AutoDSL-Planning-Figure-0DFE/file/len_count_program.png?v=7d71ce56)
>
> We select a complex protocol as an example to demonstrate the scalability of our framework. This protocol consists of 26 steps in its pseudocode program and 131 steps in its natural language procedure. Below, we present a fragment of the generated results from our best approach alongside several baseline methods:
>
> [Complex example](https://anonymous.4open.science/api/repo/AutoDSL-Planning-Figure-0DFE/file/example2.png?v=c742c1a1)
>
> 1. Overall, our framework demonstrates strong performance even when handling complex protocols;
> 2. When facing long and complex protocols, the effect will be influenced by the hallucination of LLM:
>    1. The machine designers may select devices different from those specified in the ground truth to complete the experiment;
>    2. The consistency of LLM-generated design results may decrease, underscoring the importance of validating LLM outputs through DSL.
>
> The more rigorous analysis of scalability represents a promising avenue for future research, and we appreciate the reviewer's insightful suggestion in this regard. We have included this additional discussion in the revised version.
>
> > It would be very helpful to start with a motivational example of some particular case of research design like the one in Figure 1. Currently, it reads very abstract. But the Figure is helpful.
>
> Thanks for pointing this out. We have extended both the caption of Fig 1 and the corresponding text for reference to Fig 1.
>
> > Lines 92-107 are supposed to briefly summarize the mechanism introduced by authors but it is very hard to comprehend. It would be good to correspond these to Figure 1B.
>
> Thanks for the suggestion. We have corresponded the ideas and concepts introduced in this part to Figure 1B.

---

> > ### Comment · Reviewer_WJzq · 2024-11-30
> >
> > Thank you for moving Figure 1 higher and referring to it in the main text. I do not think this change is sufficient, however. The description in lines 91-107 should use examples from Figure 1. It would be easier if the theoretical definitions of the levels were cut to the necessary minimum but the intuition behind the levels was conveyed via the figure and examples. You could move more detailed description to the Appendix.

---

> > > ### Author Response · Authors · 2024-11-30
> > >
> > > Thank you for your suggestion. We will extend this part following your suggestion. We will ground the theoretical definitions of the levels to the examples in Figure 1 (B), to make them more intuitive and, consequently, more accessible. Additionally, we will move the redundantly detailed description to the Appendix.

---

### Official Review · Reviewer_BhtL · 2024-11-08

**Soundness:** 2
**Presentation:** 1
**Contribution:** 3
**Rating:** 6
**Confidence:** 2

**Summary:**

This article studies for automatic experiment protocol design, representations in different levels of semantics : the protocol element instantialization with elementary operation representation, function abstraction as a sequential representation of the operation, and a model abstraction which specifies reagent and intermediate products.
The authors describe the 3 representations, propose an algorithm to automatically generate new protocols and demonstrate the creativity of the protocol generators based on the representation used.

The work can be framed as the hierarchical representation of policies for an MDP, trained from a natural language corpus, and used to generate new MDPs. The strength of this work lies in the fact that the methodology is tested across several experimental domains. However, the formalisation is not sound.

**Strengths:**

The article ambitiously represents a synthetic representation of experimental protocols across different experimental sciences : Genetics, Medical, Bioengineering and Ecology.
The article methodically studies the automatic computation of the representation, and then assesses the utility of the protocols generated.

**Weaknesses:**

* The article lacks a sound formalisation of the problem.

** While the issue is to represent actions (experimental operations) at different levels of hierarchy, to model the change in the environmental reagents, there is no hint of using the formalisations of Markov Decision Processes or hierarchical actions/reinforcement learning.
When the three levels of representations can be easily formalised using the MPD formalisation (for instance in terms of actions/policies, options and waypoints states/subgoals), the proposed formalisation seems imprecise.

** while the word "planning" is used several times in the text, it is only at l.444 that the authors explicit planning tasks as "the exploration
of novel experimental goals". This definition is confusing. While "planning" generally refers to finding the succession of actions to obtain a predefined goal, exploring novel goals is different, and can be referred to as "goal babbling" for instance.

** In 2.2. the vocabulary used is confusing: a precondition is generally a property of the state that allows you to carry out your operations. It is a distinct notion from an input.

** Some terms are not defined. For instance : execution context (in l.190 which is used differently from execution condition) and key-value pairs (l193)


* While the authors present comparative results of their representations and algorithms, there is no comparison with other approaches. How do the results compare quantitatively or qualitatively to the state of the art ?

*While the authors mention as objectives the "exploration of novel goals", "generating novel experimental objectives" and aim to measure "a protocol's novelty", their metrics is only based on similarity measures. Diversity is not mentioned in the criteria. Could you add diversity measures or argue how the proposed metrics take into account diversity ?

Minor comments :

* Figures need a description to better understand what is shown
* l.234: "Any status transition of the product flow is caused, and is only caused, by the effects of operations". This stance excludes general evolving systems.  What about dynamic systems, including with slow transformations, especially in biology or ecology?
* l.440 : "the three scenarios of protocol design introduced in sec 1". It seems sec 1 introduces 3 representations/levels of encapsulation.
* The results should report computing load

Typos :

* l317 : "is consist of "
* l399: "to cover as rich context as possible"
* l420 : "we report and analysis"

**Questions:**

In section 2.1, why is an experimental objective specified both by the final product and the final operation ? I believe the specification of the final product is the objective, and the final operation is in most cases, only the means.  In the example of a test of the significance of a hypothesis, I would re-formulate the objective as the observation of a property that confirms or contradicts the hypothesis. A definition of a hypothesis based on an operation is restrictive on the generalisation of hypotheses. The authors write in l/225 "operations are the methods to realize rather than the objectives to achieve".

Why were other experimental domains, such as chemistry or physics left out ?

In section 4, I do not quite understand how the testing set is evaluated. Under which criteria/input/prompt is each protocol generated ? What is then the corresponding ground truth ?

**Details Of Ethics Concerns:**

This work relies on the analysis of scientific corpus describing experimental protocols. However they declare that the corpora complies with open access policies.

---

> ### Author Response · Authors · 2024-11-25
> **Response to reviewer #BhtL - 1**
>
> > In section 2.1, why is an experimental objective specified both by the final product and the final operation? I believe the specification of the final product is the objective, and the final operation is in most cases, only the means. In the example of a test of the significance of a hypothesis, I would re-formulate the objective as the observation of a property that confirms or contradicts the hypothesis. A definition of a hypothesis based on an operation is restrictive on the generalisation of hypotheses. The authors write in l/225 "operations are the methods to realize rather than the objectives to achieve".
>
> This is a very good question. We originally considered modeling the protocol design problem in an end-to-end fashion. The intuition is that, for experimental objectives such as detecting a predicted behavior or testing a specific hypothesis, the objective not only includes the desired final product, but also the final operation to be conducted upon the final product. This is like an additional step appended to the normal protocols end by a final product. Therefore, we have two possible choices for formulation: (i) end-to-end formulation by integrating the additional step into the protocols; (ii) unified formulation with final product only and leaving out the additional step.
>
> We appreciate the modification that the reviewer has suggested. Indeed, the single additional final operation may not be able to sufficiently account for the additional steps for observation and testing. Moreover, the formulation with an objective specified by two variables can be more complicated than that with only one. In pursuit of generality and succinctness, we choose to follow the reviewer's suggestion and discard the final product variable from the formulation of the objective. We have made the revisions accordingly.
>
> > Why were other experimental domains, such as chemistry or physics left out ?
>
> Thanks for the question. The preliminary factor that restricts our choice of experimental domains is data accessibility. Our corpora are retrieved from open-sourced websites run by top-tier publishers, including Nature's [Protocolexchange](https://protocolexchange.researchsquare.com/), Cell's [Star-protocols](https://star-protocols.cell.com/), [Bio-protocol](https://bio-protocol.org/en), Wiley's [Current Protocols](https://currentprotocols.onlinelibrary.wiley.com/), and [Jove](https://www.jove.com/). We aggregated the corpora and analyzed the themes of the protocols according to the first- and second-level labels attached to them. This results in the taxonomies of the four major domains: Genetics, Medical and Clinical Research (Medical), Ecology and Environmental Research (Ecology), and Bioengineering. Therefore, we employ these four domains in this study.
>
> We recognize that physics and chemistry are also representative domains of experimental sciences, besides Biology, Medical, and Ecology. Due to the higher cost of accessing the corpora of protocols for conducting physics and chemistry experiments, for example, mining the protocol from the "method" section of relevant published papers, we leave the application to physics and chemistry for future work. We have made the revisions to clarify this point.

---

> ### Author Response · Authors · 2024-11-25
> **Response to reviewer #BhtL - 2**
>
> > In section 4, I do not quite understand how the testing set is evaluated. Under which criteria/input/prompt is each protocol generated ? What is then the corresponding ground truth ?
>
> Thanks for the question. We illustrate the criteria, input, prompt, and the corresponding ground truth of the evaluation process as follows. We have updated the appendix with these running examples.
>
> [Running example 1](https://anonymous.4open.science/api/repo/AutoDSL-Planning-Figure-0DFE/file/example1.1.png?v=93587d92)
>
> [Running example 2](https://anonymous.4open.science/api/repo/AutoDSL-Planning-Figure-0DFE/file/example1.2.png?v=8ca0c344)
>
> [Running example 3](https://anonymous.4open.science/api/repo/AutoDSL-Planning-Figure-0DFE/file/example1.3.png?v=3ca1a684)
>
> **Interpretation**
>
> Example 1
>
> |      | Strengths                                                    | Weakness                                                     |
> | ---- | ------------------------------------------------------------ | ------------------------------------------------------------ |
> | EE+  | The method provides detailed tracking of material states (e.g., "State": "Frozen"), which ensures precise control over the experimental conditions. |                                                              |
> | EE   |                                                              | The inclusion of "Lysis_Buffer" as an input in the preconditions is incorrect. |
> | II   |                                                              | This method lacks key parameters and conditions for effective grinding, such as the device type (e.g., “Mortar_and_Pestle”) and the desired output state (e.g., “Fine_Powder”). |
> | FB   |                                                              | The workflow is overly complex, with preliminary steps like sample collection and centrifugation adding unnecessary complications to a simple grinding process. |
>
> Example 2
>
> |      | Strengths                                                    | Weakness                                                     |
> | ---- | ------------------------------------------------------------ | ------------------------------------------------------------ |
> | EE+  | The method accurately matches the temperature and incubation conditions of 37°C, ensuring consistency with the protocol, and correctly tracks the flow of components from treatment to hydrolysate production, aligning well with the intended experimental workflow. |                                                              |
> | EE   |                                                              | The incubation time is incorrect, set to 1 hour instead of the required 3 hours. Additionally, the use of a generic sample input (“Incubated_Sample-1”) instead of specifying RNase_T2 treatment suggests a lack of alignment with the experimental preconditions. |
> | II   |                                                              | The incubation time is incorrectly set to 30 minutes instead of the required 3 hours. Additionally, the use of “hydrolyzed RNAs” as the input mixture is inaccurate, as the incubation step should involve RNase_T2 treatment to produce the hydrolysate, not process an already hydrolyzed sample. |
> | FB   |                                                              | The incubation time is incorrectly set to 30 minutes instead of the required 3 hours. Additionally, the use of “rna_samples” as the input lacks specificity, as it should refer to the RNase_T2-treated samples according to the protocol. |
>
> Example 3
>
> |      | Strengths                                                    | Weakness                                                     |
> | ---- | ------------------------------------------------------------ | ------------------------------------------------------------ |
> | EE+  | The temperature and time parameters are accurately specified. | The configuration lacks the inversion count parameter.       |
> | EE   |                                                              | The specified temperature is incorrect. The protocol requires incubation at 65°C to ensure optimal reaction conditions, but the generated configuration uses 37°C, which is insufficient and may compromise the reaction’s effectiveness. |
> | II   |                                                              | The incubation time is incorrect. The experiment requires a duration of 30 minutes for optimal results, but the generated configuration specifies 1 hour. |
> | FB   |                                                              | The specified incubation time is incorrect; it should be 30 minutes instead of 1 hour. Extending the incubation to 1 hour may negatively impact the sample integrity and alter the reaction dynamics, potentially leading to suboptimal experimental results. |

---

> ### Author Response · Authors · 2024-11-25
> **Response to reviewer #BhtL - 3**
>
> > While the issue is to represent actions (experimental operations) at different levels of hierarchy, to model the change in the environmental reagents, there is no hint of using the formalisations of Markov Decision Processes or hierarchical actions/reinforcement learning. When the three levels of representations can be easily formalised using the MPD formalisation (for instance in terms of actions/policies, options and waypoints states/subgoals), the proposed formalisation seems imprecise.
>
> This is a very good question. The same consideration was evaluated during the decision-making process of designing our representation. It seems that we can formulate the protocol design problem in the fashion of Markov Decision Process (MDP) and solve it by heuristic-based planning methods or Hierarchical Reinforcement Learning (HRL) approaches. However, although the formulation itself is feasible, solving the problem may not be practical. Consider solving the problem through an HRL approach designed for heterogeneous action space with parameters (as the protocol is required to decide both the key properties of an operation and the corresponding values). This hierarchical agent may be trained to converge on a fine-grained environment with a clearly designed reward function, or on a large dataset with trajectories for offline learning. Unfortunately, we have access to neither an interactive environment simulating the experiments nor sufficient data to support offline training [1].
>
> Treating the experimental procedures as a white box and creating digital twins for experiments can be an elegant solution and thereby facilitate various applications other than protocol design. This effort requires elaborated design of simulation granularity, exhaustive collection of primitive principles of the system, efficient implementation of rule production, and define precise metrics for evaluating the distance between current and objective states (serving as a reward function), which can be labor-intensive and is far out of the scope of this work. On the other hand, viewing those published protocols as trajectories for offline training, the scale of the offline dataset and the density of the reward function are much too insufficient to support training to convergence. Augmenting the data, synthesizing realistic trajectories, or enhancing the accessibility of protocols, are out of the scope of this work. Given the current obstacles, we choose not to formulate the problem in an MDP fashion. Though an MDP-style formulation can be more precise and elegant, it may misguide the readers to some extent. Instead, we decide to leverage the rich domain-specific knowledge provided by knowledge-based agents such as Large Language Models (LLMs), where knowledge may complement the lack of data and dense reward function. This design choice is also in line with the initial attempts on automatic experiment design [2, 3].
>
> In summary, our design choice of formulation is a compromise based on currently limited resources and restricted scope. Nonetheless, the exploration of more precise and elegant formulations represents a promising avenue for future research, and we appreciate the reviewer's insightful suggestion in this regard. We have made the revisions to convey these insights.
>
> References:
>
> [1] Pateria, S., Subagdja, B., Tan, A. H., & Quek, C. (2021). Hierarchical reinforcement learning: A comprehensive survey. ACM Computing Surveys (CSUR), 54(5), 1-35.
>
> [2] Boiko, D. A., MacKnight, R., Kline, B., & Gomes, G. (2023). Autonomous chemical research with large language models. Nature, 624(7992), 570-578.
>
> [3] M. Bran, A., Cox, S., Schilter, O.,  Baldassari, C., White, A. D., & Schwaller, P. (2024). Augmenting  large language models with chemistry tools. Nature Machine Intelligence, 1-11.

---

> ### Author Response · Authors · 2024-11-25
> **Response to reviewer #BhtL - 4**
>
> > while the word "planning" is used several times in the text, it is only at l.444 that the authors explicit planning tasks as "the exploration of novel experimental goals". This definition is confusing. While "planning" generally refers to finding the succession of actions to obtain a predefined goal, exploring novel goals is different, and can be referred to as "goal babbling" for instance.
>
> Thanks for the insightful suggestion. The same concern was raised when we described the task as "novel experimental goal exploration". We did not come up with other better expressions, and it comes out that the terms "novel" and "exploration" are misleading. They seem to refer to an explorative process bootstrapping an unknown model without a predefined goal, which echoes the definition of goal babbling. However, in our context, these "novel experimental goals" are predefined by scientists before they are sent to self-driving laboratories for physical validation. The "exploration" is made by scientists and AI models for scientific discovery in their hypothesis forming phase. Therefore, in the physical validation phase, self-driving laboratories are given specific goals. We aim to empower them with the capability of designing protocols to achieve the predefined goals automatically. This requirement is aligned with the definition of planning and is distinct from that of goal babbling. To make the definition clear, we have revised the description to "confirmation of unverified experimental goals". We appreciate the reviewer for pointing out this ambiguity. We have revised the paper to improve the clarity of introducing this task.
>
> > In 2.2. the vocabulary used is confusing: a precondition is generally a property of the state that allows you to carry out your operations. It is a distinct notion from an input.
>
> Thanks for the comment. We deliberately employ the term "precondition" rather than "input" to demonstrate a sense of resource requirement for an operation. An operation can only be executed when the required reagents and intermediate products are available, otherwise, it must wait until these required resources are ready. This property indicates the dependence between operations, shaped by the reagent flows. It also functions as the prerequisite of our proposed reciprocative verification mechanism based on the dual representation of operations and reagents. We believe the term precondition in our context can be viewed as a property of the state that allows for operation execution, which is in line with the concept mentioned by the reviewer.
>
> We appreciate the reviewer for pointing this out and we have revised the paper to improve the clarity of introducing this concept.
>
> > Some terms are not defined. For instance : execution context (in l.190 which is used differently from execution condition) and key-value pairs (l193)
>
> Thanks for pointing these out. We have revised the paper to improve the clarity of these terms.
>
> > While the authors present comparative results of their representations and algorithms, there is no comparison with other approaches. How do the results compare quantitatively or qualitatively to the state of the art ?
>
> Thanks for the question. Automating the design of experiments is a relatively new domain, which was initially introduced by recent works in 2023 [1, 2]. In the previous literature search, we only find the current state-of-the-art work BioPlanner [3], which explicitizes the originally implicit experiment design process in previous works [1, 2]. As we have mentioned in the paper, our baselines are developed based on the methods proposed by these previous works. The Instance-Internal (II) designer is developed based on the state-of-the-art method of BioPlanner. The Flatten-Baseline (FB) and Instance-Baseline (IB) designers are developed based on the baselines being evaluated in [3].
>
> We appreciate the reviewer for pointing this out. We have revised the paper to enhance the links between the introduction of these baseline methods in the subsection "Machine designers" and our citations of these previous works in the section "Introduction".
>
> References:
>
> [1] Boiko, D. A., MacKnight, R., Kline, B., & Gomes, G. (2023). Autonomous chemical research with large language models. Nature, 624(7992), 570-578.
>
> [2] M. Bran, A., Cox, S., Schilter, O.,  Baldassari, C., White, A. D., & Schwaller, P. (2024). Augmenting  large language models with chemistry tools. Nature Machine Intelligence, 1-11.
>
> [3] O’Donoghue, O., Shtedritski, A.,  Ginger, J., Abboud, R., Ghareeb, A., & Rodriques, S. (2023,  December). BioPlanner: Automatic Evaluation of LLMs on Protocol Planning  in Biology. In Proceedings of the 2023 Conference on Empirical Methods in Natural Language Processing (pp. 2676-2694).

---

> ### Author Response · Authors · 2024-11-25
> **Response to reviewer #BhtL - 5.1**
>
> > While the authors mention as objectives the "exploration of novel goals", "generating novel experimental objectives" and aim to measure "a protocol's novelty", their metrics is only based on similarity measures. Diversity is not mentioned in the criteria. Could you add diversity measures or argue how the proposed metrics take into account diversity ?
>
> Thanks for the comment. According to professional standards for experimental protocols in the natural sciences [1], the fundamental components of an experimental protocol include the objectives of the experiment, the operations performed, the sequence of these operations, and the reagents or intermediate products used. The diversity of experiments arises from the various combinations of these elements. Within this context, our measurement method effectively captures the diversity of experimental protocols for the following reasons.
>
> Firstly, in measuring the novelty of an experimental protocol, we comprehensively evaluate its differences from existing protocols across multiple dimensions, including experimental objectives, operation sequences, and product flows. This ensures that the new protocol exhibits dissimilarity and diversity from existing ones at both the experimental design and execution step levels. Based on these evaluations, we classify novel protocols into three levels of novelty: planning, modification, and adjustment.
>
> Secondly, the subdivision of specific domains further ensures that our measurement method captures the diversity of experimental protocols within the same domain. The fundamental components of experimental protocols vary significantly across different domains (see Fig. 2). Therefore, when evaluating the novelty of a protocol, we primarily focus on its differences from existing protocols within the same domain. This approach prevents the exclusion of protocols that may share similar distributions across different domains.
>
> [Confusion matrices on operation distribution, product distribution and device distribution](https://anonymous.4open.science/api/repo/AutoDSL-Planning-Figure-0DFE/file/domain_heatmap.png?v=653dc994)
>
> We appreciate the reviewer’s emphasis on diversity. Measuring diversity among novel protocols is indeed both informative and meaningful. To address this, we have supplemented our analysis with a t-SNE visualization of the experimental objectives (described in natural language) for the novel protocols we selected. The results demonstrate a well-dispersed distribution, indicating a sufficient level of diversity among the protocols. We have made the revisions accordingly.
>
> [Diversity among novel protocols across four domains](https://anonymous.4open.science/api/repo/AutoDSL-Planning-Figure-0DFE/file/diversity.png?v=f4e8a566)
>
> References:
>
> [1] Bartley, B., Beal, J., Rogers, M.,  Bryce, D., Goldman, R. P., Keller, B., ... & Weston, M. (2023).  Building an open representation for biological protocols. ACM Journal on Emerging Technologies in Computing Systems, 19(3), 1-21.

---

> ### Author Response · Authors · 2024-11-25
> **Response to reviewer #BhtL - 5.2**
>
> **Planning**
> | Original protocol (title)                                    | Original protocol (description)                              | Novel protocol (title)                                       | Novel protocol (description)                                 | Interpretation                                               |
> | ------------------------------------------------------------ | ------------------------------------------------------------ | ------------------------------------------------------------ | ------------------------------------------------------------ | ------------------------------------------------------------ |
> | Protocol for the determination of intracellular phase separation thresholds | The objective of this protocol is to determine the thresholds for intracellular phase separation by quantifying the relationship between GFP intensity and stress granule initiation time in G3BP knockout U2OS cells. This approach enables the analysis of endogenous protein levels and their correlation with phase separation behavior, ultimately contributing to the understanding of the mechanisms underlying stress granule assembly and other membrane-less organelles. | Personality assessment protocol                              | This molecular biology protocol aims to assess the personality of western mosquitofish (Gambusia affinis) by evaluating their boldness, activity, and sociability using well-established experimental approaches. The protocol measures boldness as the latency to emerge from the shelter, activity by counting the number of squares crossed in the test arena, and sociability by determining the time spent near a group of conspecifics. | The two protocols differ completely: the old examines cellular phase separation in human cells, while the new focuses on fish personality behavior, showcasing distinct objectives, systems, and methods. |
> | Electrophysiological measurements of synaptic connectivity and plasticity in the longitudinal dentate gyrus network from mouse hippocampal slices | The objective of this protocol is to obtain acute longitudinal dentate gyrus slices from mouse hippocampi in order to measure extracellular excitatory postsynaptic potentials (fEPSPs) and to investigate the synaptic connectivity and plasticity of granule cells within the longitudinal dentate gyrus network. This involves techniques such as whole-cell patch clamping and two-photon imaging to explore the interactions between neighboring dentate gyrus granule cells and their synaptic relationships. | Animal models for depression-like and anxiety-like behavior  | The objective of this scientific protocol is to investigate depression-like and anxiety-like behaviors in animal models, specifically mice and rats, using various behavioral assays such as the Forced Swim Test, Tail Suspension Test, Elevated Plus Maze, Open Field Test, and Novelty Induced Hypophagia. The protocol aims to assess the effects of pharmacological interventions on these behaviors to better understand the underlying mechanisms of depression and anxiety. | The two protocols differ entirely: the old examines synaptic connectivity in hippocampal slices via electrophysiology, while the new uses behavioral assays to study depression and anxiety in live animals, reflecting distinct objectives and methodologies. |
> | Protocol for oral transplantation of maternal fecal microbiota to newborn infants born by cesarean section | The objective of this protocol is to facilitate the oral transplantation of maternal fecal microbiota to newborn infants delivered via cesarean section, aiming to restore beneficial gut microbiota that may be lacking due to the mode of delivery. This process involves meticulous recruitment, screening, preparation, and administration of the transplant, while ensuring the safety and health of both the mother and infant throughout the procedure. | Continuous monitoring of health data with a wearable device in pediatric patients undergoing chemotherapy for cancer – a feasibility pilot study | The objective of this feasibility pilot study is to continuously monitor health data using a wearable device (WD) in pediatric patients undergoing chemotherapy for cancer over a 14-day period. The study aims to assess the acceptance and effectiveness of the WD in this population, including the collection of data regarding side effects, daily activities, and overall experiences with the device. | The two protocols differ significantly: the old focuses on microbiota transplantation for newborn health, while the new explores wearable devices for monitoring pediatric cancer patients, reflecting distinct objectives, populations, and methodologies. |

---

> ### Author Response · Authors · 2024-11-25
> **Response to reviewer #BhtL - 5.3**
>
> **Modification**
> | Original protocol (title)                                    | Original protocol (description)                              | Novel protocol (title)                                       | Novel protocol (description)                                 | Interpretation                                               |
> | ------------------------------------------------------------ | ------------------------------------------------------------ | ------------------------------------------------------------ | ------------------------------------------------------------ | ------------------------------------------------------------ |
> | low Virometry for Characterizing the Size, Concentration, and Surface Antigens of Viruses | The objective of this protocol is to employ flow virometry (FVM) to characterize the size, concentration, and surface antigens of viral particles, specifically focusing on the replicative murine leukemia virus (MLV) that expresses an envelope-superfolder GFP fusion protein. This technique facilitates the detection and quantification of viral particles by combining light scatter and fluorescence measurements, thereby enabling detailed analysis of viral populations produced through various methods. | Wet-mount Method for Enumeration of Aquatic Viruses          | The objective of this scientific protocol is to provide a low-cost alternative method, called wet-mount, for enumerating aquatic viruses using epifluorescence microscopy. This method is efficient, rapid, precise, and appropriate for a wide range of viral concentrations that may be encountered in field and laboratory samples. | Both protocols focus on viral characterization but differ in techniques: the old uses flow virometry for particle analysis, while the new employs a wet-mount method with epifluorescence microscopy for aquatic viruses, adapting to different samples and equipment. |
> | The sequential isolation of metabolites, RNA, DNA, and proteins from a single, undivided mixed microbial community sample | The protocol aims to sequentially isolate and purify metabolites, RNA, DNA, and proteins from a single, undivided mixed microbial community sample, facilitating comprehensive biochemical analyses. By preserving the integrity of each biomolecular fraction throughout the extraction process, the workflow allows for a detailed investigation of microbial community composition and function. | BCP-MG: A Web Server for Predicting Bacterial Community of Metagenome | The protocol outlines the steps for using the BCP-MG web server to predict the bacterial community composition of metagenomic samples based on uploaded enzyme or reaction data. It enables researchers to choose between different metabolic databases and organism selection strategies to refine their predictions and analyze the microbial diversity within their samples. | Both protocols analyze microbial communities but differ in methods and objectives: the old uses sequential biomolecule extraction, while the new employs a web server to predict community composition from metagenomic data, shifting from experimental to computational approaches. |
> | Molecular profile to cancer cell line matchmaking            | The objective of this protocol is to establish a systematic approach for pairing cancer cell lines based on their molecular profiles, specifically focusing on shared therapeutic sensitivities and genomic similarities. By employing various analysis models and metrics, the protocol aims to enhance the understanding of molecular features that correlate with treatment response in cancer therapies. | A multistep computational procedure to identify candidate master Transcriptional Regulators (TRs) of glioblastoma (GBM) | The objective of this protocol is to identify candidate master transcriptional regulators (TRs) of glioblastoma (GBM) by reconstructing a regulatory network using gene expression data and epigenetic information. This process involves scoring the TRs based on their regulatory activity in GBM stem cells and differentiating cells, ultimately identifying those with significant regulatory effects on gene expression. | Both protocols analyze molecular features in cancer but differ in focus and methods: the old matches cancer cell lines by molecular profiles for therapy, while the new identifies transcriptional regulators in glioblastoma via network reconstruction, shifting the analytical approach. |

---

> ### Author Response · Authors · 2024-11-25
> **Response to reviewer #BhtL - 5.4**
>
> **Adjustment**
> | Original protocol (title)                                    | Original protocol (description)                              | Novel protocol (title)                                       | Novel protocol (description)                                 | Interpretation                                               |
> | ------------------------------------------------------------ | ------------------------------------------------------------ | ------------------------------------------------------------ | ------------------------------------------------------------ | ------------------------------------------------------------ |
> | Re-using Criterion plastic precast gel cassettes for SDS-polyacrylamide electrophoresis. | The objective of this protocol is to outline the steps for re-using Criterion plastic precast gel cassettes to prepare and pour SDS-polyacrylamide gels using agarose, ensuring proper sealing and preventing leaks during the gel formation process. It provides detailed instructions for assembling the cassettes, preparing the agarose solution, and performing tests to confirm the integrity of the gel structure before electrophoresis. | 1% Agarose Gel Electrophoresis Prep                          | The objective of this molecular biology protocol is to prepare a 1% agarose gel for electrophoresis and utilize it for genomic DNA quality checking. This protocol details the steps for gel preparation, sample preparation, and gel loading, allowing researchers to assess the quality of their DNA samples. | Both protocols utilize gel electrophoresis, but they differ in their specific objectives: the old protocol focuses on reusing precast gels for protein analysis, while the novel protocol prepares agarose gels for DNA quality assessment. The core technique is similar, but the purpose and context differ, indicating an adjustment rather than a significant modification. |
> | Immunohistochemistry and in situ hybridization protocols     | The objective of this protocol is to perform immunohistochemistry on cryostat sections and in situ hybridization using T7-PCR based probes to visualize specific proteins and mRNA expression patterns in tissue samples. This combined approach allows researchers to study the localization and abundance of target molecules within their biological context. | Rtl1 and CD31 double-immunohistochemistry                    | The objective of the 'Rtl1 and CD31 double-immunohistochemistry' protocol is to detect and visualize the expression of Rtl1 (Peg11) and CD31 (Pecam-1) in placental tissue sections using a dual fluorescence approach. This method allows for the localization and assessment of these proteins in the context of placental morphology under a fluorescence microscope. | Both protocols focus on visualizing specific molecular markers in tissue sections using immunohistochemistry techniques. The old protocol combines immunohistochemistry with *in situ* hybridization, while the novel protocol applies a dual fluorescence method to target specific proteins (Rtl1 and CD31). Although the markers and detection methods differ, the overall approach and objectives are similar, making this an adjustment within the scope of existing techniques. |
> | Functional and Morphological Assessment of Diaphragm Innervation by Phrenic Motor Neurons | The objective of this protocol is to assess the functional and morphological characteristics of diaphragm innervation by phrenic motor neurons through compound muscle action potential (CMAP) recordings and detailed analysis of neuromuscular junction (NMJ) morphology in the hemi-diaphragm of rats. By quantifying the presence and conditions of NMJs, such as intactness and innervation status, the protocol aims to elucidate the integrity of phrenic nerve innervation and potential compensatory mechanisms in response to denervation. | Measuring Diaphragm Thickness and Function Using Point-of-Care Ultrasound | The objective of this protocol is to measure diaphragm thickness and function during tidal breathing and maximal inspiratory efforts using point-of-care ultrasound, allowing for assessment of diaphragm contractility and respiratory mechanics. This technique facilitates the evaluation of diaphragm thickness (Tdi) and thickening fraction (TFdi), providing valuable insights into respiratory muscle performance in various clinical settings. | Both protocols are focused on assessing diaphragm function, but they use different methods: the old protocol involves electrophysiological recordings and morphological analysis of neuromuscular junctions, while the novel protocol utilizes point-of-care ultrasound to measure diaphragm thickness and contractility. Despite the differences in techniques, the overall objective of studying diaphragm functionality and health remains similar, indicating an adjustment in methodology rather than a major shift in research focus. |

---

> ### Author Response · Authors · 2024-11-25
> **Response to reviewer #BhtL - 6**
>
> > Figures need a description to better understand what is shown
>
> Thanks for pointing these out. We have revised the paper to improve the readability of the figures. We have also revised the references to the figures in the main text.
>
> > l.234: "Any status transition of the product flow is caused, and is only caused, by the effects of operations". This stance excludes general evolving systems. What about dynamic systems, including with slow transformations, especially in biology or ecology?
>
> Thanks for the comment. As our objective is to automate the design of protocols for self-driving laboratories, we decided to treat these machine executable operations as *monitors*. For example, the operation "cultivate" is not an impulse-style instant which may consecutively leave the cells alone for 72 hours, where the cells are under slow transformations. Instead, "cultivate" comes with a property of *duration = 72 hours*, thereby covering the process of transformation. By incorporating these interval-style features into operations [1], we try to treat all systems as static systems.
>
> However, we recognize that the claim of "any status transition" is much too ambitious. There should exist general evolving systems that come out of our consideration. Therefore, we relaxed this claim to maintain the rigorousness of the paper. We appreciate the reviewer for pointing this out and have revised the paper accordingly.
>
> References:
>
> [1] Kuipers, B. (1994). Qualitative reasoning: modeling and simulation with incomplete knowledge. MIT press.
>
> > l.440 : "the three scenarios of protocol design introduced in sec 1". It seems sec 1 introduces 3 representations/levels of encapsulation.
>
> Thanks for pointing out this ambiguity. This sentence "the three scenarios of protocol design introduced in sec 1" aims to refer to the three purposes of experiment design requirements: (i) confirmation of unverified objectives to seek specific findings; (ii) testing parallel hypotheses or solutions; and (iii) replication of established experiments within the constraints of available laboratory resources.
>
> We appreciate the reviewer for pointing this out. We have revised the paper accordingly to enhance the clarity of this cross reference.
>
> > The results should report computing load
>
> Thanks for the comment. For automated representation generation, we primarily used GPT-4o mini with OpenAI’s Batch API for preprocessing, incurring a cost of approximately \$60 across four domains. The design of the DSL was executed on a MacBook with an M2 chip, running 1,000 iterations to ensure convergence. This process required an average of 55 seconds per iteration for the operation-centric view DSL and an average of 2 seconds per iteration for the product-centric view DSL. For the machine designer, we primarily utilized GPT-4o mini combined with RAG for design, with a total cost of approximately \\$10 (7 methods, 140 protocols). In summary, the overall computational load is relatively low, highlighting the accessibility of our machine designers when utilizing the proposed representations and the corresponding automatic representation generation modules. We have made the revisions to clarify this point.
>
> > Typos
>
> Thanks for pointing out these typos. We have made the revisions accordingly.

---

### Author Response · Authors · 2024-11-25
**General Response**

We thank all reviewers for their time and valuable comments. The feedback is both substantial and helpful for improving our paper. In this work, we study the representations for protocol design, to fully elicit the capabilities of knowledge-based machine designers, such as Large Language Models, on this task. Accordingly, we propose a multi-faceted, multi-scale representation, where instance actions, generalized operations, and product flow models are hierarchically encapsulated using Domain-Specific Languages. We further develop a data-driven algorithm that autonomously customizes these representations for specific domains. Our qualitative and quantitative results underscore the potential of the representation to serve as an auxiliary module for Large Language Models, in the realm of machine-assisted scientific exploration.

We would like to thank the reviewers for acknowledging our work to be:
1. The paper identifies "a relevant and timely problem, which is much-needed and of high added value" (reviewer #Xeav), "is well-motivated with accurately captured significance" (reviewer #LMMv), with "extremely compelling premise" (reviewer #WJzq), and "ambitiously represents a synthetic representation of experimental protocols across multiple sciences" (reviewer #BhtL).
2. The proposed representation "is particularly elegant" that treats "protocol modeling as conditional probabilities" (reviewer #WJzq), and "is especially clever" regarding "the dual verification system" (reviewer #WJzq), which "is well presented" and "appears quite novel" (reviewer #LMMv), thereby "contributes to the state of the art" (reviewer #Xeav).
3. The experiments "are well explained" with "interesting choice of metrics to assess protocol designers" (reviewer #LMMv), which is "sound and convincing" (reviewer #Xeav), demonstrating "practical utility and broad applicability" (reviewer #WJzq) through "clear presentation and discussion" (reviewer #Xeav).

Based on the reviewers' comments, we made the revisions including:

1. Clarifying specific concepts to enhance the paper's accessibility for readers with a background outside experimental sciences.
2. Demonstrating running examples of the machine designers equipped with our resulting representations in detail to make the paper more intuitive and comprehensive.
3. Conducting additional analyses and discussions regarding the computational complexity, rationality, scalability, and generality of our proposed framework to make the paper more rigorous and self-consistent.
4. Improving the readability of the paper through fixing typos, resolving ambiguities, enlarging the font size of the text in the plots, and rewriting the captions and references of the figures.

We have highlighted the changed part of the text with red color in the new version of the paper pdf file.

In the following, we address specific questions for each reviewer.

---

### Meta-Review · Area_Chair_Rxkf · 2024-12-19

**Metareview:**

This paper introduces a framework for automated scientific protocol design in self-driving labs. Unlike existing systems that only execute predefined protocols, this system automatically designs new ones using a hierarchical representation which captures protocols at three levels: actions, operations, and material flows, each encoded using domain-specific languages. The resulting protocols are verifiable (although only manually), and the system is extensively tested across three tasks in four different domains. The hierarchical representation consistently outperforms an ablation without the suggested hierarchy.

All reviewers agreed that this paper is tackling an interesting and important challenge which is well motivated in the introduction. Several reviewers also felt that the particular method described here was elegant and clever, and appreciated the extensive evaluation of the method across domains and tasks.

Several reviewers also mentioned that there were parts of the paper's presentation which could be improved. For many of these points, the authors adapted the paper during the revision period to address some of these presentation points, including better situating the work in some of the relevant literature. However, the authors should ensure that the remaining points are addressed before the camera ready deadline.

Given that all reviewers agreed that the paper tackles an important challenge with a novel method validated by experiments, I recommend acceptance.

**Additional Comments On Reviewer Discussion:**

The paper was significantly updated during the rebuttal period. As mentioned above, several reviewers brought up issues of presentation (especially the figure captions, problem formalization, and results), and the authors changed the paper accordingly (including adding large results sections to the appendix, adding expanded and descriptive captions for all figures, and completely revising the discussion section). The updated paper is significantly better than the initially submitted version.

---

### Decision · Program_Chairs · 2025-01-22

Accept (Poster)